

# N₂O changes from the Last Glacial Maximum to the preindustrial - part II: terrestrial N₂O emissions constrain carbon-nitrogen interactions

Fortunat Joos[1], Renato Spahni[1], Benjamin D. Stocker[2], Sebastian Lienert[1], Jurek Müller[1], Hubertus Fischer[1], Jochen Schmitt[1], I. Colin Prentice[3], Bette Otto-Bliesner[4], Zhengyu Liu[5]

[1]Climate and Environmental Physics, Physics Institute and Oeschger Centre for Climate Change Research, University of Bern, Bern, CH-3012, Switzerland
[2]CREAF, E08193 Bellaterra (Cerdanyola del Vallès), Spain
[3]AXA Chair of Biosphere and Climate Impacts, Department of Life Sciences, Imperial College London,Silwood Park Campus, Buckhurst Road, Ascot SL5 7PY, UK
[4]Climate and Global Dynamics Division, National Center for Atmospheric Research, Boulder, CO 80307-3000, USA
[5]Atmospheric Science Program, Department of Geography, Ohio State University, OH 43210, USA

*Correspondence to*: Fortunat Joos (joos@climate.unibe.ch)

**Abstract.** Land ecosystems currently take up a quarter of the human-caused carbon dioxide emissions. Future projections of this carbon sink are strikingly divergent, leading to major uncertainties in projected global warming. This situation partly reflects our insufficient understanding of carbon-nitrogen (C-N) interactions and particularly of the controls on biological N fixation (BNF). It is difficult to infer ecosystem responses for century time scales, relevant for global warming, from the comparatively short instrumental records and laboratory or field experiments. Here we analyse terrestrial emissions of nitrous oxide (N₂O) over the past 21,000 years as reconstructed from ice-core isotopic data and presented in part I of this study. Changing N₂O emissions are interpreted to reflect changes in ecosystem N loss, plant available N, and BNF. The ice-core data reveal a 40 % increase in N₂O emissions over the deglaciation, suggestive of a highly dynamic global N cycle whereby sources of plant-available N adjust to meet plant N demand and loss fluxes. Remarkably, the increase occurred in two steps, each realized within maximum two centuries, at the onsets of the northern hemisphere warming events around 14,600 and 11,700 years ago. We applied the LPX-Bern dynamic global vegetation model in deglacial simulations forced with Earth System Model climate data to investigate N₂O emission patterns, mechanisms, and C-N coupling. The reconstructed increase in terrestrial emissions is broadly reproduced by the model, given the assumption that BNF positively responds to increasing N demand by plants. In contrast, assuming time- and demand-independent levels of BNF in the model to mimic progressive N limitation of plant growth results in N₂O emissions that are incompatible with the reconstruction. Our results suggest the existence of (a) strong biological controls on ecosystem N acquisition, and (b) flexibility in the coupling of the C and N cycles during periods of rapid environmental change.



## 1 Introduction

Nitrous oxide (N₂O) is a sensitive proxy of biogeochemical and ecosystem processes on land and in the ocean, and its past atmospheric variations are recorded in ice cores. N₂O is an important greenhouse gas and contributes to ongoing global warming (Stocker et al., 2013). It is also involved in the destruction of stratospheric ozone (Myhre et al., 2013). N₂O is produced primarily by nitrification and denitrification both on land and in the ocean, and photochemically decomposed in the stratosphere (Ciais et al., 2013). Atmospheric N₂O increased from 270 ppb (MacFarling Meure et al., 2006) to around 330 ppb (https://www.esrl.noaa.gov/gmd/) over the industrial period due to human activities including fertilizer application, fossil fuel use and biomass burning (Bouwman et al., 2013;Ciais et al., 2013). Atmospheric N₂O varied naturally between around 180 and 300 ppb over glacial-interglacial cycles (Sowers et al., 2003;Spahni et al., 2005;Schilt et al., 2010). A quantitative explanation of these variations is lacking and this knowledge gap renders projections of the feedbacks between N₂O and climate change uncertain (Stocker et al., 2013;Battaglia and Joos, 2018;Kracher et al., 2016).

Variations in N₂O emission, and thus in tropospheric N₂O content, are closely linked to ecosystem processes governing the cycling of nitrogen (N) and carbon (C) on land and in the ocean (Gruber and Galloway, 2008). N availability to support land plant and phytoplankton growth and terrestrial and marine C storage is governed by the balance of N input and loss fluxes. Reactive N is added by biological nitrogen fixation (BNF) and, on land, by deposition, and weathering. Reactive N is lost from ecosystems through nitrification followed by denitrification, as well as leakage and mineral adsorption. Terrestrial N₂O emissions are a sensitive indicator of the flow of reactive N entering and leaving land ecosystems (Firestone and Davidson, 1989). Reconstructions of past variations in terrestrial and marine N₂O emissions from ice-core N₂O concentration and isotopic data (Schilt et al., 2014) provide information on the functioning of ecosystems and the coupled C-N cycle. They provide the opportunity to evaluate C-N-climate models, and to test alternative hypotheses for underlying ecosystem processes, such as the limitation of plant growth by N limitation and the responses of BNF to climatic and environmental change.

The land biosphere sequesters about a quarter of anthropogenic CO₂ emissions and is the largest natural N₂O source (Ciais et al., 2013). Yet key features of the C-N cycle are poorly understood, leading to major uncertainties in global warming projections (Joos et al., 2001;Arora et al., 2013;Friedlingstein et al., 2006;Friedlingstein et al., 2013;Plattner et al., 2008;Zickfeld et al., 2013). The question whether N availability will limit future land C uptake and N₂O emissions is unresolved. In particular, large uncertainties remain as to what extent BNF, net N mineralization, and N uptake will adjust to support plant growth under climatic and environmental change (Niu et al., 2016). It is debated how global warming, increasing CO₂, and increased N deposition affect the current land carbon sink (Körner, 2015;Fatichi et al., 2014;Terrer et al., 2016) and how the land carbon sink and greenhouse gas emissions will evolve. Apparently conflicting observations (Davidson et al., 2007;Luo et al., 2004;Reich et al., 2014;Vitousek et al., 2013;Xu-Ri and Prentice, 2008), theories (Zhu et al., 2017;Menge et al., 2017), and model projections (Hungate et al., 2003;Todd-Brown et al., 2013;Wieder et al., 2015b) of the role of N limitation




for plant growth and the land C sink (Meyerholt et al., 2016;Zaehle et al., 2014;Walker et al., 2015) represent a major uncertainty in future projections of atmospheric $CO_2$ and climate (Jones et al., 2013;Joos et al., 2013;Todd-Brown et al., 2013). Results from free-air $CO_2$ enrichment (FACE) and open-top chamber experiments show that N limitation at least in some ecosystems reduces the response of aboveground biomass growth to elevated $CO_2$ (McMurtrie et al., 2008;Norby et al., 2010;Terrer et al., 2016). Increased N immobilization in plant litter, biomass and soil (Luo et al., 2004) and multiple nutrient limitation (Körner, 2015;Vitousek et al., 2010;Elser et al., 2007) have been put forward as explanations of reduced growth responses. A recent synthesis of the results from $CO_2$ enrichment experiments suggests that mycorrhizal association exerts an important control on the magnitude of the realized $CO_2$ fertilization effect (Terrer et al., 2016).

Field and laboratory experiments provide important insights, but extrapolation of such results on the short-term response in BNF and C uptake to the multi-decadal-to-century time scales, relevant for global warming projections, is uncertain. For example, Reich et al. (2018) found an unexpected reversal of $C_3$ versus $C_4$ grass response to elevated $CO_2$ and shifts in soil N mineralization rates during a 20-year field experiment. These authors concluded that even the best-supported short-term drivers of plant response to global change might not predict long-term results. Additional hindrances to the improvement of our quantitative understanding of C-N cycle coupling are related to large variations in fluxes and inventories on small spatial and temporal scales (Arias-Navarro et al., 2017;Barton et al., 2015) and the diversity of responses across different organisms and ecosystems. Largely missing are long-term observational constraints on the global coupled C and N cycle that might permit us to explore and test alternative hypotheses regarding the degree of N limitation during periods of rapid climate change and increasing atmospheric $CO_2$.

High-resolution data on the isotopic composition of $N_2O$ from Antarctic ice cores have the potential to provide precise information on past variations in terrestrial and marine $N_2O$ emissions and thus on C-N coupling on time scales from decades to many centuries. A recent ice-core study on the stable isotope composition of $N_2O$ demonstrated the power of this approach (Schilt et al., 2014). The isotopic data showed that both land and oceanic sources increased during the interval from 16 to 10 thousand years before present (ka BP), when ocean circulation and climatic changes strongly affected the global cycling of both $CO_2$ and $N_2O$ (Schilt et al., 2014). Schilt et al. concluded that natural $N_2O$ emissions will probably increase in response to global warming. In part I of this study (Fischer et al., 2019), this earlier work was extended to reconstruct the evolution in terrestrial versus oceanic emissions of $N_2O$ from the Last Glacial Maximum (LGM; 21 ka BP) to the late preindustrial Holocene using a novel $N_2O$ stable isotope record. The ice core data reveal large step-like changes in terrestrial emissions at the onset of warming events, realized over decades and, given the proxy data resolution, last maximum 200 years. But a detailed process-based investigation of terrestrial $N_2O$ emission changes over the deglaciation, rapid past warming events, and the Holocene warm period, and their links to the flow of N and C in land ecosystems, has not been carried out before.



The aim of part II of this study is to improve our understanding of the cycles of $N_2O$, C and N. We use terrestrial $N_2O$ emissions as a proxy for the flow of reactive N entering and leaving land ecosystems and thus implicitly as a constraint for changes in biological N fixation and soil N availability for plants. In other words, reconstructed terrestrial $N_2O$ emissions are used to shed further light on the N limitation of terrestrial ecosystems, and the land C sink, on a global scale. The unique, new terrestrial

$N_2O$ emission record of the past 21 kyr is used to explore and test alternative mechanisms of the functioning of the C-N cycle on land and to quantify terrestrial drivers for atmospheric $N_2O$ concentration changes. Controls on variations in terrestrial emissions are elucidated in the framework of a dynamic global vegetation model.  Part I of this study (Fischer et al., 2019) presents the ice core $N_2O$ concentration and isotope data and the reconstruction of global terrestrial and marine $N_2O$ emissions for the past 21,000 years and provides a discussion on the reconstructed marine emissions in the context of past climate and

oceanographic changes. Here in part II we focus on the interpretation of the terrestrial $N_2O$ emission record using explicit terrestrial $N_2O$ emissions in a dynamical vegetation/terrestrial biogeochemistry model.

## 2 Introduction to $N_2O$, terrestrial nitrogen and carbon flows and working hypotheses

### 2.1 $N_2O$ budget and production mechanisms

Prather et al. (2015) estimated that the pre-industrial atmospheric lifetime of $N_2O$ was 123 years. Together with the atmospheric

$N_2O$ concentration from ice cores, this figure constrains the total net pre-industrial $N_2O$ source to $10.5 \pm 1$ Tg N yr$^{-1}$. Marine $N_2O$ emissions were recently estimated by an observation-constrained approach, using water-column and surface $N_2O$ observations as targets, to be 4.6 ($\pm$ 1 standard deviation range: 3.1 to 6.1) Tg N yr$^{-1}$. This calculation implies a natural terrestrial $N_2O$ source of 5.9 (4.1 to 7.7) Tg-N yr$^{-1}$ (Battaglia and Joos, 2018), in line with IPCC AR5 estimates of 6.6 (3.3 to 9.0) Tg N yr$^{-1}$.

$N_2O$ is produced through a variety of pathways both in the ocean and on land and $N_2O$ production is closely linked to the flows of C and N (Wrage et al., 2001;Chapuis-Lardy et al., 2006;Kato et al., 2013;Battaglia and Joos, 2018;Trimmer et al., 2016;Babbin et al., 2015;Gilly et al., 2013;Bange, 2008;Butterbach-Bahl et al., 2013;Firestone and Davidson, 1989). The dominant pathways of net $N_2O$ production are thought to be respiratory denitrification ($NO_3^- \rightarrow NO_2^- \rightarrow NO \rightarrow N_2O \rightarrow N_2$)

under low oxygen conditions, and autotrophic nitrification ($NH_3 \rightarrow NO_2^- \rightarrow NO_3^-$), mediated by archaea, bacteria and fungi. In addition heterotrophic nitrification, which is the oxidation of organic N to nitrite ($NO_2^-$) and subsequent reduction to $N_2O$ by incomplete denitrification, was found to be the dominant path in a grassland ecosystem after fertilizer application (Moser et al., 2018). $N_2O$ is produced as a byproduct of nitrification and as an intermediate product during denitrification. $N_2O$ is also produced from nitrite through nitrification, often termed nitrifier-denitrification, through anaerobic ammonium oxidation,

chemautotrophic denitrification, abiotic processes or from photoautotrophic organisms in cryptogamic covers (Lenhart et al., 2015;Butterbach-Bahl et al., 2013).



Terrestrial $N_2O$ production and emissions depend sensitively on environmental factors including precipitation, soil temperature, soil moisture, soil texture, soil oxygen concentration or pH, topography as well as on substrate and nutrient availability and on nutrient addition by deposition or fertilizer application (Zhuang et al., 2012;Stehfest and Bouwman, 2006;Wang et al., 2017). Field data- and model-based emission estimates show the highest emissions of $N_2O$ in moist tropical

areas, and lower emissions in high latitudes (Stehfest and Bouwman, 2006;Zhuang et al., 2012;Werner et al., 2007;Potter et al., 1996;Xu et al., 2017;Xu-Ri et al., 2012;Wells et al., 2018). Moist soils typically show relatively high $N_2O$ emission (Butterbach-Bahl et al., 2013;Zhuang et al., 2012), but the environmental dependencies of $N_2O$ emission are often complex (Diem et al., 2017;Müller et al., 2015;Schmid et al., 2001a;Matson et al., 2017) and dependent on the production pathway (Kool et al., 2011). Higher $N_2O$ fluxes at high soil water contents have been reported from laboratory and field studies, and

linked to increasing denitrification activity in response to reduced oxygen diffusion into the soil (Arias-Navarro et al., 2017). The sensitivity of denitrification to temperature is found to be higher than for $CO_2$ emissions from soil organic matter decomposition, and a positive feedback of soil warming on $N_2O$ emissions is expected (Butterbach-Bahl et al., 2013). Warming treatment increased measured $N_2O$ emissions in boreal peatlands (Cui et al., 2018). However, responses to warming treatment are found to be highly variable across a range of conditions and ecosystems and it remains unclear whether warming will

increase or reduce regional-to-global $N_2O$ emissions (Dijkstra et al., 2012). Positive or neutral responses in $N_2O$ emissions have been found in field experiments under elevated $CO_2$ in temperate and boreal forests and grasslands (van Groenigen et al., 2011;Dijkstra et al., 2012;Regan et al., 2011;Moser et al., 2018;Zhong et al., 2018). Nitrogen addition by mineral and organic fertilizer causes enhanced $N_2O$ emissions. Emission factors are reported to depend sensitively on soil pH and are typically estimated to be around 0.5 % to 2 % of added N (Charles et al., 2017;Wang et al., 2017), but may vary by more than an order

of magnitude.

## 2.2 C-N-$N_2O$ coupling

The main flow paths of reactive N and its link to $N_2O$ production and C flows on land (Gruber and Galloway, 2008;Butterbach-Bahl et al., 2013;Vitousek et al., 2013;Zähle, 2013;Firestone and Davidson, 1989) are schematically sketched in Fig. 1. These ecosystem flows can be assigned to an internal and an external N cycle. Within an ecosystem (green arrows in Fig. 1), N is

primarily taken up in the form of $NH_4^+$ and $NO_3^-$ by plant roots to support growth, while a large fraction of reactive N is taken up by soil microbes and fungi and is thus immobilized. Organic N is converted back to inorganic N during the mineralization of litter and soil organic matter. Gross N mineralization is thereby modified by the decomposers carbon-use efficiency (Manzoni et al., 2008). Net N mineralization (as in Fig. 1) thus reflects the modified gross N mineralization minus N immobilization. The $NO_3^-$ pool is replenished by nitrification, the conversion of $NH_4^+$ to $NO_3^-$. We note the acid-base

equilibrium between $NH_4^+$ and $NH_3$ in soil water; for simplicity, we generally refer to $NH_4^+$ only.

Turning to the external cycle, reactive N enters land ecosystems (blue arrows in Fig. 1) through the conversion of dinitrogen ($N_2$) to organic N and eventually to $NH_4^+$ by BNF (Cleveland et al., 1999;Vitousek et al., 2013;Zähle, 2013;Sullivan et al.,





2014;Xu and Prentice, 2017), through rock weathering (Houlton et al., 2018) and the deposition (Lamarque et al., 2011;Vet et al., 2014;Dentener et al., 2006) of $NH_x$ and $NO_y$ (including sources by lightning). Reactive N is lost from land ecosystems through gaseous losses (including $N_2O$), leaching of $NO_3^-$ and dissolved organic matter by runoff, and emissions of N compounds by fire (Bouwman et al., 2013;Ciais et al., 2013;Hedin et al., 1995;Hedin et al., 2003). In equilibrium, losses of

reactive N from ecosystems have to be compensated by inputs, mainly by BNF, on decadal to millennial time scales.

The external and internal N cycles are coupled. Reactive N in mineral forms serves as substrate for the N loss fluxes (external cycle) as well as a reservoir for plant N uptake (internal cycle). Following mass balance, mineral N concentrations change until the balance of N input by BNF and other sources matches N loss and net ecosystem N uptake (reactive N uptake minus

net N mineralization). For example, in a growing ecosystem an increasing amount of N is taken up by plants and converted from inorganic to organic forms. This net ecosystem N uptake tends to deplete reactive N in mineral forms. Correspondingly, N loss fluxes (including $N_2O$) would decrease and N limitation of plant growth would increase if N sources such as BNF do not adjust to the increasing ecosystem N demand. Whether N limitation increases or not in a growing ecosystem depends therefore critical on the flexibility of N input, hence BNF.

Empirical evidence from N-addition experiments, synthesized by Lue et al. (2011) and Niu et al. (2016), shows that the uptake of N by plants, net primary productivity and biomass as well as $NH_x$ and $NO_3^-$ pools in soils, nitrification, nitrate leaching, denitrification, and $N_2O$ emissions all increase simultaneously with N input. This finding points to a tight coupling between the availability of reactive N for plant growth, nitrification and denitrification fluxes and $N_2O$. $N_2O$ production on land is

predominantly associated with denitrification, and to a smaller extent with nitrification. Large gaseous losses and $N_2O$ emissions are thus indicative of a large N throughput (input and loss), where ecosystem functioning may have adjusted by exploiting energetically costly, but evolutionarily advantageous BNF to compensate for the losses (Batterman et al., 2013;Pons et al., 2007;Vitousek and Hobbie, 2000) or where losses are compensated by N input from weathering or deposition.

### 2.3 Working hypotheses

In a N-limited land ecosystem, N that becomes available through mineralization or deposition is expected to be quickly taken up by plants to support their growth. In turn, the pools of reactive N in soils remain small. As a consequence, $N_2O$ production is expected to be small in ecosystems with severely N-limited biomass growth and a correspondingly "closed" N cycle. On the other hand, in an ecosystem with abundant reactive N supply, mineralized N not used for biomass growth enriches the soil pools of reactive N and is eventually converted by nitrification and denitrification and lost from the ecosystem. Thus, $N_2O$

production is expected to be high in such an "open" (or "leaky") system, where plant growth is not or only weakly limited by N availability and where N input by BNF and other sources is high. These two situations are consistent with contrasting setups of the "Hole-in-the-pipe" model (Firestone and Davidson, 1989;Davidson et al., 2000) with low N flow entering and leaving ecosystems and correspondingly low $N_2O$ production in the closed case and high N flow and $N_2O$ production in the open case



(Fig. 1). Generally, mid- and high latitude ecosystems are considered to be more generally N limited than tropical ecosystems. Yet the role of BNF in potentially alleviating N limitation, and the trade–offs among N fixation and N use efficiency, soil N uptake, and plant turnover remain unclear (Menge et al., 2017).

5   To guide further discussion, we formulate two extreme "end-member" working hypotheses for the temporal evolution of the carbon and nitrogen cycle over the last 21,000 years on the global scale:

*Hypothesis I* postulates an "open" or "flexible" terrestrial N cycle whereby sources of reactive N on land increased under warming climate and increasing $CO_2$ over the deglacial period, contributed to meet the increasing N demand of plants under 10   more favorable growth conditions, and, in turn, resulted in a higher flow of N entering and leaving land ecosystems and increased $N_2O$ production from terrestrial ecosystems.

*Hypothesis II* postulates a "closed" or "inflexible" terrestrial N cycle whereby N sources did not adjust to environmental change, land ecosystems remained or became increasingly N-limited over the deglacial period, and consequently terrestrial 15   $N_2O$ production remained small and marine emissions dominated past atmospheric $N_2O$ changes.

The question posed in this study is not to what extent different ecosystems are, or have been, N limited. Rather, we ask whether BNF and the N cycle adjusted dynamically to (at least partly) meet increasing N demand by plants (hypothesis I) or not (hypothesis II) and address this question by analysing temporal changes in terrestrial $N_2O$ emissions. We note that the ice core 20   record provides a globally integrated signal; spatially differentiated responses are not resolved. Last, but not least, the ice core terrestrial $N_2O$ emission record provides information for time scales of a century or longer – potentially giving sufficient time for ecosystems to adjust.

## 3 Methods

### 3.1 The LPX-Bern(v1.4) Dynamic Global Vegetation Model

25   The dynamic global vegetation and land surface process model LPX-Bern ("Land surface Processes and eXchanges" model as implemented at the University of Bern, version 1.4) (Lienert and Joos, 2018a) is applied here in transient mode over the last 21,000 years (Ruosch et al., 2016). The LPX-Bern model describes dynamical vegetation and terrestrial biogeochemical processes, integrates representations of non-peatland (Gerten et al., 2004;Joos et al., 2004;Sitch et al., 2003) and peatland (Spahni et al., 2013;Wania et al., 2009) ecosystems and their C and N dynamics (Stocker et al., 2013;Xu-Ri and Prentice, 30   2008;Xu-Ri et al., 2012), and describes the dynamic evolution of wetland and peatland extent (Stocker et al., 2014). The model



calculates the release and uptake of the trace gases $CO_2$, $N_2O$ (Stocker et al., 2013;Xu-Ri and Prentice, 2008;Xu-Ri et al., 2012) and $CH_4$ (Spahni et al., 2011;Wania et al., 2010;Zürcher et al., 2013).

Vegetation is represented by plant functional types (PFTs) that are in competition for resources (water, light, N) on each grid cell. Here a version with fifteen PFTs is used. Eight generic tree PFTs, and PFTs for C3 and C4-type grasses grow on natural land (excluding peat) and former peat. Two PFTs representing peat mosses and flood-tolerant C3 graminoids as well as three flood-tolerant tropical PFTs grow on peat and wetlands. The model accounts for the dynamic coupling of C and water cycles through photosynthesis and evapotranspiration, which also defines plant water use efficiency (Saurer et al., 2014;Keller et al., 2017). Seven C and N pools per PFT represent leaves, sapwood, heartwood, fine roots, aboveground leaf litter, aboveground woody litter, and belowground litter. Separate soil organic C and N pools receive input from litter of all PFTs. LPX uses a vertically resolved soil hydrology, heat diffusion and an interactive thawing–freezing scheme (Wania et al., 2009).

The LPX-Bern vegetation and soil components interact with a dynamic N-cycle module (Xu-Ri and Prentice, 2008;Xu-Ri et al., 2012), here modified to include N immobilization in soils (Bengtsson et al., 2003;Li et al., 2017;Gütlein et al., 2017). The module describes the relevant N and $N_2O$ fluxes and pools for plants and soils as schematically depicted in Fig. 1 and briefly summarized below. For a detailed description, justification, and further references we refer to Xu-Ri and Prentice (2008) and Xu-Ri et al. (2012).

$N_2O$ emissions are computed by assuming that fractions of the N fluxes associated with denitrification, nitrification and N leaching are released as $N_2O$. For denitrification, the globally-dominant $N_2O$ source path, this fraction is the product of a constant (*RN2ODN*) and a temperature-dependent factor $f$(T). $f$ is unity at 22°C and its value roughly doubles for a temperature increase of 10°C. The amount of $N_2O$ released per unit N denitrified is therefore higher in warm than in cold regions. The fraction of $N_2O$ production from denitrification has been observed to be in the range of 0.2–4.7 % of the denitrification rate (see references in (Xu-Ri and Prentice, 2008)). Here, the constant *RN2ODN* is set to 1.46 % of the denitrification rate in order to arrive at pre-industrial (1500 CE) emissions of 5.9 TgN yr$^{-1}$ (Battaglia and Joos, 2018). The corresponding yield fractions for $N_2O$ emissions from nitrifications and leaching are, in the absence of better information and for simplicity, assumed to be temperature independent and constant across space and time. They are set to 0.05 % and 0.5 % of the respective N fluxes (Stocker et al., 2013).

Denitrification is modelled as a two-step process whereby $NO_3^-$ is converted to nitrite ($NO_2^-$) and $NO_2^-$ is further converted to $N_2$, following Michaelis-Menten kinetics with the substrates $NO_3^-$, $NO_2^-$, and labile C availability. Reaction rates are again temperature dependent following $f$. Nitrification is assumed to be proportional to the $NH_4^+$ soil pool with a temperature dependent rate coefficient. $NO_3^-$ leaching depends on soil $NO_3^-$, available water holding capacity, and daily runoff. Reactive N is also lost from a grid cell by fire, assuming complete loss of N from burned vegetation, and $NH_3$ volatilization.





The N source by biological N fixation (BNF) is implied by maintaining prescribed soil C:N ratios associated with each of the plant functional types, reflecting their different litter chemistries and decomposer assemblages. N is added to the soil pools when litter is transferred to the soil pools to maintain their high N:C ratios. Sources of reactive N by weathering are implicitly

included in the BNF flux. Plant net primary productivity (NPP) and a prescribed constant N:C ratio of new production in different tissues sets the N demand that is satisfied by N uptake from $NH_4^+$ and $NO_3^-$ pools which in turn depend on net N mineralization fluxes from litter and soil pools and loss fluxes of reactive N (e.g. denitrification, leaching, volatilization) (Fig. 1). In case that available inorganic N (sum of $NH_4^+$ and $NO_3^-$) is insufficient to meet the demand, NPP is down-regulated, thereby inducing an effect of N limitation. BNF tends to re-establish a balance between the input and the loss of reactive N.

In LPX-Bern, the magnitude of the simulated global N source (BNF and weathering) is partly adjustable by two scaling parameters. These are the fractions of re-mineralized N that is returned to litter and soil by immobilization, respectively. Similarly, global $N_2O$ emission is adjustable by varying *RN2ODN*. The immobilization fractions are set to 0 % for litter and 26.39 % for soil mineralization in the standard setup. This results in a N source flux (Fig. 1) that is higher than the published

range (~60 to 340 TgN yr$^{-1}$) (Vitousek et al., 2013;Cleveland et al., 1999;Xu and Prentice, 2017;Houlton et al., 2018;Lenhart et al., 2015). The extent of immobilization of freshly added N is found to vary between 35 and 95 % from one soil to another with uptake by soil microorganisms, with a typical turnover of 1-2 months, dominating over abiotic processes (Bengtsson et al., 2003). In two sensitivity simulations, the immobilization fractions are set to 25 % for both soil and litter or to 26.4 % for soil and to 50 % for litter immobilization. Immobilization lowers the $NH_4^+$ and $NO_3^-$ pools in the model and loss fluxes of

reactive N. In turn, less BNF is required to maintain a balance of reactive N. Preindustrial BNF is 310 and 188 TgN yr$^{-1}$ in these two sensitivity simulations and within the published range. Importantly, the relative changes in modelled BNF and $N_2O$ emissions over the deglaciation are only about 5 % lower than in the standard setup (see Sect. 3.2 and 5). Thus, model-based conclusions are not sensitive to this scaling.

### 3.2 Setup for transient glacial-interglacial simulations

A previously described LPX-Bern model setup for glacial-interglacial simulations is applied (Spahni et al., 2013;Ruosch et al., 2016) and input data are shown in Fig. 2. The evolution of monthly temperature, precipitation, cloud cover, and number of wet days, annual atmospheric $CO_2$ (Joos and Spahni, 2008), orbital insolation changes (Berger, 1978) modulating plant available light, and topography changes through ice-sheet and sea-level changes imposed by ICE-5G (Peltier, 2004) are prescribed. The monthly climate data are obtained by combining monthly values from the observation-based, modern

climatology compiled by the Climate Research Unit (CRU) (Mitchell and Jones, 2005) with monthly anomalies for the past 21 kyr from a transient climate simulations over this period (TraCE-21kyr) (Liu et al., 2009;Otto-Bliesner et al., 2014) with the Community Climate System Model, version 3 (CCSM3) maintained by the National Centre for Atmospheric Research (NCAR). Here the LPX-Bern model was run with a spatial resolution of 3.75° longitude × 2.5° latitude and a daily time step





was applied in the photosynthesis, water and N-cycle modules. Simulations started from an equilibrated spin-up at 21 ka BP. Annual N deposition from the atmosphere, distributed across days according to precipitation within a year, is prescribed at preindustrial values (Lamarque et al., 2011).

The same model parameter values as determined by Lienert and Joos (2018a) are used. Regarding the N module, the parameters are the same as in a previous studies (Stocker et al., 2013;Schilt et al., 2014) addressing $N_2O$ emissions over the past and under future global warming. An exception is an adjustment in the upper limit of denitrification and in *RN2OND* in response to new observation-constrained estimates of marine $N_2O$ emissions and to a slightly revised estimate of the atmospheric $N_2O$ life time from 120 yr to 123 yr (Prather et al., 2015) as well as the inclusion of N immobilization as discussed above.

Changes in $N_2O$ emissions and other model outcomes are attributed to individual driving factors (temperature, precipitation, $CO_2$, orbital insolation, and land mask). One driver is kept at its preindustrial state in factorial simulations. BNF was kept at LGM values in an additional factorial run for each land use class and grid cell. The difference in results between the standard model setup (baseline) and a factorial run is attributed to the relevant driver. An interaction or synergy term, called "other

drivers" is quantified by the difference between the change in $N_2O$ emissions ($\Delta eN_2O$) simulated in the baseline run and the sum of the emission changes attributed to individual drivers: $\Delta eN_2O_{other-drivers}= \Delta eN_2O_{baseline} - \Delta eN_2O_{temperature} - \Delta eN_2O_{precipitation} - \Delta eN_2O_{CO2}$. The dominant driver is identified as having the largest contribution to $\Delta eN_2O$ in the baseline run with the same sign of change. Grid cells that submerge under sea water or emerge from waning ice sheets during the period considered and grid cells with insignificant changes ($|\Delta eN_2O| < 1$ mg N m$^{-2}$ yr$^{-1}$) are excluded from the spatially-resolved attribution.

The response time scales of LPX are investigated in a further "step-change" sensitivity simulation. Starting from the equilibrated spin-up at 21 ka BP, climatic conditions and atmospheric $CO_2$ are abruptly changed to conditions for 2500 ka BP. The run is continued for another 1500 years with climate and $CO_2$ forcing for the period from 2500 ka BP to 1000 ka BP. The land mask is kept constant at the maximum extent possible for both LGM and late Holocene conditions. A

corresponding reference simulation without step change was performed. The simulations, together with the above factorial runs, permit us to address how fast $N_2O$ emissions in the LPX model are able to respond to a sudden warming event, similar to the onset of the B/A and the end of the Younger Dryas.

**4 Reconstructed terrestrial $N_2O$ emissions and implications**

We start the presentation of results by summarizing the main feature of the terrestrial $N_2O$ emission record (Fig. 3, green line)
presented in part I of this study (Fischer et al., 2019). In part I, the global $N_2O$ emissions from land and from the ocean are jointly reconstructed by deconvolving novel ice core data of $N_2O$ and of its isotopic signature, $\delta15N(N_2O)$, using an established method and relying on differences in the isotopic signature of land versus marine $N_2O$ emissions. Terrestrial emissions



increased between LGM and PI (1500 CE) by about 1.7 TgN yr$^{-1}$. Terrestrial emissions remained approximately invariant during the Heinrich Stadial I Northern Hemisphere (NH) cold phase (HS1; 17.4 to 14.6 ka BP, (Rasmussen et al., 2014)) until the start of the Bølling/Allerød NH warm period (B/A; 14.6 to 12.8 ka BP). Then, land emissions increased at the start of the B/A, declined again during the Younger Dryas NH cold period (YD, 12.8 to 11.7 ka BP) and peaked at the start of the Preboreal

period (PB), followed by a modest increase during the Holocene.

Remarkably, the overall deglacial increase in terrestrial N$_2$O emissions was mainly realized in two large steps at the onset of the B/A and at the end of the YD, two major northern Hemisphere warming events. The detailed analysis of the ice core N$_2$O concentration and isotope data (see Fischer et al. (2019)) reveals that global terrestrial N$_2$O emissions started to rise at the

beginning of the warming events. Each step-like increase in terrestrial N$_2$O emissions was realized within maximum two centuries, and possibly faster, given the temporal resolution of the ice archive. The enclosure process of atmospheric air into firn and ice acts like a low pass filter, smoothing any fast variations in atmospheric N$_2$O, its isotopic signature, and, correspondingly, in inferred emissions. Fischer et al. conclude that global terrestrial N$_2$O emissions reacted within maximum 200 years to the large scale climate reorganizations associated with the two major deglacial northern Hemisphere warming

events.

Overall, the ice core data show that land ecosystem N$_2$O emissions responded sensitively to climatic and environmental changes over the deglaciation. The rapid increase in terrestrial N$_2$O emissions at the onset of the B/A and at the end of the YD and the overall increase in emissions over the past 21,000 years are in line with our working hypothesis I of an increasingly

"open" N cycle whereby N input and loss fluxes increased under warming climate and increasing CO$_2$. Reactive N was available in sufficient amount to support nitrifying and denitrifying organisms and an increase in global terrestrial N$_2$O emissions during periods where environmental conditions became, on a global scale, more favorable for plant growth and C sequestration (Ciais et al., 2012;Bird et al., 1994;Joos et al., 2004). This suggests that reactive N was available to support plant growth and to fuel N loss processes.

**5 Transient simulations of terrestrial N$_2$O emissions and the C-N cycle over the past 21,000 years**

**5.1 Simulated changes in global terrestrial N$_2$O emissions and spatial patterns of change**

We next explore governing mechanisms of the deglacial terrestrial N$_2$O emissions and potential implications for the C-N cycles in the spatially resolved, mechanistic LPX-Bern model. LPX-Bern v1.4 simulates a general increase in global land N$_2$O emissions over the deglacial period (Fig. 3). The simulated evolution matches the reconstructed change in terrestrial N$_2$O

emissions from the ice core isotopic records relatively well, although modelled changes are smaller than reconstructed variations. The model represents the emission variations during the Bølling/Allerød (B/A) and Younger Dryas (YD) periods with peaks in emissions at the onset of the BA (14.6 ka BP) and the preboreal (11.7 ka BP) and a smaller emission peak around





13.5 ka BP and corresponding minima at 14 ka BP and during the YD (12.8 to 11.7 ka BP). Reconstruction and models both show small changes in global terrestrial $N_2O$ emissions over the last 11 ka, the Holocene period. Simulated terrestrial $N_2O$ emissions decreased somewhat between 9 and 8 ka BP, whereas reconstructed emissions slightly increased over the Holocene, leading to a discrepancy between simulated and reconstructed anomalies. Overall, the agreement between proxy reconstruction

and model results supports the plausibility of the LPX-Bern model as well as of the underlying TraCE-21kyr climate input data used to force LPX-Bern on these long timescales. On the other hand, the model fails to reproduce the dynamic evolution of terrestrial $N_2O$ emissions during Heinrich Stadial 1 (HS1). The reconstruction suggests constant emissions from the land biosphere during HS1, whereas the model simulates steadily increasing emissions over the HS1 interval, reflecting the gradual climate warming in the TraCE-21kyr climate input data during HS1 (see discussion below). The model also shows a much

slower and a smaller emission increase at the YD/PB transition than reconstructed.

The changes in global terrestrial $N_2O$ emissions are the result of spatially differentiated responses. LPX-Bern simulates high natural emissions of $N_2O$ in the tropics and low emissions at high latitudes (Fig. 4A). The spatial pattern and the magnitude of emissions are consistent with data-based estimates of natural $N_2O$ emissions from soils (Zhuang et al., 2012;Stehfest and

Bouwman, 2006;Potter et al., 1996). At 1500 AD, emissions in the tropics can be as high as 250 mgN m$^{-2}$ yr$^{-1}$ and the integrated flux between 20$^o$S and 20$^o$N amounts to 64 % of the global emissions, while emissions per unit area are low in high latitudes and northern and southern extra-tropics contribute only a share of 22 % and 14 % to the global terrestrial emissions. In contrast, emissions increased strongly in the northern extra-tropics over the glacial termination (Fig. 4B) with the integrated change in emissions being half as large for the tropics than for the northern extra-tropics. Large increases per unit area are simulated over

the termination in mid- and low-latitudes on the North and South American continents, in the southern boreal zone in Eurasia and in parts of eastern Asia, Indonesia and Africa. $N_2O$ emissions decreased in a few regions, namely in Africa around 15$^o$S and in northern Australia (Fig. 4B). Loss of tropical land due to rising sea level and the addition of land emerging from waning ice sheets (Peltier, 2004) are important drivers of modelled terrestrial $N_2O$ emissions between 14 and 8 ka BP. The net loss of terrestrial $N_2O$ emissions by land area changes was dominated by the submergence of the high-productivity *Sunda* and *Sahul*

*Shelf* regions. This loss offsets about one-third of the G-IG increase in global terrestrial $N_2O$ emissions (Fig. 3). Changes in the land extent caused by changes in sea level is an important factor for past global terrestrial $N_2O$ emissions.

The patterns of change in terrestrial $N_2O$ emissions (Fig. 4C), as well as spatial and seasonal patterns in precipitation and temperature, are complex for the Holocene period. Despite small changes in global terrestrial $N_2O$ emissions during the

Holocene, LPX-Bern simulated large regional shifts in source strength. This includes for example a decrease in $N_2O$ emissions from 11 to 0.5 ka BP in boreal Asia, in the sub-Sahara region in Africa and in parts of the conterminous United States of America (USA), including Alaska, and an increase in emissions in tropical Africa, parts of Australia and Latin America as well as in Canada or Scandinavia.



## 5.2 Biological Nitrogen Fixation and carbon-nitrogen coupling

Biological nitrogen fixation (BNF) is assumed to adjust dynamically to an increase in N demand and partly alleviating N limitation of plant growth in LPX-Bern. It is implicitly assumed that limitation by other nutrients does not affect the cycling of N and C through ecosystems on multi-decadal to century time scales and that nutrient input into ecosystems by deposition, and weathering (plus BNF for N) is large enough to support plant growth. These assumptions are controversial (Hungate et al., 2003;Luo et al., 2004;Körner, 2015;Wieder et al., 2015b;Vitousek et al., 2010;Fatichi et al., 2014). Our novel ice core isotope reconstruction of terrestrial $N_2O$ emissions allows us to critically evaluate this assumption in a quantitative way on the multi-decadal to centennial time scales as relevant for the anthropogenic perturbation. To this end, we implement an N cycle representation leading to strong long-term nutrient limitation in the model. This is achieved by fixing the rate of BNF to its glacial value in each grid cell and land use class and by keeping these rates of BNF constant in a sensitivity simulation over the past 21 kyr. The simulation with constant BNF completely fails to reproduce the reconstructed $N_2O$ emissions from the land biosphere. This N-limited simulation yields not an increase, as reconstructed, but a decrease in global terrestrial $N_2O$ emissions (Fig. 3, black line). The decrease in global land $N_2O$ emissions is due to the loss of land in response to sea level rise, while $N_2O$ emissions on remaining land changes little in this sensitivity simulation.

The comparison of results between the simulations with and without N limitation yields further insight into the N and C coupling (Fig. 5 and 6). The N-limited model yields smaller changes in the N cycle, a smaller increase in global net primary productivity, and a reduced increase in vegetation growth and terrestrial C stocks than the standard setup. At the LGM and in both model setups, two thirds of the simulated global input and loss of reactive N on land and two thirds of net primary productivity (NPP) occur within the tropics, while about 55 % of the global C inventory in surface soils (2 m) and vegetation is simulated to be stored in the tropics. The higher percentage storage in extratropical C compared to N and C fluxes is explained by a slower turnover time of organic C in the cooler extratropical compared to the warmer tropical regions. Simulated BNF increased by 16 % from 451 TgN yr$^{-1}$ at the LGM to 523 TgN yr$^{-1}$ at 0.5 ka BP in the standard setup. BNF increases on most land regions in the standard setup, but remained by design constant on non-flooded land in the N limited simulation (Fig. 6A,B). Over this period and on non-flooded land, BNF increased in the standard setup by 56 TgN yr$^{-1}$ in the tropics and by 56 and 17 TgN yr$^{-1}$ in the northern and southern extratropics, respectively. Corresponding changes are simulated for nitrification and denitrification in the standard setup, whereas changes in nitrification and denitrification remain small on non-flooded land in the N-limited run. In response to the increased N input, the availability of reactive N remained roughly constant or increased in most land areas in the standard run, despite increased storage of N in plant and soil organic matter (except in parts of mid-latitude Eurasia where also C stocks decreased or changed little). In contrast soil reactive N concentrations decreased in many regions in the N-limited simulation as more N became progressively locked into vegetation and soil C.



These differences in BNF and the N cycle between the N-limited and standard case have profound impacts, not only for the emission of $N_2O$, but also for NPP and C sequestration (Fig. 6). Global NPP increased by 21 % in the standard case compared to 11 % in the N-limited run from the LGM to PI. LPX-Bern predicts that most of the NPP increase is realized in the northern extratropics, while integrated NPP even decreased in the tropics in response to a shrinking land area in both model setups. As a result of the higher NPP on remaining land in the standard setup compared to the N-limited setup, C storage at 0.5 ka BP is about 700 PgC and 32 PgC higher than in the LGM in the extratropical and tropical regions, respectively.

In summary, no increases in global N loss and $N_2O$ emissions from soils over the G-IG transition (Fig. 3A) are simulated in the N-limited model setup which is in clear contradiction to the ice core derived terrestrial $N_2O$ emission record. Only if the model is allowed to satisfy the demand of N, and thereby implicitly of other elements to support the growth of N fixers, nitrifiers and denitrifiers, and plants, terrestrial $N_2O$ emissions increase as reconstructed. The combined ice core and model results suggest that the global terrestrial N cycle operated in an increasingly "open" mode as in our working hypothesis I, where excess N, not used for biomass production, is lost to the environment and replaced again, e.g. through BNF, deposition and weathering. In contrast, the emergence of an "inflexible" N cycle (hypothesis II), as expected under progressive N limitation, did not materialize over the past 21,000 years. We emphasize that the ice core $N_2O$ concentration and isotope data represent globally integrated emissions and do not permit interferences on regionally differentiated responses, including possible nutrient limitation in specific regions.

### 5.3 Attribution of simulated terrestrial $N_2O$ emissions to climatic and environmental drivers

The complex spatio-temporal changes in land $N_2O$ emissions are attributed to climatic and environmental drivers (Fig. 7 to 10). This attribution helps us to better understand the simulated changes and to elucidate why the model fails to simulate the reconstructed changes in terrestrial $N_2O$ emissions during the Heinrich Stadial 1 (17.4 to 14.6 ka BP). Globally, deglacial warming is the most important factor contributing 1 TgN yr$^{-1}$ to the overall emission increase, followed by increasing atmospheric $CO_2$ (~0.4 TgN yr$^{-1}$), and changes in precipitation (~0.25 TgN yr$^{-1}$, Fig. 7). Orbitally driven changes in photosynthetically active radiation and non-linear interactions among the different drivers slightly offset the emission increase. The change in land extent due to ice sheet melting and sea level rise leads to a reduction in terrestrial $N_2O$ emissions (−0.7 TgN yr$^{-1}$).

Turning to HS1, the simulated increase in $N_2O$ emissions during this period is only about a quarter smaller when atmospheric $CO_2$ is kept at the PI value than in the standard setup (Fig. 7). This means that the majority of the simulated emission increase is in response to changes in physical climate, i.e., changes in temperature and precipitation. In other words, the simulated rise in land $N_2O$ emissions, which occurred earlier in the simulation than the ice core reconstruction, is not primarily driven by $CO_2$ fertilization and an associated increase in N ecosystem flows, but by the prescribed climate change from TraCE-21kyr. We note, however, that the small initial rise during the first part of HS1 is attributed to changes in $CO_2$. It is not clear whether



the model's failure to represent the reconstructed emission changes during the HS1 period in the standard simulation is due to deficiency in the response of the LPX-Bern to early deglacial climate change, or to deficiencies in the climate input data, or a combination of the two.

Individual drivers exert regionally distinct influences and these may vary between different periods. Here, we attribute the spatial changes to changes in temperature, precipitation, $CO_2$ and their non-linear interactions for the glacial termination and the Holocene using factorial simulations (Fig. 8) and by examining the temperature and precipitation changes of the TraCE-21kyr input data (Fig. 9). Generally, attributed changes in emissions find their counterpart in underlying changes in individual drivers, but sometimes non-linear interactions and non-additivity of individual responses hamper the attribution to individual
drivers.

Changes in temperature over the termination caused emissions to rise substantially in Eurasia, in the conterminous USA as well as in Argentina and southern Brazil and in eastern Australia (Fig. 8A). On the other hand, regional cooling in parts of the Amazon region and in parts of eastern Asia caused emissions to fall in this period. The pattern of change in $N_2O$ emissions
attributed to changes in temperature (Fig. 8B) is very different for the Holocene compared to the termination. The summer (June, July, August) cooling also found in climate data (Wanner et al., 2008) and simulated over the Holocene period in boreal Eurasia and the western part of North America results in a decrease in $N_2O$ emissions over large parts of Eurasia and North America. The slight summer warming in eastern Canada and Scandinavia in the model has little impact on simulated emissions. An increase in terrestrial emissions is simulated in tropical Africa and Latin America, including the Amazon region, in response
to changing temperatures.

The attribution of emissions to changes in precipitation (Fig. 8C,D) reveals some well-expressed dipole patterns, partly indicative of spatial shifts in precipitation regions (Fig. 9C). Attributed emission changes reflect corresponding changes in precipitation with generally increasing emissions under increasing precipitation. As expected, increasing $CO_2$ exerts a
generally positive influence on simulated emissions, in particular in the tropical belt (Fig. 8E,F). Non-linear interactions between the three drivers can be significant regionally and either enhance or reduce simulated emission changes (Fig. 8G,H). The temperature-attributed decrease in emissions over the termination at around $15^o$ S to $20^o$ S in Africa (Fig. 8A) is not linked to a corresponding change in seasonal or annual temperatures. This particular attribution as well as the attribution of negative emission changes to increasing $CO_2$ in this region (Fig. 8E,F) most likely reflect non-linear interactions between precipitation,
temperature and $CO_2$ .

Figure 10 further illustrates which driver exerts the largest influence on simulated regional emission changes in a given grid cell. Over the termination, changing temperature is the dominant influence on the simulated emission changes in mid- and high-latitude Eurasia and North America, while changes in precipitation dominate emission changes south of the Sahara, in



India and large parts of Australia, where temperatures were already rather high. Increasing $CO_2$ exerts a dominant control in tropical Latin America, Africa and in Indonesia over the termination. Over the Holocene, temperature is the dominant driver in northern Eurasia, while changes in precipitation dominate the emission response in Africa. Changes in $CO_2$ generally play a secondary role.

**5.4 The time scales of response to a step change in climate and $CO_2$**

Next, we assess on which time scales $N_2O$ emissions adjust in LPX-Bern after a sudden change in environmental conditions. The question is whether LPX-Bern can simulate an equally fast response in $N_2O$ emissions as reconstructed from the ice core data for the abrupt warming events at the onset of the B/A and the end of the YD or whether the intrinsic response time scales of the model are too long to match reconstructed emissions. We construct an extreme bounding case for rapid warming events
by switching climate (and $CO_2$) instantaneously, step-like from LGM to late Holocene conditions in LPX-Bern (Fig. 11A).

LPX-Bern shows a fast response, followed by a relatively small century scale adjustment (Fig 11). About 80 % of the final response in global $N_2O$ emissions to the step change is realized within 40 years and about 90% within a century, while it takes about 700 years to reach a near equilibrium (Fig. 11B). Taking the atmospheric lifetime of $N_2O$ of more than 100 years into
account, such a fast multi-decadal increase in $N_2O$ emissions would cause a century scale increase in atmospheric $N_2O$ concentrations, similar to what is seen in the ice core record. The adjustments in global BNF, nitrification and denitrification fluxes evolve similar as for $N_2O$ emissions with the main increase in these fluxes again realized within about 40 years (not shown). In contrast, the model's global N–leaching flux decreases immediately (by about 10-15 % relative to LGM) after the step, followed by a century-scale increase to approach late Holocene conditions around 1000 years after the step. This transient
decrease and recovery in modelled N leakage is indicative of a corresponding evolution of soil $NO_3^-$ availability. Modelled global NPP (Figure 11C) increases immediately after the step and about three quarters of the NPP change are realized in the first year after the step. Then it takes again about 700 years to reach the new equilibrium.

The fast initial response is explained as follows. NPP responds immediately to the change in environmental conditions. The
associated enhanced plant N uptake depletes soil $NO_3^-$ leading to the initial decrease in N leakage, while warming accelerates soil decomposition and the release of $NH_4^+$. The newly assimilated C and N is allocated to leaves, sapwood and hardwood and roots, before released to litter and relatively fast overturning soils on time scales from years to decades. These annual-to-decadal vegetation and litter turnover time scales govern the initial response time of BNF, nitrification, denitrification and $N_2O$ emissions. The century-scale response is linked to the equilibration time scales of C and N in the slowly overturning soil pools
as well as to the poleward expansion of vegetation. Finally, there is a small remaining offset between the reference run and the near equilibrium of the step simulation which is linked to remaining peat in the step-simulation formed under LGM conditions, but absent in the reference simulation.



In conclusion, this sensitivity experiment demonstrates that $N_2O$ emissions in the LPX-Bern model are indeed able to adjust within decades to less than a century to abrupt warming events, given prescribed forcing is changing fast enough. We conclude that the exact timing of simulated $N_2O$ emissions at the onset of the B/A and the end of the YD depends sensitively on the climate forcing data prescribed to the LPX-Bern model.

## 6 Discussion

### 6.1 Reconstructed terrestrial $N_2O$ emissions and implications for the C-N cycle

Terrestrial $N_2O$ emissions show a 40 % increase from the Last Glacial Maximum to the late preindustrial period ((Fischer et al., 2019)). Most of the deglacial increase was realized in two large steps, linked to rapid, decadal-scale and widespread northern hemisphere warming and to shifts in the Intertropical Convergence Zone (ITCZ) and precipitation patterns. Terrestrial $N_2O$ emissions result primarily from nitrification and denitrification (Firestone and Davidson, 1989). A requirement for these processes to take place is the availability of ammonium and nitrate, as consumed by nitrifiers, denitrifiers and plants alike. Reactive nitrogen (N) lost from ecosystems must be replaced by BNF, weathering and deposition to avoid ecosystem depletion of reactive N in the long term. Variation in reactive N availability among diverse land classes are found to correspond to variations in $N_2O$ emissions (Davidson et al., 2000). According to the hole-in-the-pipe concept (Firestone and Davidson, 1989), $N_2O$ emissions are directly indicative of the flow of N entering and leaving ecosystems.

The ice core data show that reactive N must have been available in sufficient quantity to support N uptake by nitrifiers and denitrifiers and increasing terrestrial $N_2O$ production over the deglacial period. The increasing $N_2O$ emissions imply an increasing flow of reactive N through land ecosystems on the global scale. In other words, the ice core data support our working hypothesis I of an increasingly "open" or "leaky" (Niu et al., 2016) global terrestrial N cycle whereby sources of reactive N on land increased under warming climate and increasing $CO_2$ over the deglacial period and contributed to meet the N demand of plants, nitrifiers and denitrifiers under more favorable growth conditions. At the same time, the ice core reconstructions seem to falsify - on century time scales - working hypothesis II, postulating that reactive N for the production of $N_2O$ (and implicitly for plant growth) remained scarce, while marine emissions dominated past atmospheric $N_2O$ variations. The consistently high terrestrial $N_2O$ emissions reconstructed for the last 7,000 years, a period when atmospheric $CO_2$ rose by 20 ppm (Elsig et al., 2009), appear also to be in conflict with the idea that reactive N becomes increasingly limiting under increasing $CO_2$.

Wieder et al. (2015b) proposed to downscale the carbon (C) uptake as projected by CMIP5 models for the 21[st] century to account for a postulated N limitation. This proposal was based on the assumption that the global inflow of reactive N to ecosystems is limited and static (Hungate et al., 2003) and consequently N availability is not sufficient to support the C sink projected by the CMIP5 models. The ice-core data, however, do not support the assumption of a strongly N-limited future land



C sink (Hungate et al., 2003;Wieder et al., 2015b). Rather, they point to a dynamic global N cycle whereby, at a global scale, N losses from soil inorganic N pools are replaced by adjustments of biotic or abiotic sources. The ice core data and the inferred rapid increase in terrestrial emissions at the onset of the B/A and the end of the YD suggest that such adjustment processes take place on multi-decadal to century time scales .

The magnitude of possible adjustments in the N source is unclear. Wang and Houlton (2009) modelled an increase in N fixation under increasing $CO_2$ and temperature. Yet their modelled increase in BNF was too small to meet projected N demand; the authors therefore proposed to downscale future C uptake in the warming projections of C4MIP models. However, this conclusion is based on an assumed optimum temperature for BNF of around 25°C, an assumption that has been challenged by Liao et al. (2017) who found the abundance of N-fixing trees to increase monotonically with temperature by analyzing more
than 125,000 forest plots in the USA and Mexico. Taken together, the ice core and forest plot data do not support downscaling of C uptake in the CMIP4 and CMIP5 projections. More work is needed to improve projections of BNF under global warming, considering also the cost of BNF (Shi et al., 2018) and relying on data-based and model approaches (Fisher et al., 2012).

The reactive N lost from ecosystems must be replaced if ecosystem N pools are not to be depleted. This mass balance
consideration suggests that the total input flux of N from BNF (Cleveland et al., 1999;Vitousek et al., 2013;Zähle, 2013;Sullivan et al., 2014), rock weathering (Houlton et al., 2018), and deposition (Lamarque et al., 2011;Vet et al., 2014;Dentener et al., 2006) increased hand in hand with $N_2O$ emissions and related loss fluxes over the deglacial period. An alternative to a balanced N input and output flux would be that N losses are fueled by existing reservoirs of reactive N. Large nitrate deposits ("caliche") are currently found in the hyperarid Atacama desert (Ericksen, 1981;Tapia et al., 2018). But such
large deposits are very unusual, and require several hundred thousand of years to accumulate (Michalski et al., 2004) . Microorganisms require readily decomposable carbon substrate for the denitrification of nitrate, otherwise nitrate may accumulate under the absence of substrate availability (Weier et al., 1993). Carbon substrate availability for denitrifiers might have increased at the onset of the B/A and at the end of the YD when climate warmed, precipitation pattern shifted and organic matter remineralization accelerated in many regions. This in turn could have led to the conversion of nitrate that had potentially
accumulated previously. However, such a scenario appears unlikely, given that $N_2O$ emissions remained high during the early B/A and throughout the Holocene.

It has remained somewhat unclear whether warming will increase or reduce regional to global scale $N_2O$ emissions, as responses to warming treatment are found to be highly variable across a range of conditions and ecosystems (Dijkstra et al.,
2012). The ice-core record shows that past warming events, such as those at the onset of the B/A and the end of the YD, strongly promoted $N_2O$ emissions globally. This suggests, as noted earlier (Schilt et al., 2014) and as projected (Stocker et al., 2013), that terrestrial $N_2O$ emissions from natural land will likely increase further as climate warms, implying the existence of a positive climate feedback linked to the terrestrial N cycle (Xu-Ri et al., 2012).



Uncertainties in the interpretation of the terrestrial $N_2O$ emission record are linked to uncertainties in the ratio between $N_2O$ produced to N converted during nitrification and denitrification. This yield factor is known to vary with environmental conditions (Diem et al., 2017;Davidson et al., 2000;Firestone and Davidson, 1989). For nitrification, the ratio of $N_2O/NO_3^-$

produced increases with increasing acidity and decreasing oxygen (Firestone and Davidson, 1989). For denitrification, the production ratio of $N_2O/N_2$ yield increases with increasing $NO_3^-$ availability, increasing oxygen concentration, decreasing decomposable carbon, decreasing pH and decreasing temperature. Changes in precipitation may have altered the $N_2O$ yield over the deglaciation. Higher soil water content and associated anoxic conditions generally favor the conversion of $N_2O$ to $N_2$ and results in a low yield of $N_2O$ (Weier et al., 1993), though dependencies of yield on water filled pore space are sometimes

complex (Diem et al., 2017) and soil texture and drainage affect water filled pore space and yield (Bouwman et al., 2002). Yield is low under low $NO_3^-$ availability (Diem et al., 2017) and generally found to increase when $NO_3^-$ becomes more available (Weier et al., 1993). This relation, when considered in isolation, implies increasing $N_2O$ emissions to be indicative of increasing $NO_3^-$ availability. Thus, the known dependency of yield on $NO_3^-$ availability is not in conflict with the idea implicit in working hypothesis 1 that reactive N became more available with increasing $N_2O$ emissions. The ratio of $N_2O$ to $N_2$ emitted generally

decreases with increasing temperature (Firestone and Davidson, 1989). On the other hand, the warming over the glacial termination is also expected to accelerate organic matter turnover and thus C availability which would tend to lower this ratio. Further, experimental studies on grasslands yield both higher and lower ratio of $N_2$ to $N_2O$ under elevated $CO_2$ compared to ambient $CO_2$ (Zhong et al., 2018;Baggs et al., 2003). Given this complexity, we are not in the position to evaluate whether the $N_2O$ yield factor for the combined N loss processes increased or decreased over the deglaciation on the global scale. An

increase would imply that the relative change in the flow of N through ecosystems was lower than the relative change in $N_2O$ emissions. In contrast, a decrease in yield would indicate that the relative change in ecosystem N through flow was even larger than the relative increase in terrestrial $N_2O$ emissions from the Last Glacial Maximum to the preindustrial period. Changes in N supply, e.g. by BNF, would scale accordingly. In any case, changes in yield factor cannot plausibly reconcile our $N_2O$ emission reconstruction with working hypothesis II of a closed N cycle. As noted by Bouwman et al. (2002), $N_2O$ losses will

be low at low flow rates, regardless of the yield factor.

The ice-core data and emission estimates represent global values with century-scale resolution. They do not permit us to discriminate regional variations, nor interannual to decadal changes. We do not exclude the possibility that responses were regionally differentiated during the deglacial, and that our hypothesis II and progressive N limitation may have been realized

in some regions or during short periods, as found in comparably short field experiments (Luo et al., 2004). In particular, N-limitation is considered to be common in high-latitude regions. But this does not exclude a decadal-to-century scale increase in BNF under changing environmental conditions. Sufficient N and other nutrients were available to support the northward expansion of boreal forest during the deglaciation. Indeed, it has been hypothesized that an increase in BNF may become cost-effective under more severe N limitation in mid and high latitudes (Menge et al., 2017). Field data support a role for BNF in





supporting forest growth in particular for early successional forests. Experimental evidence comes from chronosequence studies in the Amazon region and in Panama (Batterman et al., 2013;Gehring et al., 2005), from forest clay and sand box studies in Brazil and in the USA (Davidson et al., 2018;Bormann et al., 2002) as well as from forest inventory data covering tropical-to-temperate climatic conditions (Liao et al., 2017).

In summary, the reconstruction of terrestrial $N_2O$ emissions from ice-core data support working hypothesis I. The data show a deglacial increase in $N_2O$ emissions. This increase occurred in two major steps, each realized within maximum 200 years. and thus on a time scale similar to that of the ongoing anthropogenic climate perturbation. The increase in $N_2O$ emissions implies an increased input of N by BNF and other sources in land ecosystem, an increased flow of N entering and leaving

ecosystems, and a global N cycle that adjusted dynamically to meet increased N demand under more favorable growth conditions. We emphasize that this conclusion does not depend on a specific land biosphere model, but emerges from the ice core terrestrial $N_2O$ emission record.

### 6.2 Simulating deglacial terrestrial $N_2O$ emission changes with LPX-Bern

We applied the LPX-Bern dynamic global vegetation model to investigate regional $N_2O$ emissions patterns, governing

mechanisms, and C-N coupling. The model was forced with climate anomalies from the TraCE-21kyr simulation that used the Community Earth System Model (Liu et al., 2009;Otto-Bliesner et al., 2014), ice core $CO_2$ data, and reconstructions of sea-level and ice-sheet extent. The model was applied in an earlier study to simulate climate-$N_2O$ feedbacks under global warming (Stocker et al., 2013) and results for global emissions for the period from 16 to 10 ka BP are presented in Schilt et al. (2014). Recently, model parameters have been updated using a set of modern observational constraints in a Bayesian approach (Lienert

and Joos, 2018b). The updated version also includes immobilization of N released during remineralization. Results for deglacial $N_2O$ emissions remain basically unchanged between the previous (Stocker et al., 2013;Schilt et al., 2014) and the updated model version. We also note that the model has not been tuned in any way towards matching the ice core reconstruction.

LPX-Bern represents the cycling of N and C in soils and plants in a simplified way. C and N is stored in PFT-specific plant and litter pools and a fast and a slow overturning soil pool in each grid cell. This chain of pools represents a spectrum of overturning time scales on each grid cell. LPX-Bern, being a spatially explicit model, reacts differently in different regions to changes in climatic and environmental drivers (Fig. 4,8). However, microbial and fungal biomass are unlike in microbial-explicit models (Schimel and Weintraub, 2003;Zhu et al., 2017;Allison and Gessner, 2012)) not explicitly modelled and

organic matter decomposition does not depend on microbial mass and physiology. Instead, C:N stoichiometry is fixed within a single pool. There is also no distinction between different classes of organic matter according to their accessibility to microbial action (Averill and Waring, 2018). Further, BNF is treated in a simplified way and the cost of nitrogen acquisition are not considered (Fisher et al., 2010). Only net $N_2O$ production during denitrification is considered, while gross production





and consumption of $N_2O$ during denitrification (Schmid et al., 2001b) are not explicitly modelled. Potential $N_2O$ release by lichens and bryophytes is also not modelled (Porada et al., 2017). Nevertheless, the results provide a first estimate of past regional $N_2O$ emission changes and their drivers. The model allows us to investigate alternative working hypotheses in a consistent model framework. It remains to be seen whether other models incorporating the cycling of N and C (e.g., (Zaehle

et al., 2014;Averill and Waring, 2018;Thornton et al., 2007;Fisher et al., 2010) and $N_2O$ emissions (Zhu et al., 2016;Saikawa et al., 2013;Tian et al., 2015;Xu et al., 2017;Huang and Gerber, 2015;Zaehle and Friend, 2010;Olin et al., 2015;Goll et al., 2017) are able to represent the reconstructed terrestrial $N_2O$ emissions over the past 21,000 years. The setup of our deglacial simulation could be used to compare and evaluate models, e.g., in the framework of the model intercomparison initiated by the Global Carbon Project (Tian et al., 2018).

LPX-Bern forced by TraCE-21kyr output represents the reconstructed emissions reasonably well, while differences between reconstructed and simulated global land $N_2O$ emission remain. The modelled deglacial increase in global $N_2O$ emissions is at the lower bound of the reconstructed range. This may be related to the model's C cycle and/or to the model formulation for denitrification. LPX-Bern is known for a relatively low sensitivity to increasing $CO_2$ and simulates a modest increase in NPP

and the terrestrial carbon sink over the industrial period (Lienert and Joos, 2018b). Further, denitrification processes are assumed to respond to the relevant substrate concentration following Michaelis–Menten kinetics (Xu-Ri and Prentice, 2008). This limits N conversion rates and $N_2O$ production at high substrate concentrations, whereas a recent synthesis of N fertilization experiments (Niu et al., 2016) points towards an exponential relationship between N load and $N_2O$ emissions. $N_2O$ emissions are simulated to increase during Heinrich Stadial 1 (17.4 to 14.6 kaBP) in contrast to the reconstruction that shows stable

emissions during this period. Most of this modelled increase is attributed to the prescribed changes in climate. It is unclear whether this early increase reflects deficiencies in the climate input data or in the LPX-Bern model. Results of a sensitivity simulation, where environmental conditions are changed step-like, demonstrate that LPX-Bern is able to represent multi-decadal scale change in global terrestrial $N_2O$ emissions as implied by the ice core reconstructions for the rapid warming events, given climate forcing is prescribed to remain sufficiently long at LGM conditions and to change fast during the warming

event. The sensitivity simulation reveals a decadal adjustment time scale for the majority of the $N_2O$ emission response, followed by a smaller century-scale adjustment. Deglacial changes in $N_2O$ emissions reflect a complex interplay of different driving factors with considerable synergistic or antagonistic interactions. Globally averaged, and in mid- and high latitudes, the changes are predominantly driven by changes in temperature. Regional changes in precipitation exert the largest control in semi-arid regions, while increasing $CO_2$ leads to the largest relative responses in some tropical regions, where changes in

precipitation and temperature were modest.

The model suggests that increases in $N_2O$ emissions per unit land area were largest in low and mid-latitude regions, but the integrated change in emissions is larger for the northern mid- and high latitudes than for the tropical belt. Flooding of shelf regions, mainly around the maritime continent, led to a considerable reduction of tropical $N_2O$ sources during the





deglaciation. About a third of the global increase in terrestrial $N_2O$ emissions over the past 21,000 was counteracted by land loss. We conclude that past sea-level variations causing the flooding and emergence of shelves were an important factor for the evolution of terrestrial $N_2O$ emissions over glacial-interglacial time scales.

We applied LPX-Bern to analyse the consequences of alternative C-N cycle hypotheses in a quantitative manner and in a self-consistent, spatially and temporally resolved setting. In the model's standard setup, the N cycle responds dynamically to changes in environmental conditions; the N source increases under increasing ecosystem N demand. The results of the standard deglacial run are in line with working hypothesis I of an increasingly open terrestrial N cycle. Modelled sources of reactive N changed little or increased in most regions under warming climate and increasing $CO_2$ over the deglacial period. This additional
N source contributed to meet the rising N demand of plants simulated under more favorable growth conditions and resulted in a higher flow of N through land ecosystems and increased $N_2O$ production from terrestrial ecosystems. The availability of inorganic N in soils increased in many mid and high-latitude regions and remained high in the tropics.

In an alternative deglacial simulation, the sources of reactive N were prescribed at the level simulated for the Last Glacial
Maximum. In this case, the results correspond to working hypothesis II but do not agree with the ice core record. The terrestrial C cycle is simulated to become progressively N limited over the deglacial period. The availability of inorganic N decreased in many regions. Consequently, simulated terrestrial $N_2O$ emissions decreased and the simulated increase in the terrestrial C inventory amounted to 169 PgC, much smaller than the 901 PgC simulated in the standard setup, which is similar to terrestrial carbon storage increases simulated by models that do not account for N limitation (Prentice et al.,
2011;Joos et al., 2004).

In the standard setup, the simulated soil C inventories and their deglacial changes are consistent with a recent measurement-based reconstruction for the northern extratropics by Lindgren et al. (2018). These authors reconstruct current C stocks in peat and mineral soils (above 3 m) of 1630 GtC on land that was permafrost during the LGM, while LPX-Bern v1.4
simulates 1420 GtC in peat and mineral soils (incl. litter; above 2 m) north of 30º N. The reconstructed deglacial increase is 420 GtC, compared to 770 GtC simulated by LPX-Bern in soil and litter north of 30º N. Current northern extratropical peat C stocks (and deglacial C changes) simulated by LPX-Bern v1.4 are with 380 GtC comparable to a recent estimate of ~430 GtC (Loisel et al., 2014). The simulated LGM-to-Holocene change in total organic carbon stocks on land of 901 GtC is also comparable to the estimate by Jeltsch-Thömmes et al. (2018). These authors constrain the deglacial increase in land carbon
inventory to 850 GtC (median estimate; 450 to 1250 GtC $\pm1\sigma$ range) by using reconstructed changes in atmospheric $\delta^{13}C$, marine $\delta^{13}C$, deep Pacific carbonate ion concentration, and atmospheric $CO_2$ as observational targets.





## 7 Conclusions

Part I (Fischer et al., 2019) and II of this study present three novel elements to gain insights into the functioning of the global carbon-nitrogen cycle: First, a record of the $N_2O$ isotopic history over the past 21,000 years, complementing the existing records over parts of the termination (Schilt et al., 2014); second, the first reconstructions of marine and terrestrial $N_2O$ emissions from the Last Glacial Maximum (LGM) to the preindustrial period as obtained by deconvolving the $N_2O$ and isotope ice core records using a robust, established method; third, the application of a dynamic global vegetation model to simulate deglacial terrestrial $N_2O$ emissions under nitrogen and non-nitrogen limited conditions and to attribute deglacial $N_2O$ emissions changes to different regions and governing processes. Our study provides insight into the multi-decadal-to-millennial dynamics of the terrestrial C-N cycling by showing that the ice core terrestrial $N_2O$ emission record is explained with a rapid adjustment of N cycling to the climate and $CO_2$-driven acceleration of the C cycle.

The records provide the opportunity to evaluate models that simulate biological nitrogen fixation and nitrogen cycling (e.g. (Meyerholt et al., 2016;Nevison et al., 2016;Thornton et al., 2007;Xu and Prentice, 2017;Wieder et al., 2015a) and feedbacks between climate change, land carbon, and terrestrial greenhouse gas emissions (Arneth et al., 2010;Stocker et al., 2013;Tian et al., 2018) for improved projections. Here, we evaluated LPX-Bern and found that the model is able to simulate the evolution in global terrestrial $N_2O$ emissions over the past 21,000 years in reasonable agreement with the ice core emission data. This adds confidence to projections of the climate-$N_2O$ feedbacks with this model (Stocker et al., 2013). Model results for regional changes and specific processes and forcings await confirmation by other studies.

We explored two alternative working hypotheses of an increasingly "open" versus an "inflexible" global terrestrial nitrogen cycle on multi-decadal to century time scales. Reconstructed changes in terrestrial $N_2O$ emissions are interpreted to reflect changes in the nitrogen flow through land ecosystems (Firestone and Davidson, 1989) and to be indicative of the availability of reactive N to support the growth of plants and other organisms on the global scale. The ice core data reveal a highly dynamic terrestrial nitrogen cycle where sources of reactive N adjusted on multi-decadal time scales to meet increasing N demand to support plant growth and to fuel ecosystem loss fluxes of reactive N. Remarkably, substantial changes in global terrestrial $N_2O$ emissions were realized within maximum 200 years, and possibly faster, in response to past northern Hemisphere warming events. Reconstructed $N_2O$ emissions changed by up to 1 TgN yr[-1] and within less than two centuries at the onset of the NH warming events around 14.6 and 11.7 ka BP. These time scales are relevant for 21[st] century climate projections and much longer than accessible in typical laboratory or field experiments. Taken at face value, these results suggest that the nitrogen cycle will also adjust on the global scale in the coming centuries towards meeting N demand to support additional carbon uptake by plants under increasing temperature and $CO_2$.





The ice-core data and the $N_2O$ emission reconstruction provide a new window to integrate across systems and relevant time scales to improve our understanding of the coupled C-N cycle in the Earth system. The results may help to put emerging results from observational field and laboratory studies into the context of decadal-to-century scale environmental and climate change.

**Competing Interests**

5  The authors declare that they have no conflict of interest.

**Acknowledgements**

We thank G. Battaglia for helpful comments on the manuscript and T. Stocker for discussion. Long-term financial support of this research by the Swiss NSF (#200020_172476 (FJ), #200020_172506 (HF)) is gratefully acknowledged. Part of the research leading to these results has received funding from the European Research Council under the European Union's
10  Seventh Framework Programme (FP/2007-2013) / ERC Advanced Grant Agreement n. 226172 (MATRICs) awarded to H.F. This work is a contribution to the AXA Chair Programme in Biosphere and Climate Impacts and the Grand Challenges in Ecosystems and the Environment initiative at Imperial College London (CP). The TraCE-21ka project was supported by the US NSF and the US Department of Energy (BOB&ZL).



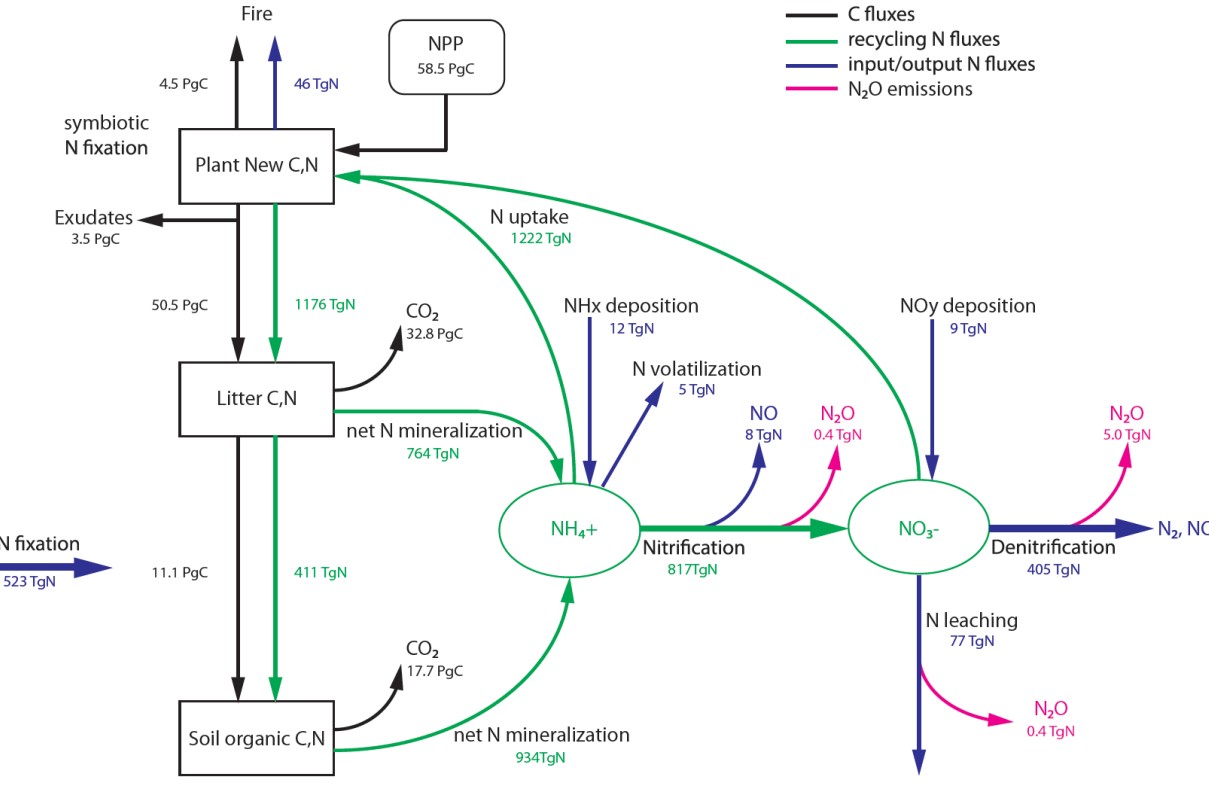

**Figure 1: Terrestrial C and N cycles. Simple schematic of global annual C fluxes (black), recycling (green) and input/output (blue) N fluxes, and N₂O emissions (magenta) from nitrification and denitrification processes. Fluxes are illustrated with a quantitative budget from the LPX-Bern(v1.4) model for pre-industrial conditions (1500 CE) Plant net primary productivity (NPP) sets the N demand that is satisfied by N uptake from ammonium (NH₄⁺) and nitrate (NO₃⁻). N enters the ecosystem mainly through biological**
10  **N fixation (BNF) by plant symbiotic and free-living microorganisms in the soil.**



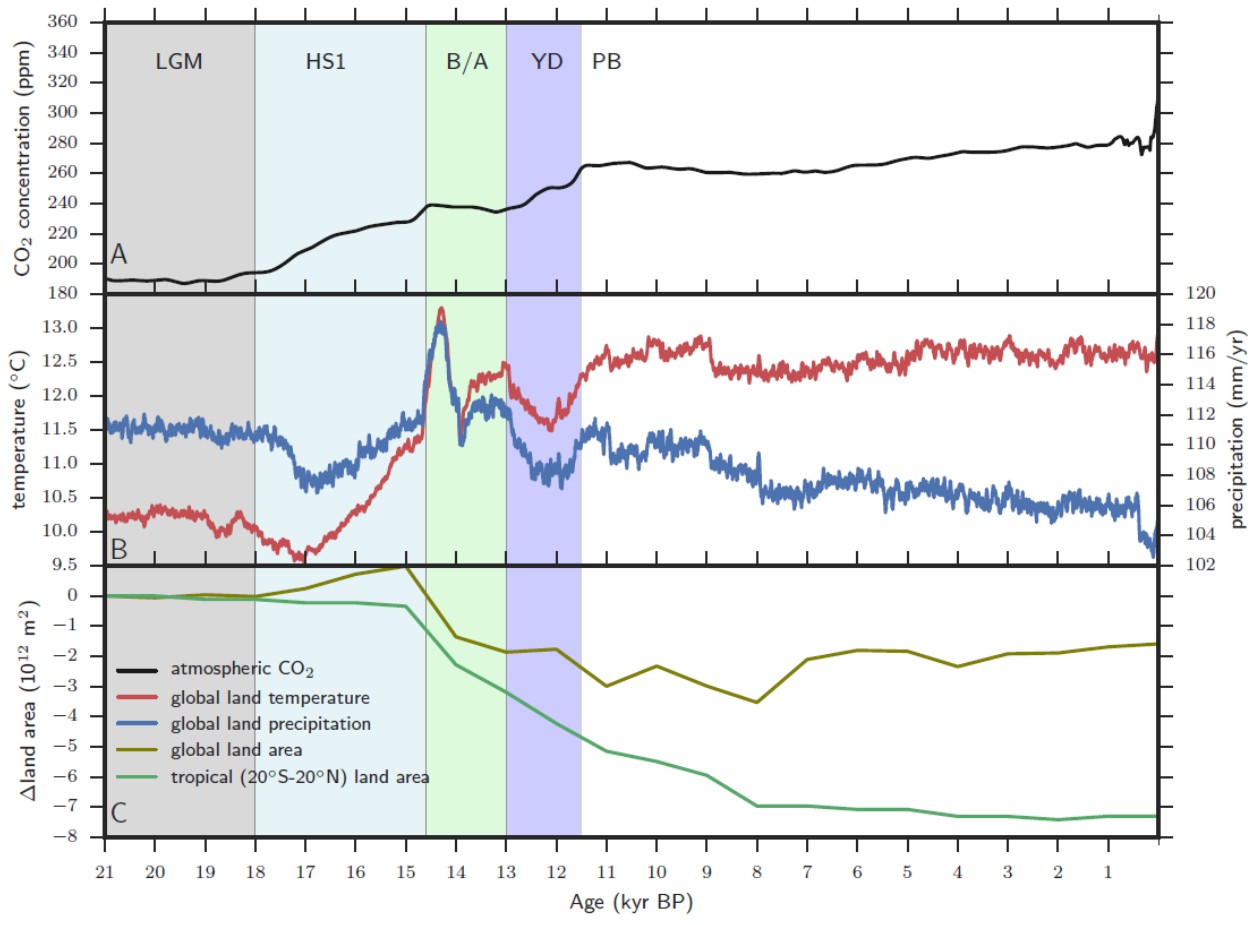

**Figure 2:** Aggregated LPX-Bern(v1.4) input data. (A) Evolution of atmospheric $CO_2$, (B) temperature and precipitation, and (C) of changes in tropical and global land as available for plant growth relative to 21 ka BP (bottom). Temperature and precipitation represent mean values over global land areas that were not flooded and not ice-covered at 21 ka PB and at 0.5 ka BP.





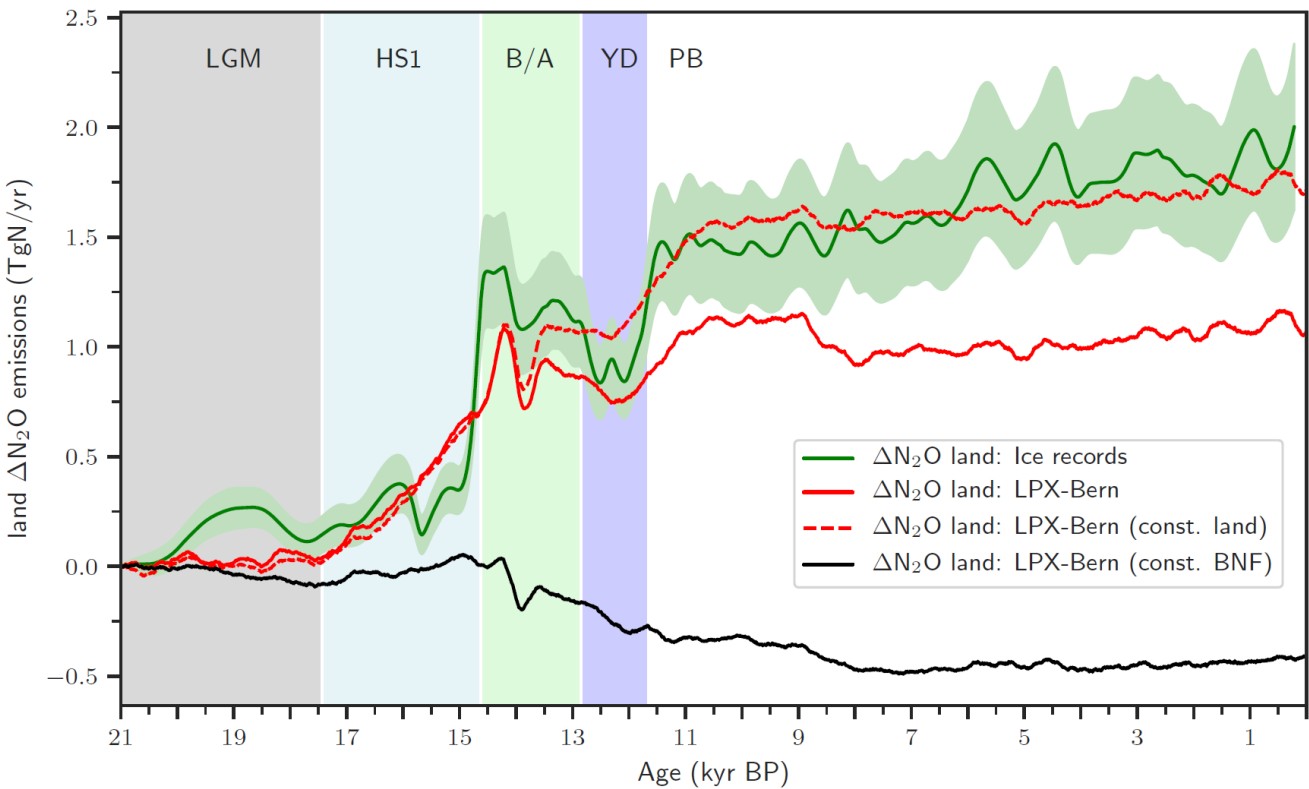

**Figure 3: Changes in global terrestrial emissions of N₂O. Reconstructed changes in global land N₂O emissions are shown by the solid green line together with an uncertainty estimate (±1 standard deviation; shaded area). Reconstructed changes are calculated with an atmospheric two-box model (Schilt et al., 2014) from ice records of N₂O concentration and δ¹⁵N(N₂O). Reconstructed changes are compared to LPX-Bern(v1.4) for the baseline simulation (red line), a constant land area simulation (dashed red line), and a simulation with biological nitrogen fixation kept constant at Last Glacial Mximum values (black line).**



**Figure 4: Preindustrial patterns (A) and changes in terrestrial N₂O emissions for the last glacial termination (B; 11 ka BP minus 21 ka BP) and the Holocene (C; 0.5 ka BP minus 11 ka BP) as simulated with the LPX-Bern(v1.4). Numbers indicate integrated values for three latitudinal bands (90° S–20° S; 20°S–20° N; 20°N–90° N).**





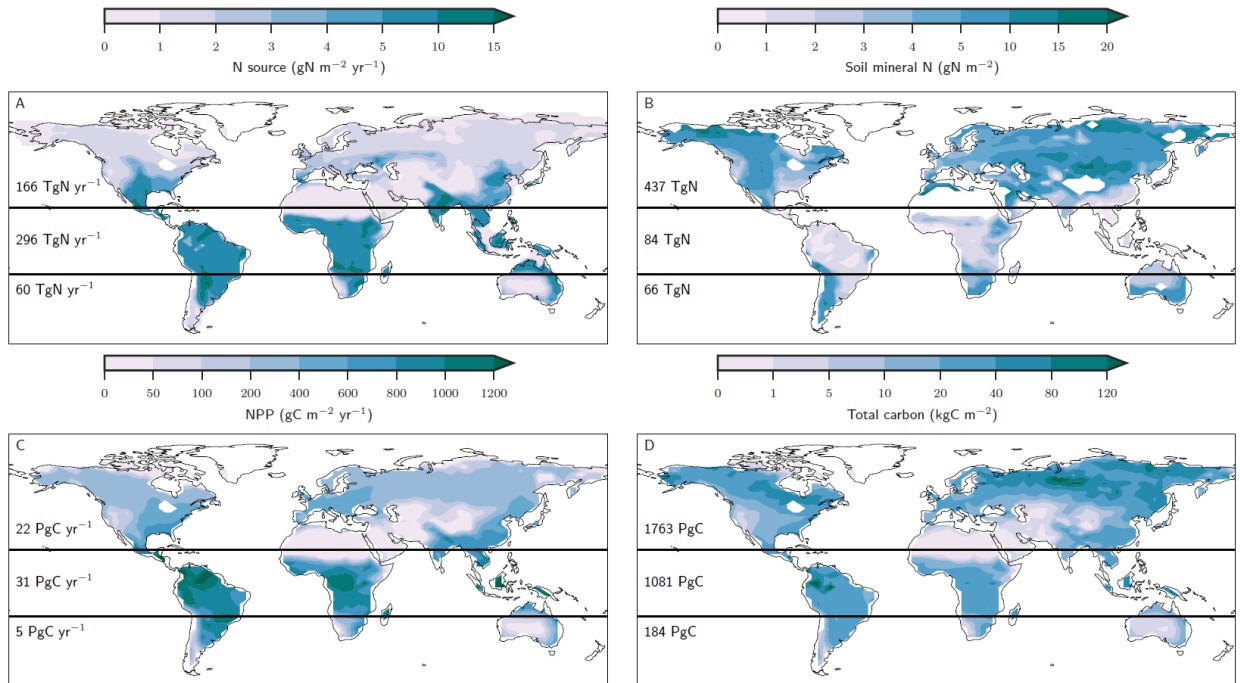

**Figure 5: Preindustrial patterns of the input flux of new nitrogen (N) into ecosystems (A), the bioavailable concentration of ammonium and nitrate in soils (B) net primary productivity (C) and total carbon in the land biosphere (D) as simulated by LPX-Bern(v1.4). The N source in (A) represents ecosystem input by biological nitrogen fixation and weathering and does not include the prescribed N deposition. Deserts with high mineral nitrogen concentrations are masked for clarity in (B). Numbers indicate integrated values for three latitudinal bands (90° S–20° S; 20° S–20° N; 20° N–90° N).**





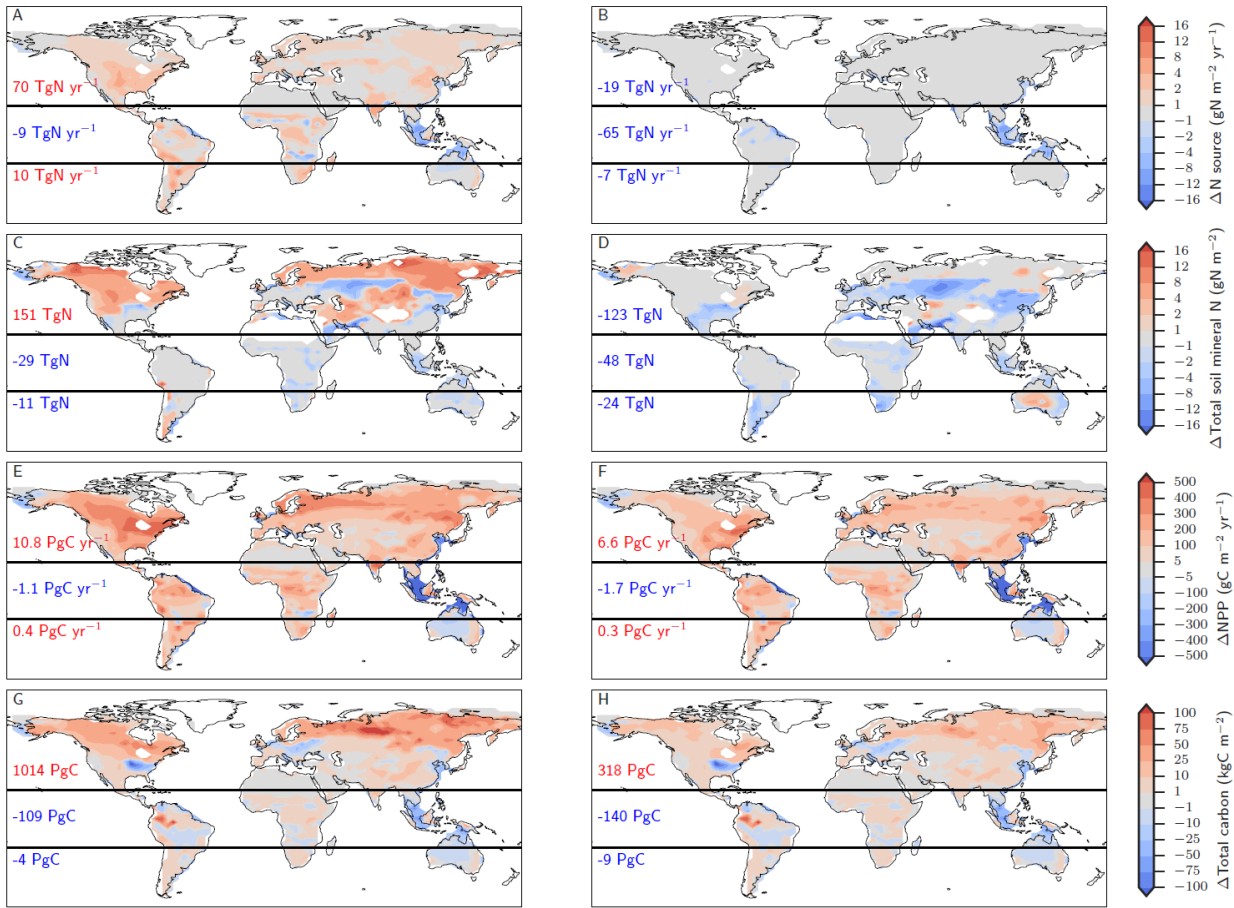

**Figure 6: Biological nitrogen fixation adjusts to plant demand (left) versus time-invariant nitrogen fixation (right). The figures shows simulated deglacial changes (0.5 ka BP minus 21 ka BP) for the standard setup of LPX-Bern(v1.4) (left column) and a setup where the nitrogen source flux is kept constant at values simulated for the Last Glacial Maximum (right column) for the same variables shown in Fig. 5. Note decreasing BNF is shown in panel B for land areas lost due to deglacial sea level rise and in some tropical cells where mineral soils with relatively high BNF are replaced by peat with relatively low BNF.**





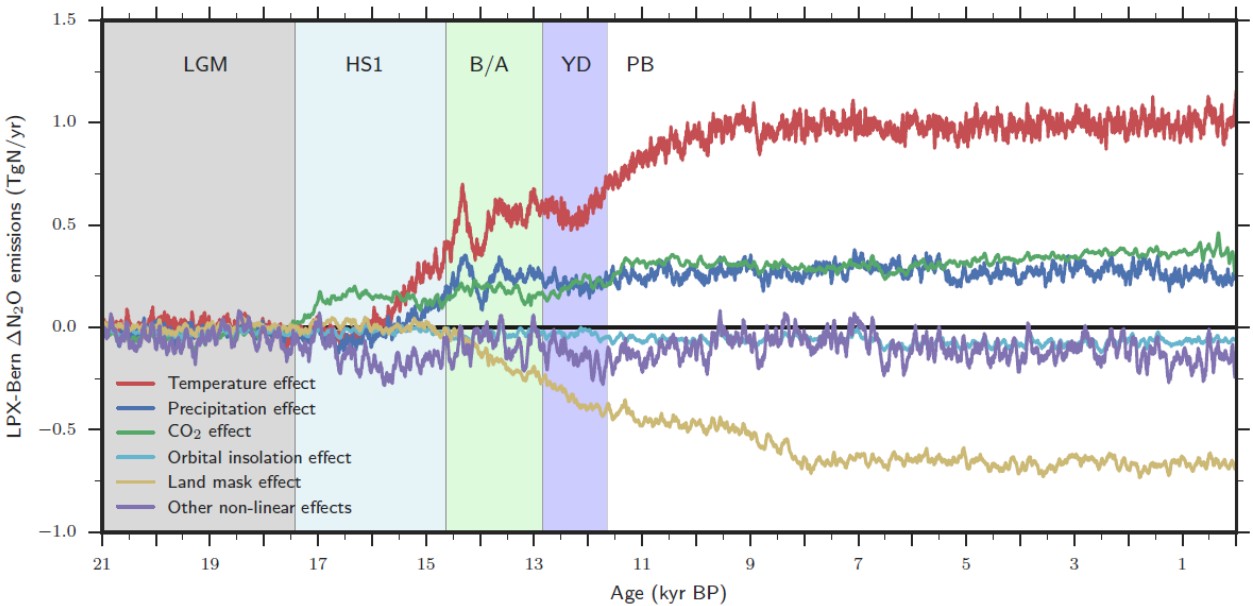

**Figure 7: Attribution of simulated global terrestrial N₂O emission changes to individual drivers. Results are from LPX-Bern(v1.4) for the standard and factorial simulations, where an individual driver (temperature (red), precipitation (blue), atmospheric CO₂ (green), orbital parameters affecting photosynthetic active radiation (blue), land mask accounting for sea level changes) is kept at preindustrial level. The violet curve (Other non-linear effect) is the difference between simulated emissions in the standard setup and the sum of all attributed emissions; it reflects the effect of non-linear interactions between different drivers.**



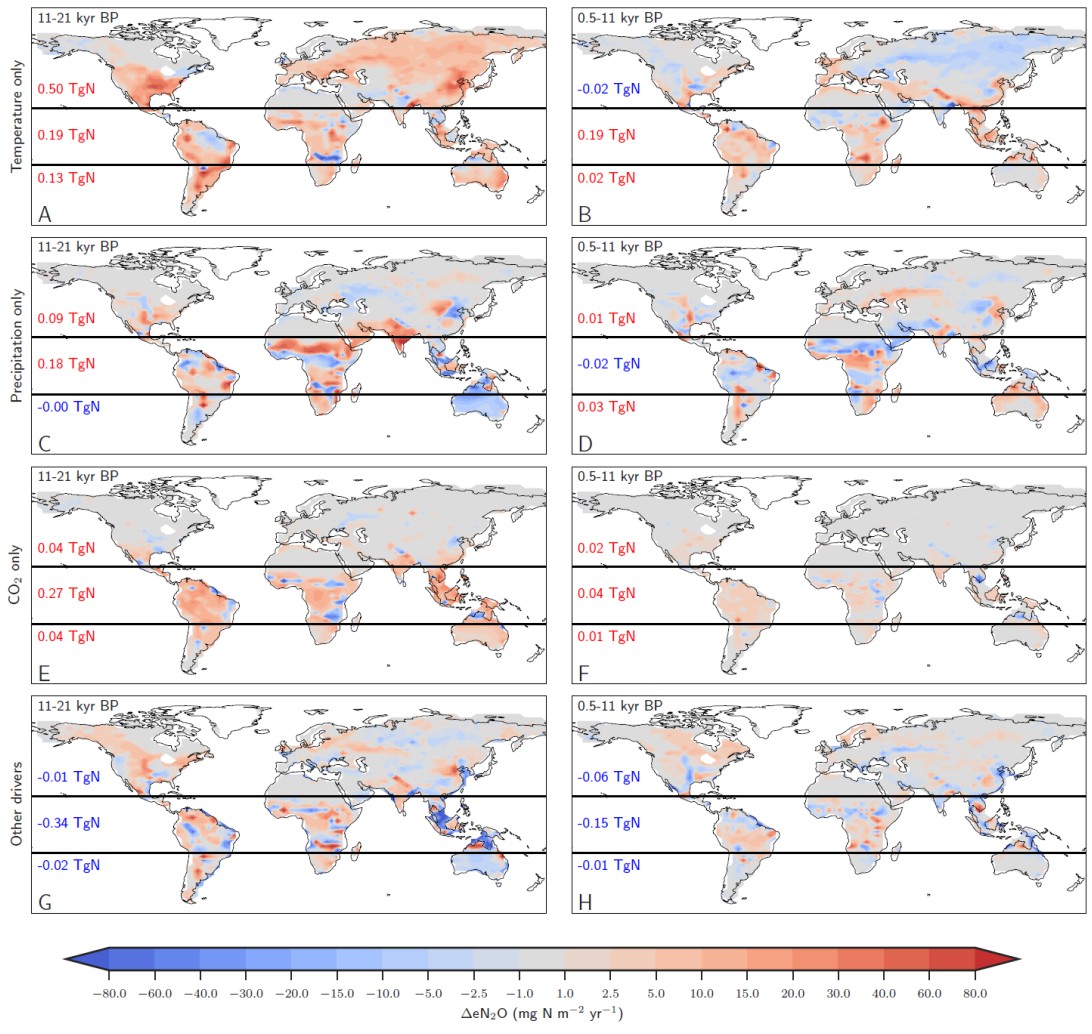

**Figure 8: Attribution of simulated global terrestrial N₂O emission changes to individual drivers for the last glacial termination (11 ka BP minus 21 ka BP; left) and the Holocene (0.5 ka BP minus 11 ka BP; right). Changes are attributed to changes in precipitation (A,B), temperature (C, D), atmospheric CO₂ (E,F) and to non-linear interactions among the drivers plus the small effect due to changes in incident solar radiation (G,H).**





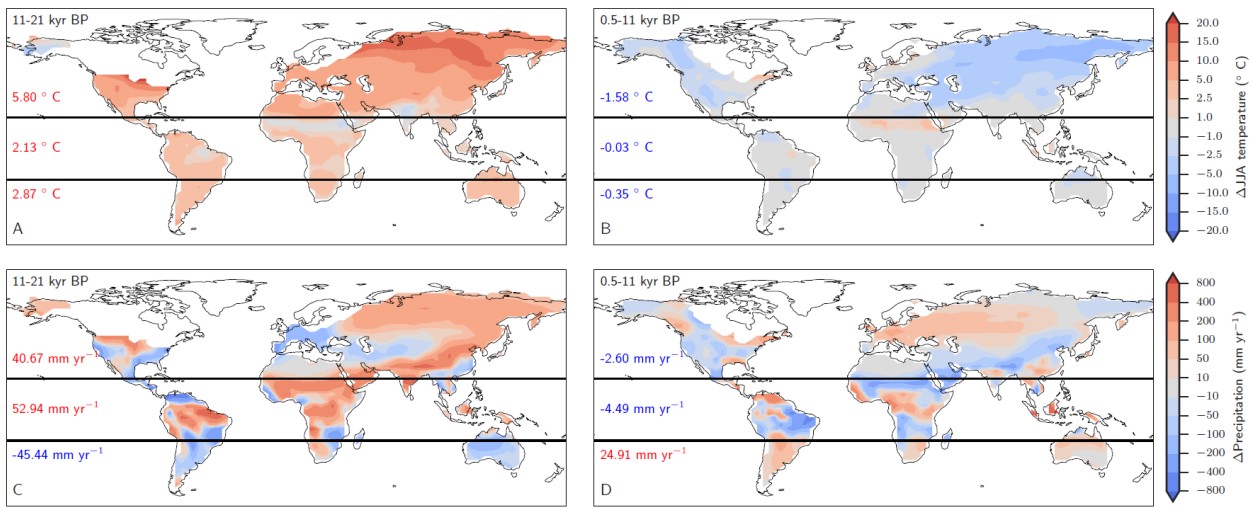

**Figure 9: Changes in northern hemisphere summer temperature (June, July, August, panels A and B) and annual precipitation (C,D) as simulated by CCSM3 and prescribed to LPX-Bern(v1.4) for the glacial termination (A,C) and the Holocene (B,D)**




**Figure 10: The dominant drivers for changes in terrestrial N₂O emissions over (A) the glacial termination and (B) the Holocene are temperature (red), precipitation (blue), CO₂ (green), and other non-linear effects in LPX-Bern(v1.4) to these driver combinations (violet). The color shading indicates the fraction of the total emission change attributed to the dominant driver typically explaining between 40 and 90 %. Areas with minimal changes are masked.**





**Figure 11: Response to a step-like change in (A) environmental conditions for globally averaged terrestrial (B) N$_2$O emissions and (C) NPP. Atmospheric CO$_2$ (dotted line in (A)) and climate is prescribed to suddenly change from Last Glacial Maximum conditions to late Holocene conditions in a sensitivity simulation with LPX-Bern(v1.4) ("step run", blue lines). Annual values for the step run are shown in dark blue and smoothed (31-yr average) values in light blue for the step run and in dark green for a corresponding reference run. The dashed horizontal lines represent averages from the reference run over the model period 0 to 1200 years. Surface air temperature in (A) represent the average over the model's land area. Note the change in the time axis scaling at year 100.**





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
