# Peer review of "$N_2O$ changes from the Last Glacial Maximum to the preindustrial - part II: terrestrial $N_2O$ emissions and carbon-nitrogen cycle interactions"

_Biogeosciences, 2019_

## Referee Comment (RC1) · Anonymous Referee #1 · 1 May 2019

This paper takes new ice-core data on nitrous oxide emissions over the deglaciation and compares it to modelled results so as to better understand biological nitrogen fixation (BNF) in the same period. There follows a section on the contribution of different climate drivers on nitrous oxide emissions in separate attribution simulations. The abstract is rather misleading, focusing on the work of the 'part I' paper which describes the ice-core data, and BNF rather than attribution modelling. However, there are two problems with the main BNF part of this paper: the model's representation of BNF, and the hypotheses posed. These two issues combined make the model runs virtually

meaningless and the conclusions baseless.

The model BNF

The model BNF (page 9, 13 and Fig.1) is 523 TgN/yr globally in the pre-industrial simulation. The authors acknowledge this is "higher than the published range" but are selective about what range they are referring to and what the implication of this is. Their BNF is an order of magnitude more than low budget-based estimates (Vitousek et al., 2013) (44/58 TgN/yr) and almost 200TgN/yr larger than the upper model estimate (Xu-Ri and Prentice, 2017) (340 TgN/yr) they reference. The authors cite Cleveland et al., (1999) (195 TgN/yr), but interestingly fail to cite the more recent paper by the same author, revising the estimate down to 127.5 TgN/yr (Cleveland et al., 2013). The authors' reference to Lenhart et al., (2015) (page 9, line 16) as part of the "published range" of global BNF estimates is baffling, as the paper discusses nitrous oxide and methane emissions, not BNF. Moreover, the authors fail to mention that of all estimates of BNF in the last half century, only one (Xu-Ri and Prentice, 2017) is over 300 TgN/yr and most are around 100 – 150 TgN/yr.

These global BNF estimates of around 100 TgN/yr are because field experiments show that nitrogen fixation is relatively unusual in the terrestrial biosphere. Whilst individual nitrogen fixing plants or organisms have the potential to fix large amounts of nitrogen, they are best suited to 'pioneer' environments with low soil nitrogen, are usually found at very low densities in mature ecosystems, and may be facultative (rather than obligate) fixers. Taking one example, tropical forest is generally thought to be the highest BNF region due to its high NPP and low nitrogen limitation. Recent work by Sullivan et al., (2014) found that tropical forest in Costa Rica has BNF of ∼0.5 gN/m2/yr. From the map of pre-industrial BNF in LPJ-Bern (Figure 5 A) it seems BNF in Costa Rica is modelled at ∼10 gN/m2/yr. i.e. the model overestimates BNF in the tropics by a factor of ∼20. Though no present day BNF value is given in the paper, from the information available the present-day modelled value for Costa Rica is likely higher. Even compared to the upper bound for tropical forest of 6 gN/m2/yr from the meta-analysis done

by Cleveland et al., (1999), the values in LPJ-Bern are high. Cleveland et al., (1999) said their upper limit was "extremely unlikely" and the global BNF from those upper values was 290 TgN/yr (compared to LPJ-Bern's 523 TgN/yr).

The BNF in LPJ-Bern is entirely disjointed from reality. This puts significant doubt on the ability of this model to produce meaningful results about or based on BNF. The authors infer on page 9 that the high BNF is irrelevant to their results. They describe two sensitivity experiments with lower global BNF and the nitrous oxide emissions are broadly similar. However, the two sensitivity experiment global BNF values (310 and 188 TgN/yr) are still unrealistically high. Therefore, these sensitivity experiments reveal that the problem with BNF in the model may not be a parameter based issue but could be something more fundamental.

The hypotheses

The hypotheses set out on page 7 present a false dilemma. They are based on the premise that the nitrous oxide emissions are attributable to a nitrogen system that is either 'open' or 'closed'. The idea of either an open or closed system stems from the assumption that the nitrogen cycle is in equilibrium during the deglaciation (stated by the authors on page 6, line 5). However, there's no evidence either way on this question. Even if the assumption of equilibrium is accepted, an open system is not the only mechanism of increasing nitrous oxide emissions, thus the hypotheses are a false dilemma. The changes in nitrous oxide could also be caused by changes to the internal dynamics of the system (e.g. soil nitrogen turnover, or flexible C:N ratios, or the authors' assumption of homogeneous nitrous oxide yield fractions over space and time (page 8, line 26)).

The answer to the false dilemma presented is pre-decided by the definitions the authors give on page 6. Under the false dilemma, an 'open' (high-input-high-output) system is presented as the only mechanism that can produce high terrestrial nitrous oxide emissions. We know we have increasing (nitrous oxide) output, so under the assumption

of either 'open' (high-input-high-output) or 'closed' (low-input-low-output), there is only one possible answer. This pre-determined result is exacerbated by the model. The high BNF in the model means the N cycle in the model must be 'leaky' and 'open' otherwise there would be no N limitation at all (contrary to the evidence, see LeBauer and Treseder, (2008)). Some models take a more 'closed' approach (often resulting in low BNF). But it stands to reason that closing the input of N in a model reliant on high N input will cause the model to produce results inconsistent with reality. These simulations might inform somewhat about the model but can't say anything about real world BNF.

The combination of an invalid hypothesis and an inappropriate model is results that mean nothing and conclusions that mislead. An unwary reader could easily take it at face value that BNF increased by 72 TgN/yr during the deglaciation and an 'open' terrestrial nitrogen system was the only, or most likely, way the observed nitrous oxide changes could have occurred. But there is no reliable evidence for either of these assertions.

The second part of the paper on attribution appears more sensible (issues with the BNF representation in the model not withstanding). This part of the paper might be appropriate for resubmission separately, without the BNF model results. With the two presented together it's difficult to assess the attribution section fairly.

References

Cleveland, C.C., Houlton, B.Z., Smith, W.K., Marklein, A.R., Reed, S.C., Parton, W., Grosso, S.J.D., Running, S.W., 2013. Patterns of new versus recycled primary production in the terrestrial biosphere. Proc. Natl. Acad. Sci. 110, 12733–12737. https://doi.org/10.1073/pnas.1302768110 Cleveland, C.C., Townsend, A.R., Schimel, D.S., Fisher, H., Howarth, R.W., Hedin, L.O., Perakis, S.S., Latty, E.F., Fischer, J.C.V., Elseroad, A., Wasson, M.F., 1999. Global patterns of terrestrial biological nitrogen (N2) fixation in natural ecosystems. Glob. Biogeochem. Cycles 13, 623–

645. https://doi.org/10.1029/1999GB900014 LeBauer, D.S., Treseder, K.K., 2008. Nitrogen limitation of net primary productivity in terrestrial ecosystems is globally distributed. Ecology 89, 371–379. https://doi.org/10.1890/06-2057.1 Sullivan, B.W., Smith, W.K., Townsend, A.R., Nasto, M.K., Reed, S.C., Chazdon, R.L., Cleveland, C.C., 2014. Spatially robust estimates of biological nitrogen (N) fixation imply substantial human alteration of the tropical N cycle. Proc. Natl. Acad. Sci. 111, 8101–8106. https://doi.org/10.1073/pnas.1320646111 Vitousek, P.M., Menge, D.N.L., Reed, S.C., Cleveland, C.C., 2013. Biological nitrogen fixation: rates, patterns and ecological controls in terrestrial ecosystems. Philos. Trans. R. Soc. B Biol. Sci. 368, 20130119. https://doi.org/10.1098/rstb.2013.0119 Xu-Ri, Prentice, I.C., 2017. Modelling the demand for new nitrogen fixation by terrestrial ecosystems. Biogeosciences 14, 2003–2017. https://doi.org/10.5194/bg-14-2003-2017

---

## Referee Comment (RC2) · Anonymous Referee #2 · 12 May 2019

The manuscript of Joos et al combines ice-core derived terrestrial N2O time-series with process-based N2O simulations to derive constraints on terrestrial nitrogen dynamics. The manuscript is well written and easy to read despite a little long and redundant. I suggest to be cautious about points mentioned below before the final acceptance. One of my concerns is that the conclusion related to biological controls on N acquisition is already pre-included in the assumptions/definitions based on which the model is built. BNF in the manuscript refers to any N inputs, other than atmospheric N deposition, that satisfy ecosystem N demand. With a constant annual N deposition rate, any changes in

ecosystem N content and losses are attributable to BNF. Here BNF incorporates both biological and non-biological sources, which might come from weathering, be undiscovered N sources existed in pre-industrial time, or errors from assuming constant N deposition rate. A second concern is about the adjustments of global inflow of reactive N on multi-decadal to century time scales derived from N2O dynamics. As the authors mentioned, there are multiple-steps and many factors come into play in global N cycle. Different N2O production pathways may evolve through time, alternations of soil organic matter decomposition, stoichiometry and other N loss pathways are likely to shift N2O emissions. These adjustments are likely to occur without significantly alteration of real biological nitrogen fixations. For example, nitrifying and denitrifying microbes may have different temperature sensitivities vs. BNF. Vegetation and microbial evolution are largely unconsidered in this study. There is no strong evidence that N input flux would adjust as quickly as that of N2O emissions. A third concern is related to insights to be learned from this study. Is it necessary to conduct spatially explicit model simulation to test constant vs. dynamic BNF? The magnitude of N2O emissions can be easily tuned through RN2ODN, whereas the "openness" or 'tightness' of N cycle is, to a large extent, conceptual and not new in literature. The spatial pattern is also within our general understanding of global ecosystems as the model is built upon the contemporary (not paleo-) biogeochemistry and driven by historical climate. I feel the climate sensitivity of N2O emissions are valuable information that worth exploring for models like LPX-Bern. Specific points: 1. P6L25-30. Does it worth discussion on losses of plant-available vs. plant-unavailable (e.g., through fire and leaching of DON) N and how losses of plant-unavailable N alter system dynamics? 2. P10L5. It is unclear when the upper limit of denitrification is used

---

## Author Comment (AC1) · 26 Jun 2019

**Reply to review comments - bg-2019-118**

$N_2O$ *changes from the Last Glacial Maximum to the preindustrial - part II: terrestrial* $N_2O$ *emissions constrain carbon-nitrogen interactions*

We thank the editor and both reviewers for assessing this manuscript and for their time and effort. All changes, we aim to implement in the revised manuscript, are shown in a draft. Please find this draft, with planned text changes highlighted and with updated figures, at the end of this document. We understand that the editorial decision is with respect to the originally submitted manuscript and we are looking forward for further editorial advice guiding the revision.

The main changes implemented are as follows:

- Reviewer I was concerned that a high value of BNF may invalidate our results. We adjusted the model parameter values, repeated all simulations, and updated figures and text. The simulated global source of reactive N, including BNF, is now in line with empirical estimates and other model results. In spite of this adjustment, our main findings on deglacial change in $N_2O$ emissions remain unchanged.

- The formulation of our working hypotheses apparently lead to misunderstanding. We deleted section 2.3 "Working Hypotheses" and corresponding text in the manuscript. We modified the framing of our results and now discuss uncertainties and different mechanisms to potentially explain the reconstructed $N_2O$ emissions in section 2 and 4, complementing the existing discussion in section 6. We rewrote the abstract and state in the abstract: "The increase [in $N_2O$ emissions] may be explained by an increase in the flux of reactive N entering and leaving ecosystems or by an increase in $N_2O$ yield per unit N converted. ... Our results appear consistent with suggestions of (a) biological controls on ecosystem N acquisition, and (b) flexibility in the coupling of the C and N cycles during periods of rapid environmental change. Alternative mechanisms to explain the reconstructed $N_2O$ emissions include changes in $N_2O$ yield per N converted and changes in N sources from weathering."

- We restructured and revised the text, in particular section 5.2 and 6 to avoid redundancy, to adjust to the revised framing of our results and in response of specific review comments.

The original review comments are given below in black, our reply in blue, and quotes from the revised manuscript in gray. Please note that page, line and section numbers all refer to the originally submitted manuscript.

**Reviewer #1**

*Interactive comment on Biogeosciences Discuss., https://doi.org/10.5194/bg-2019-118, 2019.

This paper takes new ice-core data on nitrous oxide emissions over the deglaciation and compares it to modelled results so as to better understand biological nitrogen fixation (BNF) in the same period. There follows a section on the contribution of different climate drivers on nitrous oxide emissions in separate attribution simulations.

The abstract is rather misleading, focusing on the work of the 'part I' paper which describes the ice-core data, and BNF rather than attribution modelling.

The reconstruction of the marine and terrestrial $N_2O$ emissions is described in part I. Part I also provides a first interpretation of marine emission changes (see page 4 l7 to 9). However, the analysis and interpretation of the reconstructed terrestrial $N_2O$ emission changes is the topic of part II. This is stated in the abstract. The corresponding text is found in sections 4 and 6.1 in the MS. We shorten the text summarizing specific findings of part I on l18 to l24 in abstract. The new text reads: Here we analyse the large increase in terrestrial $N_2O$ emissions over the past 21,000 years as reconstructed from ice-core isotopic data and presented in part I of this study. Remarkably, the increase occurred in two steps, each realized over decades and within maximum two centuries, at the onsets of the major deglacial northern hemisphere warming events. The data suggest a highly dynamic and responsive global N cycle. The increase may be explained by an increase in the flux of reactive N entering and leaving ecosystems or by an increase in $N_2O$ yield per unit N converted.

We modified the last sentence of the introduction (p4, l10) to read: Here in part II we focus on the interpretation of the terrestrial $N_2O$ emission record and discuss terrestrial $N_2O$ emissions and C-N responses in transient deglacial simulations with a dynamical vegetation/terrestrial biogeochemistry model.

However, there are two problems with the main BNF part of this paper: the model's representation of BNF, and the hypotheses posed. These two issues combined make the model runs virtually meaningless and the conclusions baseless.

We understand that the reviewer is concerned about the high value of BNF. We also understand that our formulations of working hypotheses did not help to convey our results as intended. These two issues are now addressed. We present results for modelled BNF (N source) close to the estimate by Cleveland et al., 2013. We revised the presentation of our results and removed section 2.3 "Working hypotheses" and related text from the MS. Nevertheless, some key statements by the reviewer regarding model formulations, model sensitivity, and assumptions are not correct (see below). We address these issues in the discussion below and have revised the text in the MS to clarify these issues.

The model BNF

The model BNF (page 9, 13 and Fig.1) is 523 TgN/yr globally in the pre-industrial simulation. The authors acknowledge this is "higher than the published range" but
5 are selective about what range they are referring to and what the implication of this is. Their BNF is an order of magnitude more than low budget-based estimates (Vitousek et al., 2013) (44/58 TgN/yr) and almost 200TgN/yr larger than the upper model estimate (Xu-Ri and Prentice, 2017) (340 TgN/yr) they reference. The authors cite Cleveland et al., (1999) (195 TgN/yr), but interestingly fail to cite the more recent paper by the
10 same author, revising the estimate down to 127.5 TgN/yr (Cleveland et al., 2013). The authors' reference to Lenhart et al., (2015) (page 9, line 16) as part of the "published range" of global BNF estimates is baffling, as the paper discusses nitrous oxide and methane emissions, not BNF. Moreover, the authors fail to mention that of all estimates of BNF in the last half century, only one (Xu-Ri and Prentice, 2017) is over 300 TgN/yr
15 and most are around 100 – 150 TgN/yr.

We added the following text on line p9, l10: Estimates of the global BNF for non-agricultural ecosystems are uncertain. They range from 40 to 470 TgN yr$^{-1}$, with most published estimates around 100 to 150 TgN yr$^{-1}$. Cleveland et al. (1999) used 100 plot-scale estimates of BNF to estimate global BNF on natural ecosystems to 195
20 TgN yr$^{-1}$ (range: 100 to 290 Tg N yr$^{-1}$). Vitousek et al. (2013) suggest a plausible range for preindustrial BNF of 40 to 100 TgN yr$^{-1}$ by computing BNF as the difference from all other global sources and sink fluxes of N. Cleveland et al. (2013) estimate symbiotic BNF to 105 TgN yr$^{-1}$, based on cost-benefit modelling for N fixation. The same authors estimate asymbiotic N fixation to 22 TgN yr$^{-1}$ by upscaling measurements
25 reported in Cleveland et al. (1999). In contrast, Elbert et al. (2012) estimate asymbiotic N fixation by cryptogamic covers alone to 49 TgN yr$^{-1}$ (27-99 TgN yr$^{-1}$) by integrating experimental data from 200 studies. In addition, rock weathering is estimated to add 10 to 20 TgN yr$^{-1}$ to land ecosystems (Houlton et al, 2018). Xu-Ri and Prentice (2017) estimate global BNF to 340 (230–470) TgN yr$^{-1}$ based on stoichiometric constraints
30 within the LPX model; this estimate includes contributions from rock weathering and other inputs that are not explicitly prescribed or simulated by LPX. Meyerholt et al. (2016) implemented six different BNF formulations in their model and predict modern BNF ranging from 108 to 148 TgN yr$^{-1}$.

Lenhart et al. 2015 discuss $N_2O$ emissions from cryptogamic cover and summarize
35 the results for N fixation by cryptogamic covers from Elbert et al., 2012. We deleted the reference to Lenhart et al., 2015 and added the reference to Cleveland et al., 2013 on p9, l16. We now explicitly cite the results of Elbert et al., 2012 on p9, l10 (see above).

These global BNF estimates of around 100 TgN/yr are because field experiments
40 show that nitrogen fixation is relatively unusual in the terrestrial biosphere. Whilst individual nitrogen fixing plants or organisms have the potential to fix large amounts of nitrogen, they are best suited to 'pioneer' environments with low soil nitrogen, are

usually found at very low densities in mature ecosystems, and may be facultative (rather than obligate) fixers. Taking one example, tropical forest is generally thought to be the highest BNF region due to its high NPP and low nitrogen limitation. Recent work by Sullivan et al., (2014) found that tropical forest in Costa Rica has BNF of 0.5 gN/m2/yr.

5 From the map of pre-industrial BNF in LPJ-Bern (Figure 5 A) it seems BNF in Costa Rica is modelled at 10 gN/m2/yr. i.e. the model overestimates BNF in the tropics by a factor of 20. Though no present day BNF value is given in the paper, from the information available the present-day modelled value for Costa Rica is likely higher. Even compared to the upper bound for tropical forest of 6 gN/m2/yr from the meta-

10 analysis done by Cleveland et al., (1999), the values in LPJ-Bern are high. Cleveland et al., (1999) said their upper limit was "extremely unlikely" and the global BNF from those upper values was 290 TgN/yr (compared to LPJ-Bern's 523 TgN/yr).

The parameters for N immobilization were inadvertently set to low values by Lienert and Joos (2018). This in turn leads to a high N source in the model. We decided to

15 use this published version for the originally submitted MS, complemented by the results from sensitivity simulations with higher immobilization and lower BNF. This approach was selected for reasons of traceability and transparency. We re-ran all simulations with an updated parameter set to yield a global preindustrial and modern N source of 128 and 140 TgN yr$^{-1}$, respectively. This flux of reactive N to ecosystems implicitly includes

20 contributions from symbiotic and asymbiotic N fixation, rock weathering, and possibly other, yet to be discovered, pathways. In view of this, we adjusted the text and now refer to 'ecosystem N inputs' or 'N source' in general rather than BNF.

This is also relevant for comparing this flux with empirical estimates. Accounting for the N weathering flux of around 15 TgN yr$^{-1}$ (Houlton et al. 2018), the simulated

25 flux of reactive N inputs presented here implies a modern BNF close to the reviewers preferred value of 127.5 TgN yr$^{-1}$. As noted already in the original MS (p9, l20-23), relative changes in modelled N source and N$_2$O emissions and related model-based conclusions are not sensitive to the exact parameter values. In addition, the yield factor for nitrification is assumed to vary with temperature in the same way as the yield

30 for denitrification in this revised setup. This has a modest impact on modelled N$_2$O as only 10 % of the simulated N$_2$O flux stems from nitrification.

All figures in the manuscript were updated with results for this new parameter set-ting and included in the draft provided at the end of this document. Specifically, Fig, 5A displays the N source (BNF plus weathering and potential other abiotic sources) sim-

35 ulated by LPX-Bern. Typical values in Central America are around 1 gN m$^{-2}$ yr$^{-1}$ for the new setup, and comparable to field measurements (Sullivan et al., 2014; Wurzburg and Hedin, 2016). For example, Wurzburg and Hedin, 2016, report BNF of N fixers from 0 (for about half of the trees) to up to 12 gN m$^{-2}$ yr$^{-1}$ in their supplementary material. New text is added in section 5.2 to describe results displayed in Figure 5:

40 The N source (Fig. 5A) is typically smaller than 0.5 gN m$^{-2}$ yr$^{-1}$ in northern mid and high latitude and around 1 gN m$^{-2}$ yr$^{-1}$ in tropical rainforest and in Central Amer-ica, comparable to observational estimates (Sullivan et al., 2014; Wurzburger and Hedin,

2016a, b). An N source of 2 to 6 gN $m^{-2}$ $yr^{-1}$ is simulated in many semi-arid regions, including southern Africa, the sub-Sahara region, India, northern Australia, and in the southern parts of North America. Soil mineral N is typically below 0.5 TgN $m^{-2}$ in the tropics ....

The BNF in LPJ-Bern is entirely disjointed from reality. This puts significant doubt on the ability of this model to produce meaningful results about or based on BNF. The authors infer on page 9 that the high BNF is irrelevant to their results. They describe two sensitivity experiments with lower global BNF and the nitrous oxide emissions are broadly similar. However, the two sensitivity experiment global BNF values (310 and 188 TgN/yr) are still unrealistically high. Therefore, these sensitivity experiments reveal that the problem with BNF in the model may not be a parameter based issue but could be something more fundamental.

As outlined above, the assumption by the reviewer regarding the sensitivity of the model to the absolute value of the N source and BNF is not correct. The text starting on p9, l11 is modified to read: The magnitude of the modelled N source co-governs the rate of nitrification, denitrification, and leaching and, together with the yield factors, $N_2O$ emissions. As noted above, the model parameters governing the $N_2O$ yield are calibrated towards a preindustrial $N_2O$ emission of 5.9 TgN $yr^{-1}$. A high N source in the model, therefore, implies low $N_2O$ yield factors and vice versa. In LPX-Bern, the magnitude of the simulated global N source is partly adjustable by two scaling parameters. These are the fractions of re-mineralized N that is returned to litter and soil by immobilization, respectively. The immobilization fractions are set to 0 % for litter and 60 % for soil mineralization in the standard setup. This results in a N source flux of 128 TgN $yr^{-1}$ (Fig. 1) with an implied weathering source of 15 TgN $yr^{-1}$ and BNF of 113 TgN $yr^{-1}$, well within the published range. The extent of immobilization of freshly added N is found to vary between 35 and 95 % from one soil to another with uptake by soil microorganisms (with a typical turnover of 1-2 months) dominating over abiotic processes (Bengtsson et al., 2003). The global mean yield factor for denitrification, expressed as $N_2O$ per $N_2$ produced, is 5.6 % and thus higher than the range of estimates (0.2–4.7 %) summarized by Xu-Ri and Prentice (2008). Similarly, the yield factor for nitrification (0.26 %) is higher than observation-based estimates (0.01-0.2 %). The mismatches in these estimates for yield may suggest that current best estimates for the N source, $N_2O$ yield, and preindustrial $N_2O$ emissions are not fully consistent. We carried out a sensitivity simulations to explore uncertainties: the immobilization fractions is set to 0 % for litter and to 26.39 % for soil mineralization. Lowering immobilization increases the $NH4^+$ and $NO3^-$ pools in the model and loss fluxes of reactive N. In turn, a higher N source is required to maintain a balance of reactive N, while implied $N_2O$ yield factors become lower. The preindustrial N source is 523 TgN $yr^{-1}$ in this sensitivity simulation and thus about a factor of four higher than in the standard setup. Correspondingly, yield factors are about a factor four lower in this sensitivity run than in the standard. The absolute magnitude of the N source has some implication for N stress and the increase

in NPP over the deglaciation is smaller in simulations with a low compared to a high N source. Importantly, the difference in relative changes in modelled global N source is small (14 % versus 10 % in the standard) and the deglacial increase in $N_2O$ emissions is only 0.1 TgN $yr^{-1}$ higher in the sensitivity than in the standard run (see Sect. 3.2 and 5), despite the large difference in the implied N source. Thus, related model-based conclusions are not sensitive to the parameter settings for immobilization and yield.

The hypotheses

The hypotheses set out on page 7 present a false dilemma. They are based on the premise that the nitrous oxide emissions are attributable to a nitrogen system that is either 'open' or 'closed'. The idea of either an open or closed system stems from the assumption that the nitrogen cycle is in equilibrium during the deglaciation (stated by the authors on page 6, line 5). However, there's no evidence either way on this question. Even if the assumption of equilibrium is accepted, an open system is not the only mechanism of increasing nitrous oxide emissions, thus the hypotheses are a false dilemma. The changes in nitrous oxide could also be caused by changes to the internal dynamics of the system (e.g. soil nitrogen turnover, or flexible C:N ratios, or the authors' assumption of homogeneous nitrous oxide yield fractions over space and time (page 8, line 26)).

We do not assume equilibrium over the deglaciation – neither when discussing the ice core record, nor the model results. The model applied here accounts for the transient dynamics of all C and N pools and fluxes and the main underlying principle is mass conservation. The highly transient nature of the ice core $N_2O$ emission record and of the N cycle in LPX-Bern is evident in Figs. 3, 4, 6, 7, 8, 10, and 11. To avoid the misunderstanding by the reviewer, we modified the sentence on p6, l5 to read: Following mass balance, losses of reactive N from ecosystems and changes in ecosystem and abiotic N stocks have to be compensated by N inputs, mainly by BNF. In addition, we modified the last sentence of the introduction (p4, l10) to point the reader early on to the transient nature of our simulations. .. and discuss terrestrial $N_2O$ emissions and C-N responses in transient deglacial simulations with a dynamical vegetation/terrestrial biogeochemistry model.

Furthermore, we do not assume a constant $N_2O$ yield factor in space and time as asserted by the reviewer. The yield factor for denitrification, responsible for more than 85 % of the $N_2O$ flux in our model, is significantly varying in space and time as described on p8 l20 to l23 in the original MS.

It was not our intention to present a dilemma, but rather to guide the reader and place our analysis in the wider scientific context, connecting to fundamental question about ecosystem functioning and the nature of N limitation. We deleted section 2.3 and all text related to working hypotheses to avoid potential misinterpretations. Alternative explanations for the deglacial $N_2O$ increase are discussed in the original MS, e.g. related to abiotic N source (p6, l23), related to pre-existing stocks of reactive N on p18 l14 to

26), and related to changes in the yield factor per unit N converted on p19, l1 to 25. However, these additional explanations may be viewed as coming too late in the MS. We therefore adjusted the abstract as described in the second bullet at the beginning of this reply and added text in section 2 and 4. We added the following text at the end of section 2.1 on production mechanisms: The amount of $N_2O$ produced per unit of N converted varies with environmental conditions and production pathway. For nitrification and denitrification this yield (or emission) factor depends on substrate availability linked to soil organic matter decomposition and C:N stoichiometry, on oxygen level influenced by soil moisture status, on acidity and temperature (Diem et al., 2017;Davidson et al., 2000;Firestone and Davidson, 1989;Smith, 1997;Phillips et al., 2015;Saggar et al., 2013). In addition, different $N_2O$ production pathways and N loss processes and their relative importance may evolve through time and influence the $N_2O$ yield on local to global scales.

We added the following text at the end of section 2.2 on C-N-$N_2O$ coupling: In summary of section 2.1 and 2.2, changes in terrestrial $N_2O$ emissions may be linked (i) to changes in the magnitude of reactive N entering and leaving ecosystems, and (ii) to changes in the $N_2O$ yield per unit reactive N converted in land ecosystems. We added the following text towards the end of section 4 on reconstructed $N_2O$ emission: The rapid increase in terrestrial $N_2O$ emissions at the onset of the B/A and at the end of the YD and the overall increase in emissions over the past 21,000 years either point (i) to an increase in $N_2O$ yield per unit N converted for emissions to the atmosphere, averaged globally and across all $N_2O$ production pathways, or/and (ii) to an increase in the global flux of converted N.

The answer to the false dilemma presented is pre-decided by the definitions the authors give on page 6. Under the false dilemma, an 'open' (high-input-high-output) system is presented as the only mechanism that can produce high terrestrial nitrous oxide emissions. We know we have increasing (nitrous oxide) output, so under the assumption of either 'open' (high-input-high-output) or 'closed' (low-input-low-output), there is only one possible answer. This pre-determined result is exacerbated by the model. The high BNF in the model means the N cycle in the model must be 'leaky' and 'open' otherwise there would be no N limitation at all (contrary to the evidence, see LeBauer and Treseder, (2008)). Some models take a more 'closed' approach (often resulting in low BNF). But it stands to reason that closing the input of N in a model reliant on high N input will cause the model to produce results inconsistent with reality. These simulations might inform somewhat about the model but can't say anything about real world BNF.

The combination of an invalid hypothesis and an inappropriate model is results that mean nothing and conclusions that mislead. An unwary reader could easily take it at face value that BNF increased by 72 TgN/yr during the deglaciation and an 'open' terrestrial nitrogen system was the only, or most likely, way the observed nitrous oxide

changes could have occurred. But there is no reliable evidence for either of these assertions.

We have to stress that these statements by the reviewer are not correct. As we show (please compare the figures in the original and the revised manuscript) results for changes in $N_2O$ emissions are similar for low (128 TgN yr$^{-1}$) and high (523 TgN yr$^{-1}$) input of reactive N as already discussed above. In addition, we would like to emphasize the distinction between state and change – We do not make statements whether current ecosystems are N limited or not, but are focussing on the *temporal change* as stated on p7,l5 ("To guide further discussion .. for the temporal evolution ..) and on p7,l16-22 ("The question posed in this study is not to what extent different ecosystems are, or have been N, limited. Rather, we ask the question whether BNF and the N cycle adjusted dynamically ...") in the original manuscript. Apparently, we failed to make this point sufficiently clear to the reviewer. As noted above, section "2.3 Working hypotheses" on p6 and 7 is deleted to avoid confusion and misinterpretations. As also noted above, different mechanisms of change are now discussed in the abstract and in section 2 and 4. In addition, we restructured the discussion section and discuss potential changes in $N_2O$ yield per unit N converted early in section 6.1. We state in the conclusion (sect.7): Our model results provide insight into the multi-decadal-to-millennial dynamics of the terrestrial C-N cycling by showing that the ice core terrestrial $N_2O$ emission record could potentially be explained with a rapid adjustment of N cycling to the climate and $CO_2$-driven acceleration of the C cycle, but they do not exclude the possibility that alternative explanations linked to changes in $N_2O$ yield could be important.

In summary, we aim to present our results in a balanced way and acknowledge the complexity of the C-N cycle. We thank the reviewer for pointing out shortcomings in the presentation of our results, however, we have to reject some of her/his statements and conclusions regarding our results as detailed above. We feel that our additional model runs and the revisions of the manuscript sufficiently address the more fundamental concerns of reviewer #1

The second part of the paper on attribution appears more sensible (issues with the BNF representation in the model not withstanding). This part of the paper might be appropriate for resubmission separately, without the BNF model results. With the two presented together it's difficult to assess the attribution section fairly.

The manuscript of Joos et al combines ice-core derived terrestrial N2O timeseries with process-based N2O simulations to derive constraints on terrestrial nitrogen 25 dynamics. The manuscript is well written and easy to read despite a little long and redundant. I suggest to be cautious about points mentioned below before the final acceptance.

Thank you for your general support, the positive recommendation, and for your constructive comments.

30  We edited the text to reduce redundancy.

One of my concerns is that the conclusion related to biological controls on N acquisition is already pre-included in the assumptions/definitions based on which the model is built. BNF in the manuscript refers to any N inputs, other than atmospheric 35 N deposition, that satisfy ecosystem N demand. With a constant annual N deposition rate, any changes in ecosystem N content and losses are attributable to BNF. Here BNF incorporates both biological and non-biological sources, which might come from weathering, be undiscovered N sources existed in pre-industrial time, or errors from assuming constant N deposition rate.

40  Thank you for this helpful comment. We adjusted the wording to make clear that

the N source in the model includes not only BNF. We generally refer now to 'N source' or 'N input' instead to 'BNF' in the manuscript. Text on p9, l1 is modified to read: The source of reactive N is implied by maintaining prescribed soil C:N ratios associated with each of the plant functional types, reflecting their different litter chemistries and decom­poser assemblages. This modelled N source, therefore, accounts for the input of reactive N by symbiotic and asymbiotic BNF, weathering, and any other potential N sources, apart from prescribed N deposition. The final sentences in the abstract reads now: Alternative mechanisms to explain the reconstructed $N_2O$ emissions include changes in $N_2O$ yield per N converted and changes in N sources from weathering.

A second concern is about the adjustments of global inflow of reactive N on multi-decadal to century time scales derived from N2O dynamics. As the authors mentioned, there are multiple-steps and many factors come into play in global N cycle. Different N2O production pathways may evolve through time, alternations of soil organic matter decomposition, stoichiometry and other N loss pathways are likely to shift N2O emissions. These adjustments are likely to occur without significantly alteration of real biological nitrogen fixations. For example, nitrifying and denitrifying microbes may have different temperature sensitivities vs. BNF. Vegetation and microbial evolution are largely unconsidered in this study. There is no strong evidence that N input flux would adjust as quickly as that of N2O emissions.

We agree with the reviewer that a range of factors complicate the link between sources of reactive N and $N_2O$ emissions. We adjusted the formulations in the abstract (see previous comment). Potential changes in the yield factors are discussed on page 19 in the submitted MS. We now discuss these factors early in the MS and acknowledge in section 2, 4 and 6 that the $N_2O$ emission increase may also be explained by the changes mentioned by the reviewer. For example, we add at the end of section 2.1: The amount of $N_2O$ produced per unit of N converted varies with environmental conditions and production pathway. For nitrification and denitrification this yield (or emission) factor depends on substrate availability linked to soil organic matter decomposition and C:N stoichiometry, on oxygen level influenced by soil moisture status, on acidity and temperature (Diem et al., 2017;Davidson et al., 2000;Firestone and Davidson, 1989;Smith, 1997;Phillips et al., 2015;Saggar et al., 2013). In addition, different $N_2O$ production pathways and N loss processes and their relative importance may evolve through time and influence the $N_2O$ yield on local to global scales. Please see also our response to reviewer 1 or the attached MS.

Further, we quantified the influence of variations in yield on $N_2O$ emissions changes in an additional sensitivity simulation discussed in section 5.2 (see reply to next point).

A third concern is related to insights to be learned from this study. Is it necessary to conduct spatially explicit model simulation to test constant vs. dynamic BNF? The magnitude of N2O emissions can be easily tuned through RN2ODN, whereas the "openness" or 'tightness' of N cycle is, to a large extent, conceptual and not new

in literature. The spatial pattern is also within our general understanding of global ecosystems as the model is built upon the contemporary (not paleo-) biogeochemistry and driven by historical climate. I feel the climate sensitivity of N2O emissions are valuable information that worth exploring for models like LPX-Bern.

5      We are not aware of any publication where the influence of a constant versus a dynamic N source is quantified in deglacial simulations. Therefore, we would like to inform the reader about these model results. The absolute magnitude of $N_2O$ emissions is tunable by adjusting the yield factor. However, model outcomes in terms of deglacial change are not a priori clear and easy to predict as NPP, vegetation growth, soil carbon 10 storage, N remineralisation, BNF, N loss fluxes, and yield factors undergo complex changes in space and time over the last 21,000 years as evident in figures 3, 4, 6, 7, and 8.

     We are not sure what the reviewer means with "historical climate"; in the context of Earth System modeling the historical period is typically taken from 1800 AD to present. 15 Here, the model was driven with temperature and precipitation output from a transient simulation with the Community Earth System Model over the last 21,000 years and with transient $CO_2$ and time varying orbital parameters.

     In response to this comment, we rewrote section 5.2 (please see the attached MS). We provide now additional context to clarify the implications of the simulations with 20 variable versus constant N source. We added results from an additional sensitivity simulations to quantify the influence of deglacial changes in the $N_2O$ yield factors. The first paragraph of section 5.2 reads now: In this section, we address C-N coupling in LPX-Bern and analyze the spatial patterns for the source of reactive N, soil mineral N, net primary productivity (NPP), and C stocks (Fig. 5) and their changes over the 25 deglaciation (Fig. 6). We quantify two decisive factors for $N_2O$ emission change in the model: (i) changes in the source of reactive N, fueling nitrification and denitrification, and (ii) changes in the $N_2O$ yield per unit N converted.

     The paragraph describing the new sensitivity run reads: The $N_2O$ yield factors, i.e., the $N_2O$ produced per unit N converted by denitrification and nitrification, are 30 assumed to vary with temperature and thus in space and time in LPX-Bern v1.4N. In a sensitivity run, these yield factors are set constant with all other settings as in the standard. The deglacial warming leads to a higher $N_2O$ yield in the standard compared to this sensitivity run and 0.44 TgN yr$^{-1}$ of the deglacial increase in land $N_2O$ emissions are attributed to this change in yield (Fig. 7, black line). In other words, changes in 35 the yield factors further amplify the increase in $N_2O$ emissions as driven by the increase in the flow of reactive N in LPX-Bern.

     The last paragraph of section 5.2 reads now: In summary, the simulation with constant N source completely fails to reproduce the reconstructed $N_2O$ emissions from the land biosphere. The increase in $N_2O$ yield, as well as in soil and litter C and 40 N turnover rates, under deglacial warming are not sufficient to overcome the effect of N limitation on $N_2O$ emissions in this sensitivity simulation. We note, however, that changes in yield due to processes not incorporated in LPX-Bern could potentially

explain the reconstructed increase in $N_2O$ emissions. If the model is allowed to satisfy the demand of N, and thereby implicitly of other elements to support the growth of N fixers, nitrifiers and denitrifiers, and plants, terrestrial $N_2O$ emissions increase as reconstructed.

We acknowledge that other factors than N input may had influenced $N_2O$ emissions and deleted the text on p14,l11 to l18

Specific points:

1. P6L25-30. Does it worth discussion on losses of plant-available vs. plant-unavailable (e.g., through fire and leaching of DON) N and how losses of plantunavailable N alter system dynamics?

Text added as requested. The text on p6,l25-30 has been deleted in response to reviewer 1. We have extended the discussion on fire, DON, and mineral adsorption in the paragraph on p6, l4: On larger scale, N lost by fires will be deposited again and a large part of this N flux is therefore fed again to land ecosystems. In this sense, the fraction of the fire flux not lost to the ocean may be viewed to belong to the internal global land N cycle. N leached as dissolved organic N is typically remineralized downstream and may undergo nitrification and denitrification or be taken up by aquatic organisms. N may also be absorbed by minerals and become unavailable for plants and microbiological assemblages.

2. P10L5. It is unclear when the upper limit of denitrification is used

[revised manuscript text omitted]

5   their high N:C ratios.  Plant net primary productivity (NPP) and a prescribed constant N:C ratio of new production in different tissues sets the N demand that is satisfied by N uptake from $NH_4^+$ and $NO_3^-$ pools which in turn depend on net N mineralization fluxes from litter and soil pools and loss fluxes of reactive N (e.g. denitrification, leaching, volatilization) (Fig. 1). In case that available inorganic N (sum of $NH_4^+$ and $NO_3^-$) is insufficient to meet the demand, NPP is down-regulated, thereby inducing an effect of N limitation. BNF tends to re-

10  establish a balance between the input and the loss of reactive N.

Estimates of the global BNF for non-agricultural ecosystems are uncertain. They range from 40 to 470 TgN yr$^{-1}$, with most published estimates around 100 to 150 TgN yr$^{-1}$. Cleveland et al. (1999) used 100 plot-scale estimates of BNF to estimate global BNF on natural ecosystems to 195 TgN yr$^{-1}$ (range: 100 to 290 Tg N yr$^{-1}$). Vitousek et al. (2013) suggest a plausible

15  range for preindustrial BNF of 40 to 100 TgN yr$^{-1}$ by computing BNF as the difference from all other global sources and sink fluxes of N. Cleveland et al. (2013) estimate symbiotic BNF to 105 TgN yr$^{-1}$, based on cost-benefit modelling for N fixation. The same authors estimate asymbiotic N fixation to 22 TgN yr$^{-1}$ by upscaling measurements reported in Cleveland et al. (1999). In contrast, Elbert et al. (2012) estimate asymbiotic N fixation by cryptogamic covers alone to 49 TgN yr$^{-1}$ (27-99 TgN yr$^{-1}$) by integrating experimental data from 200 studies. In addition, rock weathering is estimated to add 15 (10 to 20) TgN yr$^{-1}$ to

20  land ecosystems (Houlton et al., 2018). Xu-Ri and Prentice, (2017) estimate global BNF to 340 (230–470) TgN yr$^{-1}$ based on stoichiometric constraints within the LPX model; this estimate includes contributions from rock weathering and other inputs that are not explicitly prescribed or simulated by LPX. Meyerholt et al. (2016) implemented six different BNF formulations in their model to predict modern BNF ranging from 108 to 148 TgN yr$^{-1}$.

The magnitude of the modelled N source co-governs the rate of nitrification, denitrification, and leaching and, together with the yield factors, $N_2O$ emissions. As noted above, the model parameters governing the $N_2O$ yield are calibrated towards a preindustrial $N_2O$ emission of 5.9 TgN yr$^{-1}$. A high N source in the model, therefore, implies low $N_2O$ yield factors and vice

30  versa. In LPX-Bern, the magnitude of the simulated global N source  is partly adjustable by two scaling parameters. These are the fractions of re-mineralized N that is returned to litter and soil by immobilization, respectively.  The immobilization fractions are set to 0 % for litter and 60 % for soil mineralization in the standard setup. This results in a N source flux of 128 TgN yr$^{-1}$ (Fig. 1) with an implied weathering source of 15 TgN yr$^{-1}$ and BNF of 113 TgN yr$^{-1}$, well within the published range ~~(~60 to 340~~

TgN yr⁻¹) (Vitousek et al., 2013;Cleveland et al., 1999;Xu and Prentice, 2017;Houlton et al., 2018;Cleveland et al., 2013). The extent of immobilization of freshly added N is found to vary between 35 and 95 % from one soil to another with uptake by soil microorganisms (, wwith a typical turnover of 1-2 months), dominating over abiotic processes (Bengtsson et al., 2003). The global mean yield factor for denitrification, expressed as $N_2O$ per $N_2$ produced, is 5.6 % and thus higher than the range of estimates (0.2–4.7 %) summarized by Xu-Ri and Prentice (2008). Similarly, the yield factor for nitrification (0.23 %) is higher than observation-based estimates (0.01-0.2 %). The mismatches in these estimates for yield may suggest that current best estimates for the N source, $N_2O$ yield, and preindustrial $N_2O$ emissions are not fully consistent. We carried out In two a sensitivity simulations to explore uncertainties: the immobilization fractions are is set to 25 % for both soil and litter or to 26.4 % for soil and to 50 % for litter immobilization0 % for litter and to 26.39 % for soil mineralization. Lowering iImmobilization increases lowers the $NH_4^+$ and $NO_3^-$ pools in the model and loss fluxes of reactive N. In turn, less a higher N sourceBNF is required to maintain a balance of reactive N, while implied $N_2O$ yield factors become lower. The pPreindustrial N source BNF is 310 and 188523 TgN yr⁻¹ in thisese two sensitivity simulation and thus about a factor of four higher than in the standard setup. Correspondingly, yield factors are about a factor four lower in this sensitivity run than in the standard. s and within the published range. The absolute magnitude of N source has some implication for N stress and 
[revised manuscript text omitted]

---

## Author Response (AR1)

**Reply to comments by reviewers and the editor - bg-2019-118**

$N_2O$ *changes from the Last Glacial Maximum to the preindustrial - part II: terrestrial* $N_2O$ *emissions constrain carbon-nitrogen interactions*

We thank the editor and both reviewers for assessing this manuscript and for their time and effort. Please find a revised manuscript, with text changes highlighted and updated figures, at the end of this document. Changes implemented after our reply in the journal's open discussion in response to the comments by the editor are marked in yellow. We hope that the reviewers and the editor find the revised version suitable for publication in Biogeosciences.

The main changes implemented are as follows:

- Reviewer I was concerned that a high value of BNF may invalidate our results. We adjusted the model parameter values, repeated all simulations, and updated figures and text. The simulated global source of reactive N, including BNF, is now in line with empirical estimates and other model results. In spite of this adjustment, our main findings on deglacial change in $N_2O$ emissions remain unchanged.

- The formulation of our working hypotheses apparently lead to misunderstanding. We deleted section 2.3 "Working Hypotheses" and corresponding text in the manuscript. We modified the framing of our results and now discuss uncertainties and different mechanisms to potentially explain the reconstructed $N_2O$ emissions in section 2 and 4, complementing the existing discussion in section 6. We rewrote the abstract and state in the abstract: "The increase [in $N_2O$ emissions] may be explained by an increase in the flux of reactive N entering and leaving ecosystems or by an increase in $N_2O$ yield per unit N converted. ... Our results appear consistent with suggestions of (a) biological controls on ecosystem N acquisition, and (b) flexibility in the coupling of the C and N cycles during periods of rapid environmental change. Alternative mechanisms to explain the reconstructed N2O emissions include changes in N2O yield per N lost through gaseous pathways."

- We restructured and revised the text, in particular parts of the model description in section 3.1, section 5.2 and 6 to avoid redundancy, to adjust to the revised framing of our results and in response of specific review and editor comments.

The original review comments are given below in black, our reply in blue, and quotes from the revised manuscript in gray. Please note that page, line and section numbers refer to the originally submitted manuscript or where indicated in parentheses to the revised manuscript (without track change).

**Reply to Reviewer #1**

*Interactive comment on Biogeosciences Discuss., https://doi.org/10.5194/bg-2019-118, 2019.

This paper takes new ice-core data on nitrous oxide emissions over the deglaciation
and compares it to modelled results so as to better understand biological nitrogen fixation (BNF) in the same period. There follows a section on the contribution of different climate drivers on nitrous oxide emissions in separate attribution simulations.

The abstract is rather misleading, focusing on the work of the 'part I' paper which
describes the ice-core data, and BNF rather than attribution modelling.
The reconstruction of the marine and terrestrial $N_2O$ emissions is described in part I. Part I also provides a first interpretation of marine emission changes (see page 4 l7 to 9). However, the analysis and interpretation of the reconstructed terrestrial $N_2O$ emission changes is the topic of part II. This is stated in the abstract. The corresponding text is found in sections 4 and 6.1 in the MS. We shorten the text summarizing specific findings of part I on l18 to l24 in abstract. The new text reads: Here we analyse the large increase in terrestrial $N_2O$ emissions over the past 21,000 years as reconstructed from ice-core isotopic data and presented in part I of this study. Remarkably, the increase occurred in two steps, each realized over decades and within maximum two centuries, at the onsets of the major deglacial northern hemisphere warming events. The data suggest a highly dynamic and responsive global N cycle. The increase may be explained by an increase in the flux of reactive N entering and leaving ecosystems or by an increase in $N_2O$ yield per unit N converted.
We modified the last sentence of the introduction (p4, l10) to read: Here in part II we focus on the interpretation of the terrestrial $N_2O$ emission record and discuss terrestrial $N_2O$ emissions and C-N responses in transient deglacial simulations with a dynamical vegetation/terrestrial biogeochemistry model.

However, there are two problems with the main BNF part of this paper: the model's
representation of BNF, and the hypotheses posed. These two issues combined make the model runs virtually meaningless and the conclusions baseless.
We understand that the reviewer is concerned about the high value of BNF. We also understand that our formulations of working hypotheses did not help to convey our results as intended. These two issues are now addressed. We present results for modelled BNF (N source) close to the estimate by Cleveland et al., 2013. We revised the presentation of our results and removed section 2.3 "Working hypotheses" and related text from the MS. Nevertheless, some key statements by the reviewer regarding model formulations, model sensitivity, and assumptions are not correct (see below). We address these issues in the discussion below and have revised the text in the MS to clarify these issues.

The model BNF

The model BNF (page 9, 13 and Fig.1) is 523 TgN/yr globally in the pre-industrial simulation. The authors acknowledge this is "higher than the published range" but are selective about what range they are referring to and what the implication of this is. Their BNF is an order of magnitude more than low budget-based estimates (Vitousek et al., 2013) (44/58 TgN/yr) and almost 200TgN/yr larger than the upper model estimate (Xu-Ri and Prentice, 2017) (340 TgN/yr) they reference. The authors cite Cleveland et al., (1999) (195 TgN/yr), but interestingly fail to cite the more recent paper by the same author, revising the estimate down to 127.5 TgN/yr (Cleveland et al., 2013). The authors' reference to Lenhart et al., (2015) (page 9, line 16) as part of the "published range" of global BNF estimates is baffling, as the paper discusses nitrous oxide and methane emissions, not BNF. Moreover, the authors fail to mention that of all estimates of BNF in the last half century, only one (Xu-Ri and Prentice, 2017) is over 300 TgN/yr and most are around 100 – 150 TgN/yr.

We added the following text on line p9, l10 (revised MS: p8, l21) Estimates of the global BNF for non-agricultural ecosystems are uncertain. They range from 40 to 470 TgN $yr^{-1}$, with most published estimates around 100 to 150 TgN $yr^{-1}$. Cleveland et al. (1999) used 100 plot-scale estimates of BNF to estimate global BNF on natural ecosystems to 195 TgN $yr^{-1}$ (range: 100 to 290 Tg N $yr^{-1}$). Vitousek et al. (2013) suggest a plausible range for preindustrial BNF of 40 to 100 TgN $yr^{-1}$ by computing BNF as the difference from all other global sources and sink fluxes of N. Cleveland et al. (2013) estimate symbiotic BNF to 105 TgN $yr^{-1}$, based on cost-benefit modelling for N fixation. The same authors estimate asymbiotic N fixation to 22 TgN $yr^{-1}$ by upscaling measurements reported in Cleveland et al. (1999). In contrast, Elbert et al. (2012) estimate asymbiotic N fixation by cryptogamic covers alone to 49 TgN $yr^{-1}$ (27-99 TgN $yr^{-1}$) by integrating experimental data from 200 studies. In addition, rock weathering is estimated to add 10 to 20 TgN $yr^{-1}$ to land ecosystems (Houlton et al, 2018). Xu-Ri and Prentice, (2017) estimated global N sources to 340 (230–470) TgN yr-1 for the parameter settings they adopted within the LPJ-DyN model; this estimate includes contributions from rock weathering and other inputs that are not explicitly prescribed or simulated by LPJ-DyN. Meyerholt et al. (2016) implemented six different BNF formulations in their model and predict modern BNF ranging from 108 to 148 TgN $yr^{-1}$.

Lenhart et al. 2015 discuss $N_2O$ emissions from cryptogamic cover and summarize the results for N fixation by cryptogamic covers from Elbert et al., 2012. We deleted the reference to Lenhart et al., 2015 and added the reference to Cleveland et al., 2013 on p9, l16. We now explicitly cite the results of Elbert et al., 2012 on p9, l10 (see above).

These global BNF estimates of around 100 TgN/yr are because field experiments show that nitrogen fixation is relatively unusual in the terrestrial biosphere. Whilst individual nitrogen fixing plants or organisms have the potential to fix large amounts of nitrogen, they are best suited to 'pioneer' environments with low soil nitrogen, are usually found at very low densities in mature ecosystems, and may be facultative (rather than obligate) fixers. Taking one example, tropical forest is generally thought to be the highest BNF region due to its high NPP and low nitrogen limitation. Recent work by Sullivan et al., (2014) found that tropical forest in Costa Rica has BNF of 0.5 gN/m2/yr. From the map of pre-industrial BNF in LPJ-Bern (Figure 5 A) it seems BNF in Costa Rica is modelled at 10 gN/m2/yr. i.e. the model overestimates BNF in the tropics by a factor of 20. Though no present day BNF value is given in the paper, from the information available the present-day modelled value for Costa Rica is likely higher. Even compared to the upper bound for tropical forest of 6 gN/m2/yr from the meta-analysis done by Cleveland et al., (1999), the values in LPJ-Bern are high. Cleveland et al., (1999) said their upper limit was "extremely unlikely" and the global BNF from those upper values was 290 TgN/yr (compared to LPJ-Bern's 523 TgN/yr).

The parameters accounting for N immobilization were inadvertently set to low values by Lienert and Joos (2018). This in turn leads to a high N source in the model. We decided to use this published version for the originally submitted MS, complemented by the results from sensitivity simulations with a higher N flux representing immobilization and lower BNF. This approach was selected for reasons of traceability and transparency. We re-ran all simulations with an updated parameter set to yield a global preindustrial and modern N source of 128 and 140 TgN yr$^{-1}$, respectively. This flux of reactive N to ecosystems implicitly includes contributions from symbiotic and asymbiotic N fixation, rock weathering, and possibly other, yet to be discovered, pathways. In view of this, we adjusted the text and now refer to 'ecosystem N inputs' or 'N source' in general rather than BNF.

This is also relevant for comparing this flux with empirical estimates. Accounting for the N weathering flux of around 15 TgN yr$^{-1}$ (Houlton et al. 2018), the simulated flux of reactive N inputs presented here implies a modern BNF close to the reviewers preferred value of 127.5 TgN yr$^{-1}$. As noted already in the original MS (p9, l20-23), relative changes in modelled N source and N$_2$O emissions and related model-based conclusions are not sensitive to the exact parameter values. In addition, the yield factor for nitrification is assumed to vary with temperature in the same way as the yield for denitrification in this revised setup. This has a modest impact on modelled N$_2$O as only 10 % of the simulated N$_2$O flux stems from nitrification.

All figures in the manuscript were updated with results for this new parameter setting and included in the draft provided at the end of this document. Specifically, Fig, 5A displays the N source (BNF plus weathering and potential other abiotic sources) simulated by LPX-Bern. Typical values in Central America are around 1 gN m$^{-2}$ yr$^{-1}$ for the new setup, and comparable to field measurements (Sullivan et al., 2014; Wurzburg and Hedin, 2016). For example, Wurzburg and Hedin, 2016, report BNF of N fixers from 0 (for about half of the trees) to up to 12 gN m$^{-2}$ yr$^{-1}$ in their supplementary material. New text is added in section 5.2 (revised MS: p14, l31) to describe results displayed in Figure 5:

The N source (Fig. 5A) is typically smaller than 0.5 gN m$^{-2}$ yr$^{-1}$ in northern mid and high latitude and around 1 gN m$^{-2}$ yr$^{-1}$ in tropical rainforest and in Central America, comparable to observational estimates (Sullivan et al., 2014;Wurzburger and Hedin, 2016a, b). An N source of 2 to 6 gN m$^{-2}$ yr$^{-1}$ is simulated in many semi-arid regions, including southern Africa, the sub-Sahara region, India, northern Australia, and in the southern parts of North America. Soil mineral N is typically below 0.5 TgN m$^{-2}$ in the tropics ....

The BNF in LPJ-Bern is entirely disjointed from reality. This puts significant doubt on the ability of this model to produce meaningful results about or based on BNF. The authors infer on page 9 that the high BNF is irrelevant to their results. They describe two sensitivity experiments with lower global BNF and the nitrous oxide emissions are broadly similar. However, the two sensitivity experiment global BNF values (310 and 188 TgN/yr) are still unrealistically high. Therefore, these sensitivity experiments reveal that the problem with BNF in the model may not be a parameter based issue but could be something more fundamental.

As outlined above, the assumption by the reviewer regarding the sensitivity of the model to the absolute value of the N source and BNF is not correct. The text starting on p9, l11 (revised MS: p9, l26) is modified to read: The two-step calibration described above resulted in yield factors that are higher than the range of published estimates. The global mean yield for denitrification, expressed as N$_2$O per N$_2$ produced, is 5.6 % and thus higher than the range of estimates (0.2–4.7 %) summarized by Xu-Ri and Prentice (2008). Similarly, the global yield for nitrification (0.26 %) is higher than observation-based estimates (0.01-0.2 %). The mismatches in these estimates for yield may suggest that current best estimates for the N source, N$_2$O yield, and preindustrial N$_2$O emissions are not fully consistent. We carried out a sensitivity simulation to explore uncertainties: the immobilization fraction is set to 0 % for litter and to 26.39 % for soil mineralization, leading to a preindustrial N source of 523 TgN yr$^{-1}$. This is about a factor of four higher than in the standard setup. Correspondingly, yield factors are about a factor four lower in this sensitivity run than in the standard simulation.

The sensitivity of simulated N2O emission changes over the deglacial period to these parameter choices is relatively small, while the absolute magnitude of the N source has some implications for N stress and thus NPP. The increase in NPP over the deglaciation is larger in simulations with a high compared to a low N source (10.1 versus 5.9 GtC yr$^{-1}$ in the standard). Importantly, the difference in relative changes in modelled global N source is small (16 % versus 10 % in the standard) and the deglacial increase in N$_2$O emissions is only 0.2 TgN yr$^{-1}$ higher in the sensitivity than in the standard run (see Sect. 3.2 and 5), despite the large difference in the implied N source. Thus, related model-based conclusions for N$_2$O emissions are not sensitive to the parameter settings for the yield and the flux representing immobilization.

The hypotheses

The hypotheses set out on page 7 present a false dilemma. They are based on the premise that the nitrous oxide emissions are attributable to a nitrogen system that is either 'open' or 'closed'. The idea of either an open or closed system stems from the assumption that the nitrogen cycle is in equilibrium during the deglaciation (stated by the authors on page 6, line 5). However, there's no evidence either way on this question. Even if the assumption of equilibrium is accepted, an open system is not the only mechanism of increasing nitrous oxide emissions, thus the hypotheses are a false dilemma. The changes in nitrous oxide could also be caused by changes to the internal dynamics of the system (e.g. soil nitrogen turnover, or flexible C:N ratios, or the authors' assumption of homogeneous nitrous oxide yield fractions over space and time (page 8, line 26)).

We do not assume equilibrium over the deglaciation – neither when discussing the ice core record, nor the model results. The model applied here accounts for the transient dynamics of all C and N pools and fluxes and the main underlying principle is mass conservation. The highly transient nature of the ice core $N_2O$ emission record and of the N cycle in LPX-Bern is evident in Figs. 3, 4, 6, 7, 8, 10, and 11. To avoid the misunderstanding by the reviewer, we modified the sentence on p6, l5 (revised MS p6, l21) to read: Following mass balance, losses of reactive N from ecosystems and changes in ecosystem and abiotic N stocks have to be compensated by N inputs, mainly by BNF. In addition, we modified the last sentence of the introduction (p4, l10) to point the reader early on to the transient nature of our simulations. .. and discuss terrestrial $N_2O$ emissions and C-N responses in transient deglacial simulations with a dynamical vegetation/terrestrial biogeochemistry model.

Furthermore, we do not assume a constant $N_2O$ yield factor in space and time as asserted by the reviewer. The yield factor for denitrification, responsible for more than 85 % of the $N_2O$ flux in our model, is significantly varying in space and time as described on p8 l20 to l23 in the original MS.

It was not our intention to present a dilemma, but rather to guide the reader and place our analysis in the wider scientific context, connecting to fundamental question about ecosystem functioning and the nature of N limitation. We deleted section 2.3 and all text related to working hypotheses to avoid potential misinterpretations. Alternative explanations for the deglacial $N_2O$ increase are discussed in the original MS, e.g. related to abiotic N source (p6, l23), related to pre-existing stocks of reactive N on p18 l14 to 26), and related to changes in the yield factor per unit N converted on p19, l1 to 25. However, these additional explanations may be viewed as coming too late in the MS. We therefore adjusted the abstract as described in the second bullet at the beginning of this reply and added text in section 2 and 4. We added the following text at the end of section

2.1 on production mechanisms: The amount of $N_2O$ produced per unit of N converted varies with environmental conditions and production pathway. For nitrification and denitrification this yield (or emission) factor depends on substrate availability linked to soil organic matter decomposition and C:N stoichiometry, on oxygen level influenced by soil moisture status, on acidity and temperature (Diem et al., 2017;Davidson et al., 2000;Firestone and Davidson, 1989;Smith, 1997;Phillips et al., 2015;Saggar et al., 2013). In addition, different $N_2O$ production pathways and N loss processes and their relative importance may evolve through time and influence the $N_2O$ yield on local to global scales.

We added the following text at the end of section 2.2 on C-N-$N_2O$ coupling: In summary of section 2.1 and 2.2, changes in terrestrial $N_2O$ emissions may be linked (i) to changes in the magnitude of reactive N entering and leaving ecosystems, and (ii) to changes in the $N_2O$ yield per unit reactive N converted in land ecosystems. We added the following text towards the end of section 4 on reconstructed $N_2O$ emission: The rapid increase in terrestrial $N_2O$ emissions at the onset of the B/A and at the end of the YD and the overall increase in emissions over the past 21,000 years either point (i) to an increase in $N_2O$ yield per unit N converted for emissions to the atmosphere, averaged globally and across all $N_2O$ production pathways, or/and (ii) to an increase in the global flux of converted N.

The answer to the false dilemma presented is pre-decided by the definitions the authors give on page 6. Under the false dilemma, an 'open' (high-input-high-output) system is presented as the only mechanism that can produce high terrestrial nitrous oxide emissions. We know we have increasing (nitrous oxide) output, so under the assumption of either 'open' (high-input-high-output) or 'closed' (low-input-low-output), there is only one possible answer. This pre-determined result is exacerbated by the model. The high BNF in the model means the N cycle in the model must be 'leaky' and 'open' otherwise there would be no N limitation at all (contrary to the evidence, see LeBauer and Treseder, (2008)). Some models take a more 'closed' approach (often resulting in low BNF). But it stands to reason that closing the input of N in a model reliant on high N input will cause the model to produce results inconsistent with reality. These simulations might inform somewhat about the model but can't say anything about real world BNF.

The combination of an invalid hypothesis and an inappropriate model is results that mean nothing and conclusions that mislead. An unwary reader could easily take it at face value that BNF increased by 72 TgN/yr during the deglaciation and an 'open' terrestrial nitrogen system was the only, or most likely, way the observed nitrous oxide changes could have occurred. But there is no reliable evidence for either of these assertions.

We have to stress that these statements by the reviewer are not correct. As we show (please compare the figures in the original and the revised manuscript) results for changes in $N_2O$ emissions are similar for low (128 TgN $yr^{-1}$) and high (523 TgN $yr^{-1}$) input of reactive N as already discussed above. In addition, we would like to emphasize the distinction between state and change – We do not make statements whether current ecosystems are N limited or not, but are focussing on the *temporal change* as stated on p7,l5 ("To guide further discussion .. for the temporal evolution ..) and on p7,l16-22 ("The question posed in this study is not to what extent different ecosystems are, or have been N, limited. Rather, we ask the question whether BNF and the N cycle adjusted dynamically ...") in the original manuscript. Apparently, we failed to make this point sufficiently clear to the reviewer. As noted above, section "2.3 Working hypotheses" on p6 and 7 is deleted to avoid confusion and misinterpretations. As also noted above, different mechanisms of change are now discussed in the abstract and in section 2 and 4. In addition, we restructured the discussion section and discuss potential changes in $N_2O$ yield per unit N converted early in section 6.1. We state in the conclusion (sect.7): Our model results provide insight into the multi-decadal-to-millennial dynamics of the terrestrial C-N cycling by showing that the ice core terrestrial $N_2O$ emission record could potentially be explained with a rapid adjustment of N cycling to the climate and $CO_2$-driven acceleration of the C cycle, but they do not exclude the possibility that alternative explanations linked to changes in $N_2O$ yield could be important.

In summary, we aim to present our results in a balanced way and acknowledge the complexity of the C-N cycle. We thank the reviewer for pointing out shortcomings in the presentation of our results, however, we have to reject some of her/his statements and conclusions regarding our results as detailed above. We feel that our additional model runs and the revisions of the manuscript sufficiently address the more fundamental concerns of reviewer #1

The second part of the paper on attribution appears more sensible (issues with the BNF representation in the model not withstanding). This part of the paper might be appropriate for resubmission separately, without the BNF model results. With the two presented together it's difficult to assess the attribution section fairly.

The manuscript of Joos et al combines ice-core derived terrestrial N2O time-series with process-based N2O simulations to derive constraints on terrestrial nitrogen dynamics. The manuscript is well written and easy to read despite a little long and redundant. I suggest to be cautious about points mentioned below before the final acceptance.

Thank you for your general support, the positive recommendation, and for your constructive comments.

We edited the text to reduce redundancy.

One of my concerns is that the conclusion related to biological controls on N acquisition is already pre-included in the assumptions/definitions based on which the model is built. BNF in the manuscript refers to any N inputs, other than atmospheric N deposition, that satisfy ecosystem N demand. With a constant annual N deposition rate, any changes in ecosystem N content and losses are attributable to BNF. Here BNF incorporates both biological and non-biological sources, which might come from weathering, be undiscovered N sources existed in pre-industrial time, or errors from assuming constant N deposition rate.

Thank you for this helpful comment. We adjusted the wording to make clear that the N source in the model includes not only BNF. We generally refer now to 'N source' or 'N input' instead to 'BNF' in the manuscript. Please see also our response to the last comment by the editor given further below. The text in the model description (originally p9, l1; revised p8, l11) is modified to read: .. The remainder determines the total input of reactive N into the ecosystem, implicitly subsuming symbiotic and asymbiotic BNF, and any other potential N sources that may support plant growth, in addition to prescribed N deposition.

Text on p17, l20 (revised p20, l3) is modified to read: Sources of reactive N on land, e.g., from BNF and weathering, may possibly have increased under warming climate and increasing $CO_2$ over the deglacial period and contributed to meet the N demand of plants, nitrifiers, denitrifiers and cryptogamic covers under more favorable growth conditions.

Text on p18, l13 (revised p20, l14) is added: Similarly, it remains unclear how other smaller sources of reactive N changed over the deglacial period and influenced $N_2O$ emissions.

A second concern is about the adjustments of global inflow of reactive N on multi-decadal to century time scales derived from N2O dynamics. As the authors mentioned, there are multiple-steps and many factors come into play in global N cycle. Different N2O production pathways may evolve through time, alternations of soil organic matter decomposition, stoichiometry and other N loss pathways are likely to shift N2O emissions. These adjustments are likely to occur without significantly alteration of real biological nitrogen fixations. For example, nitrifying and denitrifying microbes may have different temperature sensitivities vs. BNF. Vegetation and microbial evolution are largely unconsidered in this study. There is no strong evidence that N input flux would adjust as quickly as that of N2O emissions.

We agree with the reviewer that a range of factors complicate the link between sources of reactive N and $N_2O$ emissions. We adjusted the formulations in the abstract (see previous comment). Potential changes in the yield factors are discussed on page 19 in the submitted MS. We now discuss these factors early in the MS and acknowledge in section 2, 4 and 6 that the $N_2O$ emission increase may also be explained by the changes mentioned by the reviewer. For example, we add at the end of section 2.1: The amount of $N_2O$ produced per unit of N converted varies with environmental conditions and production pathway. For nitrification and denitrification this yield (or emission) factor depends on substrate availability linked to soil organic matter decomposition and C:N stoichiometry, on oxygen level influenced by soil moisture status, on acidity and temperature (Diem et al., 2017;Davidson et al., 2000;Firestone and Davidson, 1989;Smith, 1997;Phillips et al., 2015;Saggar et al., 2013). In addition, different $N_2O$ production pathways and N loss processes and their relative importance may evolve through time and influence the $N_2O$ yield on local to global scales. Please see also our response to reviewer 1 or the attached MS.

Further, we quantified the influence of variations in yield on $N_2O$ emissions changes in an additional sensitivity simulation discussed in section 5.2 (see reply to next point).

A third concern is related to insights to be learned from this study. Is it necessary to conduct spatially explicit model simulation to test constant vs. dynamic BNF?

The magnitude of N2O emissions can be easily tuned through RN2ODN, whereas the "openness" or 'tightness' of N cycle is, to a large extent, conceptual and not new in literature. The spatial pattern is also within our general understanding of global ecosystems as the model is built upon the contemporary (not paleo-) biogeochemistry and driven by historical climate. I feel the climate sensitivity of N2O emissions are valuable information that worth exploring for models like LPX-Bern.

We are not aware of any publication where the influence of a constant versus a dynamic N source is quantified in deglacial simulations. Therefore, we would like to inform the reader about these model results. The absolute magnitude of $N_2O$ emissions is tunable by adjusting the yield factor. However, model outcomes in terms of deglacial change are not a priori clear and easy to predict as NPP, vegetation growth, soil carbon storage, N remineralisation, N source, N loss fluxes, and yield factors undergo complex changes in space and time over the last 21,000 years as evident in figures 3, 4, 6, 7, and 8.

We are not sure what the reviewer means with "historical climate"; in the context of Earth System modeling the historical period is typically taken from 1800 AD to present. Here, the model was driven with temperature and precipitation output from a transient simulation with the Community Earth System Model over the last 21,000 years and with transient $CO_2$ and time varying orbital parameters.

In response to this comment, we rewrote section 5.2 (please see the attached MS). We provide now additional context to clarify the implications of the simulations with variable versus constant N source. We added results from an additional sensitivity simulations to quantify the influence of deglacial changes in the $N_2O$ yield factors. The first paragraph of section 5.2 reads now: In this section, we address C-N coupling in LPX-Bern and analyze the spatial patterns for the source of reactive N, soil mineral N, net primary productivity (NPP), and C stocks (Fig. 5) and their changes over the deglaciation (Fig. 6). We quantify two decisive factors for $N_2O$ emission change in the model: (i) changes in the source of reactive N, fueling nitrification and denitrification, and (ii) changes in the $N_2O$ yield per unit N converted.

The paragraph describing the new sensitivity run reads: The $N_2O$ yield factors, i.e., the $N_2O$ produced per unit N converted by denitrification and nitrification, are assumed to vary with temperature and thus in space and time in LPX-Bern v1.4N. In a sensitivity run, these yield factors are set constant with all other settings as in the standard. The deglacial warming leads to a higher $N_2O$ yield in the standard compared to this sensitivity run and 0.44 TgN yr$^{-1}$ of the deglacial increase in land $N_2O$ emissions are attributed to this change in yield (Fig. 7, black line). In other words, changes in the yield factors further amplify the increase in $N_2O$ emissions as driven by the increase in the flow of reactive N in LPX-Bern.

The last paragraph of section 5.2 reads now: In summary, the simulation with constant N source completely fails to reproduce the reconstructed $N_2O$ emissions from the land biosphere. The increase in $N_2O$ yield, as well as in soil and litter C and N turnover rates, under deglacial warming are not sufficient to overcome the effect of N limitation on N$_2$O emissions in this sensitivity simulation. We note, however, that changes in yield due to processes not incorporated in LPX-Bern could potentially explain the reconstructed increase in N$_2$O emissions. If the model is allowed to satisfy the demand of N, and thereby implicitly of other elements to support the growth of N fixers, nitrifiers and denitrifiers, and plants, terrestrial N$_2$O emissions increase as reconstructed.

We acknowledge that other factors than N input may had influenced N$_2$O emissions and deleted the text on p14,l11 to l18

Specific points:

1. P6L25-30. Does it worth discussion on losses of plant-available vs. plant-unavailable (e.g., through fire and leaching of DON) N and how losses of plantunavailable N alter system dynamics?

Text added as requested. The text on p6,l25-30 has been deleted in response to reviewer 1. We have extended the discussion on fire, DON, and mineral adsorption in the paragraph on p6, l4 (revised p6, l17): On larger scale, N lost by fires will be deposited again and a large part of this N flux is therefore fed again to land ecosystems. In this sense, the fraction of the fire flux not lost to the ocean may be viewed to belong to the internal global land N cycle. N leached as dissolved organic N is typically remineralized downstream and may undergo nitrification and denitrification or be taken up by aquatic organisms. N may also be absorbed by minerals and become unavailable for plants and microbiological assemblages.

2. P10L5. It is unclear when the upper limit of denitrification is used

The statement should refer to nitrification. Text clarified to read: An exception is an adjustment in the upper limit of the fraction of NH$_4^+$ nitrified per day from 0.1 day$^{-1}$ to 0.09096 day$^{-1}$ at 20$^o$C

**Reply to Comments by the Editor**

Associate Editor Decision: Reconsider after major revisions (05 Jul 2019) by Sönke Zaehle
Comments to the Author:

Dear authors, many thanks for your comments and also for providing a quantitative response to the criticism raised by the reviewers. The proposed revisions are of a nature that suggest that a suitably revised and reworked manuscript may become acceptable to Biogeosciences. Please make sure that the revised manuscript reflects all points raised by the reviewers, and provides a balanced discussions of these points raised. At this stage, I do not provide detailed comments or a profound assessment, but offer some further guidance for revision with respect to your responses, should you decide to submit a revised version of the manuscript. Please note that a revised manuscript would undergo a full second round of peer-review.

Thank you for your editorial guidance.

(1) The new model version has a much improved estimate of BNF, which makes the results appear more plausible. However, how you have gotten to this result remains unclear from the revisions. I would recommend to detail the changes you have made between the first submission and the revised version (that could be an Appendix), and also offer some key statistics as to how the models differ (e.g. global BNF, NPP, C storage, N leaching and total N gas loss). Since BNF in your model seems to be calibrated by the N immobilisation process, a more in-depth description on how this process works would be appreciated. Somewhere in the model description it reads that the immobilisation process was introduced later into the LPJ-DyN, which is a curious statement, given that N immobilisation is an essential part of the SOM decomposition process, and it is hard to imagine that one can describe a model as prognostic and full representation of the N cycle if this process is not represented.

An appendix is added to the revised manuscript that documents the changes in model parameters and selected model outcomes between LPX-Bern v1.4 and v1.4N as suggested. Please see the appendix in the re-submitted MS for further details.

Thank you for pointing out that our description of how N uptake by N immobilization is accounted for was too brief and not clear. The clause on p8, l14 "here modified to include N immobilization in soils" is clarified to read: accounting for the uptake of mineral N by N immobilization in soils (Bengtsson et al., 2003;Li et al., 2017;Gütlein et al., 2017) as in Xu-Ri and Prentice (2017).

The text from p8, l13 to p9, l23 in the submitted MS has been restructured and expanded to better explain the approach for the N source, the flux accounting for immobilization, and for model calibration. The paragraph reads now: In LPX, the source of reactive N is implied by maintaining prescribed soil N:C ratios associated with each of the plant functional types, reflecting their different litter chemistries and decomposer assemblages. Due to lower N:C ratios of litter than soil pools, the transfer of mass from litter to soil pools during litter decomposition therefore implies a given amount of N, required to satisfy the soil N:C ratio. The required N is partly satisfied by a flux representing immobilization of mineral N. The remainder determines the total input of reactive N into the ecosystem, implicitly subsuming symbiotic and asymbiotic BNF, and any other potential N sources that may support plant growth, in addition to prescribed N deposition. The amount of N input required to close the N balance of soils and to maintain the soil pools at their high N:C ratios depends on the flux representing

N immobilization. Constant fractions ($frac\_soil\_immob$, $frac\_litter\_immob$) of the N flux released by soil or litter remineralization are immediately returned to its pool of origin. Hence, the choice of these parameters simultaneously co-determine reactive N input rates and net N mineralization.

We assumed the parameters $frac\_soil\_immob$ and $frac\_litter\_immob$ to be invariant over time and space and calibrated their values (Table A.1) to match a total preindustrial reactive N source of 128 TgN yr$^{-1}$. This value implicitly includes contributions from symbiotic and asymbiotic BNF, as well as other inputs of reactive N not included in the prescribed N deposition. ..

The wording for "immobilization" is adjusted in the MS (e.g.: "on the flux representing N immobilization"; "account for the uptake of mineral N"; "representation of N uptake by N immobilization")

     The model caveat on p20, l30 (revised p21, l1) is expanded by: and a constant fraction of remineralized N is returned immediately to its source soil pool. to read: However,
microbial and fungal biomasses are - unlike in microbial-explicit models (Schimel and Weintraub, 2003;Zhu et al., 2017;Allison and Gessner, 2012) - not explicitly modelled and organic matter decomposition does not depend on microbial mass and physiology. Instead, a mass balance approach is applied with C:N stoichiometry prescribed at observation-based PFT-dependent values for litter and soils. A constant fraction
of remineralized N is returned immediately to its source soil pool and the N budget is closed by the implied N source flux (Fig. 1). There is also no distinction between different classes of organic matter according to their accessibility to microbial action (Averill and Waring, 2018).

It is surprising that you can tune the model to have a five-fold difference in BNF, but no perceivable difference in C and N cycle trajectories across the LGM to present-day discussion. This fact deserves some discussion (because this is certainly not the case for other N modelling concepts), in particular if the claim is that increased BNF is an important cause of the observed N2O increase.

[revised manuscript text omitted]

I do not see the need to explicitly refer to rock-based N weathering in this manuscript. The BNF in LPX-Bern is driven by the litter-layer decomposition process (which is largely devoid of weathering material), and does not depend on deeper soil layers N production rate (nor does it reflect the geographic patterns of weathering and lithology). As far as I understand, the essence of BNF in LPX-Bern is asymbiotic, and it would be clearer if that was stated rather than subsuming it with many terms that aren't actually represented in the model.

We agree with the editor that it is unlikely that changes in rock-based weathering are responsible for the increase in $N_2O$ emissions. We still mention the weathering source as a potential N source in section 1 and 2 as it is an integral part of the N cycle and once in section 6, here to address the concern raised by reviewer 2. However, we have deleted the term weathering in the abstract and whenever it occurs in connection with the LPX-Bern model as well as in other places in the MS. We believe that the discussion of the weathering is not central to our MS. The text for the N source in the model description (revised MS p8, l11) reads now: 
[revised manuscript text omitted]

---

## Referee Report (RR1)

Review of **"N$_2$O changes from the Last Glacial Maximum to the preindustrial - part II: terrestrial N$_2$O emissions and carbon-nitrogen cycle interactions"** by Fortunat Joos, Renato Spahni, Benjamin D. Stocker, Sebastian Lienert, Jurek Müller, Hubertus Fischer, Jochen Schmitt, I. Colin Prentice, Bette Otto-Bliesner, and Zhengyu Liu.

In the submitted manuscript – by now apparently into the third revision – Joos and co-authors investigate the changes in terrestrial N$_2$O emissions from the last glacial maximum to preindustrial times. They describe their model setup and the model experiments performed, and describe model results, as well as results from two sensitivity experiments. The authors can explain about half of the change in terrestrial N$_2$O emissions between LGM and preindustrial and discuss some factors that might lead to the underestimates by their model.

Overall, the manuscript is very well written, and about ready for publication. I am torn between recommending publication as is, and publication with small changes. However, the manuscript should definitely be published, as it is a pioneering effort in modelling the changes in N$_2$O emissions from the LGM to PI. While the authors cannot yet explain the full change, this publication is required by the community as a base to build upon in improving our understanding of well-documented biospheric changes from the past to the present.

I have no major issues with the manuscript. While the previous discussion between authors and reviewers indicates that there may have been some rather strong claims in previous iterations of the paper, I can find no fault in this respect with the present version of the manuscript. Claims by the authors appear well-supported by the author's results, and the model appears to be documented adequately. It may well be that some details of the implementation of the Nitrogen cycle in LPX-Bern leave something to be desired in the light of the most recent findings, but personally I am rather glad that there still is room for improvement, as it gives us something to build upon in the future.

However, there are two minor issues that may warrant revisiting the manuscript. Looking at the ice core reconstruction in Figure 3, it is striking that more than half of the emission change was realised in two very rapid steps. The model reproduces these step-like changes, although it underestimates the magnitude and the rapidity of these step-changes, as discussed in the manuscript. Later on in the Holocene, however, a further quarter of the reconstructed emission change was realised as a slow upward trend in N$_2$O emission, but the model completely fails to reproduce this upwards trend, it rather shows nearly constant emissions. I may have overlooked it, but so far I am missing a discussion of this feature.

A second issue is that I find some of the Figures slightly confusing, but I appreciate that this may be a matter of personal taste. In Figures 4-6, 8, and 9 I was irritated by the fact that I needed to read the Figure caption very carefully in order to understand what was shown. I was expecting to see a Figure title, indication Figure content, and overlooked the units and quantity shown next to the colour bar – obviously my mistake, but maybe the authors can find a way to make this slightly clearer.

---

## Author Response (AR2)

**Point-by-point reply to the comments by the editor and the reviewers for manuscript bg-2019-118**
(Joos et al. N$_2$O changes from the Last Glacial Maximum to the preindustrial - part II: terrestrial N$_2$O emissions and carbon-nitrogen cycle interactions)

We thank the editor for his time and effort to assess our manuscript. A point-by-point reply to the comments by the editor and reviewer #1 is provided in the following. Reviewer #2 had no further comments on the manuscript and judged its scientific significance and its scientific quality as well as the presentation quality as excellent and recommended publication of the MS as is. The original comments are given in normal fonts, our reply in blue, and modifications to the manuscript text in gray. Line and page numbers refer to the manuscript version submitted in July 2019. In addition to the changes outlined in our reply below, we also corrected for typographical errors and updated figure 7 to correct for an inconsistency in the sign of one curve. A revised version of the manuscript, with changes highlighted, is attached at the end of this reply.

**Reply to comments by editor**

Dear authors,

My sincere apologies for the delay in coming to a decision with regard to your manuscript. This was partly due to the IPCC internal deadlines, but also because the two reviewer assessments had diverging views on the merit of the revised draft (accept as is / reject).

The revised manuscript by Joos et al. is somewhat improved with respect to the overall plausiblity of the N cycle simulation (in particular related to the magnitude of the pre-industrial terrestrial N source (in the wording of the authors), and also with respect to removing the hypotheses critisied by one reviewer. There is a suitable set of analysis to understand the trends simulated by the model. There is a extensive discussion of the results, which in some cases could be shortened and focussed.
We appreciate any editorial suggestions.

The dilemma with the current version of the manuscript is that the model does not convincingly capture the magnitude of the LGM to Holocene N2O compared to what has been inferred in a companion paper based on ice core data. The authors provide an assessment of the underlying model trends, which is an interesting model exercise. However, the lacking ability of the model to reproduce the terrestrial N2O increase makes it questionable whether the increase in biologically driven N fixation is indeed a major cause for the increase in the terrestrial N2O source between LGM and Holocene as put forward by the authors, or the consequence of other processes or process rates not represented well in the model, or perhaps also issues with using the climatic reconstruction as input for the biosphere model. This weakens the message of the paper.
Our paper presents, to our knowledge, the first analysis that compares terrestrial N$_2$O emission changes from the Last Glacial Maximum to the preindustrial as simulated by a dynamic global vegetation model and as reconstructed from ice core data. We also provide a comprehensive analysis of underlying mechanisms and drivers for the spatio-temporal evolution of modelled N$_2$O response; results from nine

simulations extending over the past 21,000 years as well as additional sensitivity simulations are presented. The results are all novel and the work represents, in our opinion, a substantial scientific effort that has not been attempted before.

We do not agree that it is a dilemma of the manuscript that the model does not fully capture the magnitude of the LGM to Holocene $N_2O$ change compared to what has been inferred from the ice core data. The $CO_2$ community is struggling since 40 years ago to model glacial-interglacial changes in atmospheric $CO_2$. Despite, or rather because of this, many model studies are published towards an improved understanding of past $CO_2$ changes. Here, we present a first attempt to model land biosphere $N_2O$ emission changes over the deglacial and find that the direction and order of magnitude of change in modelled $N_2O$ emissions are consistent with the ice core reconstruction, although with reduced amplitude compared to the proxy data. Major climate-related excursions such as the Heinrich Stadial1-Bølling/Allerød –Younger Dryas warm-cold swings are also to some extent visible in the model outcome. We see the glass here as half full rather than half empty, in particular, as the model is only calibrated relative to preindustrial conditions and no tuning has been carried out to optimize the reconstruction of the glacial/interglacial amplitude.

Given the complexity of the subject, any model is by necessity a simplification and subject to parameter and structural uncertainties. The LPX model is well documented in the literature and we describe the various components in an open and transparent manner in the method section. We document the model's response to individual drivers and the dependency of the response on model assumptions (e.g., regarding biological N fixation) in carefully designed factorial simulations.

An important novel element of our work is that we expose a model to a new benchmark that captures the policy-relevant multi-decadal-to-century time scales for $N_2O$ emissions, a time scale missing in modern observational and experimental data. We recommend that the setup of our deglacial simulation could be used to compare and evaluate models, e.g., in the framework of the model intercomparison initiated by the Global Carbon Project (P23, L30 in MS). In this sense, our work prepares the ground for follow-up studies that may overcome the limitations of our study.

We acknowledge that the model fails to simulate the full amplitude of the deglacial change. We discuss potential reasons on P23 L1-L7: "LPX-Bern is known for a relatively low sensitivity to increasing $CO_2$ and simulates a modest increase in NPP and the terrestrial carbon sink over the industrial period (Lienert and Joos, 2018a). A larger deglacial increase in NPP would tend to increase the implied N source in the model and potentially increase nitrification, denitrification and $N_2O$ emissions. Further, denitrification processes are assumed to respond to the relevant substrate concentration following Michaelis–Menten kinetics (Xu-Ri and Prentice, 2008). This limits N conversion rates and $N_2O$ production at high substrate concentrations, whereas a recent synthesis of N fertilization experiments (Niu et al., 2016) points towards an exponential relationship between N load and $N_2O$ emissions."

We did not attempt to change the Michaelis-Menten formulation to an exponential formulation as we view this as unwarranted model tuning at this point.

We extensively discuss uncertainties associated with $N_2O$ emissions in the discussion section. We refined the formulation of the last sentence in the abstract to emphasize that the major uncertainty in the explanation of the  deglacial $N_2O$ increase arises from the poorly known sensitivity of $N_2O$ yield: A dominant uncertainty in the explanation of the reconstructed $N_2O$ emissions is the poorly known $N_2O$ yield per N lost through gaseous pathways and its sensitivity to soil conditions.

Unfortunately, I furthermore find the presentation of the results overreaching and misleading. This primarily relates to the following points:

1) In abstract and conclusion, the authors write that their study results are in reasonable agreement with the observations, which they are clearly not. While the N2O emissions increase from LGM to Holocene, the magnitude of this increase is at best 2/3 (probably even less) of the inferred increase from ice cores, decidedly outside the error range of that estimate. In addition there is a time-delay in that response compared to the icecores, and the temporal development in the Holocene is opposite to the trend in the ice-cores. This misrepresentation is emphasised even clearer in the main text, where the auhors state (p 22 L 32-33) that the model results are at the lower bound of the reconstructed range, whereas Fig. 3 clearly shows this not to be the case.

We thank you very much for pointing out these incorrect or misleading statements. We failed to update the text at several places after an upward revision in the ice core estimates during the writing process. We apologize for this mistake.

We reformulated the sentence on p1, L27 in the abstract to better reflect the data-model mismatch: LPX-Bern simulates a deglacial increase in $N_2O$ emissions, but underestimates the reconstructed increase by 47 %.

We reformulated the first paragraph of section 5.1 (p12 L14ff) to better reflect the data-model mismatch: LPX-Bern v1.4N simulates a general increase in global land $N_2O$ emissions over the deglacial period (Fig. 3). The modelled deglacial increase in $N_2O$ emissions is only 0.96 TgN yr$^{-1}$ and about 47 % smaller than the reconstructed increase of 1.8±0.3 TgN yr$^{-1}$ (±1 standard deviation; Fig. 3).  Modelled changes are typically less abrupt than reconstructed variations. The model shows emission variations during the Bølling/Allerød (B/A) and Younger Dryas (YD) periods with peaks in emissions at the onset of the BA (14.6 ka BP) and the preboreal (11.7 ka BP) and an emission peak around 13.5 ka BP and corresponding minima at 14 ka BP and during the YD (12.8 to 11.7 ka BP). Reconstruction and models show moderate and small changes in global terrestrial $N_2O$ emissions over the last 11 ka, the Holocene period. Simulated terrestrial $N_2O$ emissions decreased by 0.06 TgN yr$^{-1}$ between 10 and 0.5 ka BP, whereas reconstructed emissions increased by 0.41 TgN yr$^{-1}$ over the Holocene, leading to a discrepancy between simulated and reconstructed anomalies. The reconstruction suggests relatively constant emissions from the land biosphere during Heinrich Stadial 1 (HS1) and a rapid rise in emissions at the start of the B/A, whereas the model simulates steadily increasing emissions over the second half of the HS1 interval, reflecting the gradual climate warming in the TraCE-21kyr climate input data during HS1 (see discussion below). The model also shows a much slower and a smaller emission increase at the YD/PB transition than reconstructed.

The last part of the sentence on p15 L25 was modified by replacing "as reconstructed" with "substantially" to read: ..simulated terrestrial $N_2O$ emissions increase substantially.

The statement on P22 L32-33 was incorrect and deleted. The text now reads: LPX-Bern forced by TraCE-21kyr output simulates a deglacial increase in $N_2O$ emission but underestimates the reconstructed increase by about 50 %. This may be related to the model's …

We reformulated the sentence on p24, L19-L20 in the conclusion to better reflect the data-model mismatch: Here, we evaluated LPX-Bern in transient simulations over the past 21,000 years. The model simulates an increase in $N_2O$ emission over the deglaciation, but underestimates the magnitude of the deglacial increase in comparison with the ice core data.

Finally, we replace the word "constrain" by "and" and add the word cycle in the title to read: $N_2O$ changes from the Last Glacial Maximum to the preindustrial - part II: terrestrial $N_2O$ emissions and carbon-nitrogen cycle interactions

2) To this effect, the key figure of this manuscript, Figure 3, shows two model results in support of the results discussed in the abstract, one with the N source fixed and consequently little change in N2O, and another one with the full simulation, which produces ~2/3 or less of the signal. This would have been sufficient to make the case the authors presented in the abstract. It remains unclear to me (and misleading) why Figure 3 also displays a third model output (which coincidentally is the only model configuation that nearly reaches the lower confidence bound of the N2O source inferred from the ice-cores in agreement with the main text p22 L 32-33), based on the completely unrealistic assumption that land mass change did not change from LGM to present-day. While it is a useful scientific question to ask whether changes in N fixation did contribute to the change in terrestrial N2O, it is not a reasonable to assume that between LGM and Holocene neither sea level nor glacial extend did change. It is therefore unclear why it has been included here.

This criticism is not justified. The criticism is apparently a consequence of the issues raised under point 1). We included the result of the factorial simulation with constant land mask to demonstrate that changes in sea level are important and must be considered in any modelling work addressing deglacial $N_2O$ emissions from the land biosphere. We make this, in our opinion, important point for the modelling community in the abstract (p1, L27) and in the main text (P13 L8-L12, P15 L32-L33) and in the discussion (p23 L11-L13). The factorial run in Fig. 3 is clearly labelled as such in the figure legend and the figure caption and a thin dashed line is used to distinguish it from our standard run (thick solid line). We did not anticipate that the reader would assume this to be something else than a factorial, hence counterfactual, simulation. We removed the simulation with constant land mask from Fig. 3 to avoid any further misinterpretation. We did not change the corresponding text (except removing the corresponding reference to Fig. 3). The effect of land mask change is also shown in Fig. 7, albeit less clear than it was the case in Fig. 3.

3) The model has undergone extensive calibration to address the reviewer concerns of round 1. The current version of the manuscript compares the old and the new models versions, as if this was a simple sensitivity study. This glosses over the fact that the updated version from the previous manuscript now has N2O yield increasing with temperature, without there being a clear explanation for the experimental basis of this function, and the fact that literature cited by the authors suggests the opposite (their page 19, L 9, Firestone and Davidson, 1989). This is noteworthy, because this model change contributes to an increase of N2O of about half of the simulated LGM to Holocene increase in Fig 3. This fundamentally undermines the authors assertion that the previous and revised model versions show comparable changes in LGM to Holocene N2O irrespective of the pre-industrial or glacial N source magnitude, because this was only achieved by adding an additional temperature sensitivity on top of the

temperature sensitivities of organic N turnover and that of nitrification and denitrification.

The assumption of the editor is incorrect. The temperature function for the $N_2O$ yield of denitrification is exactly the same in all our previous submissions of this manuscripts and the same as in the original model published by Xu-Ri and Prentice.

The modification of the yield function for nitrification was introduced in response to the criticism of reviewer #1 who challenged the application of a spatially constant yield function. However, this change is of minor importance for global $N_2O$ emissions and contributed only by about 0.06 TgN yr$^{-1}$ or 6 % to the deglacial emission increase. This change in parameterization is described in the main text (p9 L20-24) and documented in the appendix (p25 L14-L15 and Tab. A1). The sign of this function is supported by work of Smith 1997 as cited in the MS. The text on p25 L14-L15 in the appendix has been complemented with an additional sentence and reads now: In v1.4, the yield factor from nitrification is assumed constant. In v1.4N the same temperature dependency as for the yield for denitrification is assumed. This is a relatively minor modification as nitrification contributes only 12 % to the modelled preindustrial $N_2O$ emissions.

Our statement still holds that the previous and revised model versions show comparable changes in LGM to Holocene $N_2O$ emissions. This agreement was NOT achieved by adding an additional temperature sensitivity. We added the magnitude of the deglacial emission change for v1.4 and v1.4N in the main text to allow the reader to judge the difference in emissions The text on P22 L24-L26 reads now: The most striking difference is that the implied N source is a factor of four higher in v1.4 compared to v1.4N. Nevertheless, results for deglacial $N_2O$ emission changes remain similar between the version v1.4 (change 1.13 TgN yr$^{-1}$) and v1.4N (0.96 TgN yr$^{-1}$).

We describe how yield may change under environmental conditions on P18 L24 to P 19 L16. We write on p18 L29-30: "This yield factor is known to vary with environmental conditions and across $N_2O$ production pathways, though quantitative experimental evidence is often equivocal." We refined the discussion on the temperature sensitivity of the yield factor for denitrification and the text reads now: The ratio of $N_2O$ to $N_2$ emitted has been thought to decrease with increasing temperature (Firestone and Davidson, 1989). However, the observational challenges to determine $N_2O$ and $N_2$ emissions are substantial and available results are equivocal. Butterbach-Bahl et al. (2002) analyzing results for a series of soil cores at spruce and beech sites in Germany, find it difficult to derive a general trend on the effect of temperature on the ratio of $N_2O$ and $N_2$ produced via denitrification. Phillips et al. (2015) shows for New Zealand pasture soil samples that denitrified-N shifts towards $N_2O$ with increasing temperature under anaerobic conditions.

Any parameterization of yield must be viewed with caution. The factorial experiment added in the previously resubmitted version allow the reader to gauge the magnitude of the influence of our parameterization on yield. We added on P21, L10 in the discussion: In LPX-Bern v1.4N, the yield factor is varying as a function of temperature for nitrification and denitrification, whereas dependencies of the yield on $NO_3^-$ availability, oxygen status, decomposable carbon, or pH are not considered (see discussion in previous section).

If the change in $N_2O$ yield has the wrong sign in the model under deglacial change as compared to the real world, this would imply that the conversion of N by nitrification and denitrification and thus the N loss term would need to increase more than simulated by LPX. The need for additional BNF to balance N loss and N uptake by vegetation would likely be larger than suggested by our results. Such a scenario

would point towards an enhancement of BFN over the deglaciation. As this is already discussed in the MS we do not add further text on this issue.

4) The manuscript further suffers from a misrepresentation of their model. In P1 L28 and elsewhere, the authors claim that their N source estimates are plant-demand based, implying that changes in plant growth and/or N limitation would have direct consequences for fixation, ecosystem N availability (and thereby N2O emissions), but this is not the case: neither the N requirement for growth nor the lack of N availability for plant uptake do not enter the calculation of the N source. Rather it is simply based on the stoichiometric N requirement for litter decomposition compared to the soil N availability. If anything therefore, this representation is a soil decomposer demand based N fixation model. Other models have been published in which plant N demand for growth enters the calculation (Gerber et al. 2010, Meyerholt et al. 2016), and the labelling of the scheme of this paper as plant N demand based is therefore highly misleading. It is also incorrect to label such a representation of the nitrogen cycle as mechanistic (p 12 L 14), when it is based the assumption that biological N sources are in fact simply the consequence of the difference between organic N accumulation and ecosystem N losses (p 6 L 21-22).

Our wording is not interpreted as intended. We do not claim that the N source formulation is plant-demand based. The way the N source is implemented is clearly described in the method section and in the discussion (e.g., P21 L13: "The N source and its changes in LPX are implied by maintaining soil C:N ratio at observed PFT-dependent values.") . The N-demand based models by Gerber et al., 2010 and Meyerholt et al.,2016 are discussed in the section on N source (P21 L12 to P22 L5). The implicit assumption of the formulation in LPX is that the N source will increase in response to a higher NPP, involving higher N uptake, on the timescales of carbon turnover in vegetation and soils. We deleted the sentence on p1 L27-29 and reformulated the sentence on p1 L29-30 to not include the term N-demand in the abstract to avoid a potential misinterpretation by the reader: Assuming time-independent N sources in the model to mimic progressive N limitation of plant growth results in a decrease in emissions in contrast to the reconstruction. The text on P13 L12-L13 is modified to read: Biological nitrogen fixation (BNF), computed to match prescribed N:C ratios in soils (see methods), is implicitly assumed to adjust on the timescales of vegetation and soil carbon turnover to an increase in N demand by rising NPP and partly alleviating N limitation of plant growth in LPX-Bern. We also adjusted the caption of Fig. 6 and deleted the word mechanistic on P12 L14.

5) All these above points reduce the validity of the claim that these results adds confidence of future simulations (p. 24 L 21) of this model published by Stocker et al. 2013, because the model has been strongly modified in terms of the N source, the scaling of N2O emissions to N availability and the temperature sensitivity of that scaling of N2O emissions. Together with the lower than inferred changed in the terrestrial N2O source, these results would if anything suggest that the climate-N2O feedbacks suggested by Stocker et al. is low biased.

We clarified our view on this point and modified the text on P24 L21 to read: This may add confidence in the sign of the climate-$N_2O$ feedback projected with this model for the next centuries (Stocker et al., 2013), while suggesting that the magnitude of the projected feedback may be underestimated.

While I do realise that the authors have put a lot of effort into addressing the comments of the authors during revision, and there is no question that this modelling effort is novel, the combination of a model that does only capture a proportion of the observed signal with the overreach of the presentation of the results leads me to follow the recommendation of reviewer #2 and deny publication of the manuscript in

Biogeosciences.

Thank for honoring our effort during the revision and in the preparation of the manuscript. We apologize for our unintended misrepresentation of results in important places, cf. your point 1. We hope that our modifications and explanations now mitigate all remaining misunderstandings. We believe that the proposed modifications lead to a manuscript with a balanced presentation of our results and of the topic as a whole which is of interest for the paleo and nitrogen cycle/$N_2O$ community.

Sincerely,
Sönke Zaehle

**Reply to comments by Reviewer 1**

I'm glad to see the hypotheses and the most serious over-reach of the inferences of this paper have been removed. And model tuning has at least moved the model towards something sensible on BNF. Given that, now the elephant in the room in this paper is that the model doesn't do a very good job of simulating N2O over the deglaciation. The default model is over 1 TgN yr-1 out (figure 3). The best simulation (we presume, since we're only directly shown 3/9) is the purely theoretical simulation that assumes land area hasn't changed in the last 21 kyr (clearly not realistic).
Please see our response to the editorial comments regarding the data-model mismatch and the factorial simulations.

Looking at the basic constraints on the model, two seem pertinent:
- The model is tuned to 5.9 TgN yr-1 of N2O emissions at pre-industrial (p.9 l.15). This seems very sensible initially, as it is constraining the model to the observation-based data from the ice cores.
- The N 'BNF' input = the N output within the terrestrial realm. Putting aside whether this equates to an equilibrium, that assumption predicates that there is no net loss/gain of N in the terrestrial biosphere to the atmosphere or ocean (I'd call that equilibrium personally, but the authors prefer 'mass conservation or mass balance' (p6. l.22) and it amounts to the same thing). But what evidence is there for this? Especially over the deglaciation where there's a build up of soil carbon and nitrogen, it makes no sense for the inputs to equal the outputs. This determines much of the model behavior.
These latter statements are incorrect. There is no assumption that N input to the land biosphere equals N output. Rather, the sum of all N input fluxes to the land biosphere ($F_{in}$) equals the sum of all N loss fluxes from the biosphere ($F_{out}$) plus the change in N storage within the land biosphere ($dN/dt$). In other words, $dN/dt = F_{in} - F_{out}$. The sentence on P6 L22 in the introduction refers to this fundamental principle: "Following mass balance, losses of reactive N from ecosystems and changes in N stocks have to be compensated by N inputs, mainly by BNF."

LPX-Bern simulates a build-up of carbon and nitrogen in soils and vegetation over the deglaciation as for example illustrated in Fig. 6 or in Tab. A1 and noted for example on p15 L4-L6: "In response to the increased N input, the availability of reactive N remained roughly constant or increased in most land areas in the standard run, despite increased storage of N in plant and soil organic matter and accelerated nitrification and denitrification (Fig. 6C)."

We agree that it would make no sense to assume equilibrium with $F_{in}$ equal $F_{out}$ over the deglaciation. This is not the case in LPX and the statement that such an assumption determines much of the model behavior is not justified.

Despite these constraints, particularly the model tuning to pre-industrial N2O and a more sensible BNF, the model still doesn't do a good job (see figure 3, thick red line vs. thick green line). But this issue isn't clear as the results, particularly the figures, seem to be very selective.
For instance, why isn't BNF shown in figure 7? Why aren't the climate aspects shown in figure 3? Why aren't the yield effects shown in figure 3? Inferring from figure 3, if you put BNF on figure 7 it would show up as a negative of about 1 degree, i.e. less influential than temperature. Acknowledging this would change the main outcomes of the paper, as well as make clear that the model isn't reliable.
We reject these statements. All figures have been carefully selected to best illustrate the results from our simulations.  Regarding individual points:
- Spatial and integrated changes in BNF are already shown in Fig. 6 for the standard setup and for the setup with constant local N source. We follow the recommendation of the editor and show now the

result from our standard simulation and the result from the setup with constant local N source in Fig. 3. The influence of changes in environmental drivers and of temperature-induced changes in yield on $N_2O$ emissions are shown in Fig. 7 and there is no need to repeat the information in Fig. 3.
- We are not exactly sure what the wording of the reviewer means, but guess that with "put BNF" the reviewer refers to the difference in results between the standard model setup and the setup with constant local N source and with "1 degree" the reviewer means a change in $N_2O$ emissions by -1 TgN yr$^{-1}$ (?). In case our interpretation is correct, this proposal is scientifically not permissible for the following reason.
The influence of environmental parameters on deglacial $N_2O$ emissions is quantified for the standard model setup. The changes attributed to the individual environmental drivers are roughly additive and their sum is comparable to the result from the standard simulation (the non-linear term is given in the Fig 7). This, however, does not hold when factoring in the change due to a structural model modification, here in N source.
Repeating the same factorial simulations for the setup with constant local N source would yield different results for each climatic parameter. In such a model setup changes in temperature and precipitation have very likely a negligible influence on $N_2O$ emissions.  Such an additional set of factorial simulations would not change the main outcomes of the paper.

This is in addition to the fact that looking at constant BNF is no more valid than looking at constant temperature, and no more can be inferred from it. You wouldn't hold temperature constant over the deglaciation and say it shows the temperature limitation effect. Similarly, it's nonsense to set up a model to need BNF, restrict the BNF, and then show the results as meaning something other than the model is being used outside of its operating remit. And that isn't even touching on the fact that the BNF in the model can be reduced 5-fold and apparently have no effect on the proportion of N2O emissions, while changing almost all other carbon and nitrogen fluxes (Table A1), but still being held as being valid by the authors.

Once you strip away all of the hearsay and 'context' that attempts to make this paper's results significant, you're left with an experiment with a poor model that doesn't show much. The authors even acknowledge as much in the last two lines of the abstract – it might support biological controls, or flexible C:N, or it might be due to N2O emission differences. i.e. it doesn't take us any further forward scientifically.
We reject these statements and the disrespectful wording.
- Factorial simulations are widely used in the scientific community and are rather standard than "nonsense".
- As documented in Tab. A1, percentage changes in nitrogen fluxes from LGM to preindustrial are comparable in version v1.4 and v1.4N, though typically a bit smaller in v1.4N than in v1.4 (e.g. 12 % versus 15 % for denitrification). Therefore, relative changes in $N_2O$ emissions are also similar as absolute emissions have been scaled to match preindustrial emissions of 5.9 TgN yr$^{-1}$.
- We present for the first time simulations of terrestrial $N_2O$ emissions for the last 21,000 years and compare the results to a novel reconstruction of these emissions. We provide an in-depth analysis and interpretation using factorial simulations and document the results using 11 figures. It is not appropriate to dismiss this work as "hearsay" and "context".
- LPX-Bern and its modules are described in the literature. The model is included in many international modelling intercomparison studies including those targeted to establish the global carbon, methane, and $N_2O$ budget by the Global Carbon Budget.

We would like to point to the assessment by reviewer #2 who rated the scientific significance and the scientific quality as well as the presentation quality of our manuscript as excellent.

Some specific points (this is not a comprehensive list, just a few things I noticed and couldn't not mention):

P1 l.21 Normative value judgments like 'remarkably', without any evidence or context to back-up why something is remarkable, have no place in an academic paper, especially not in the abstract.
We prefer to highlight to the reader that the reconstructed changes in $N_2O$ emissions are fast.
p.8 l.19 The authors provide no reference for the "pre-industrial" BNF of 128 TgN yr-1. I speculate that is because they know that it's a mis-representation. Assuming that 128 TgN yr-1 comes from Cleveland et al. (2013), as that's basically the only legitimate source of that number, it isn't pre-industrial BNF. Just like where they say in their response to reviewers that 128 is the value "preferred" by me as a reviewer, (which they have no evidence of) they're hijacking numbers for their own convenience with no regard to the truth.
p.8 l.22 The authors have lifted the generalization of 100 – 150 TgN yr-1 BNF from my review without any citation to original sources.
We reject this unfair criticism. We provide 7 references for BNF on non-agricultural ecosystem in this paragraph on P 8 L18-L32. We deleted the clause "with most published estimates around 100 to 150 TgN yr$^{-1}$".
Figure 3. Green and red lines on this plot are not color-blind friendly. It's really disrespectful to use color schemes that are exclusionary in 2019. Same for figure 10.
A dashed red and a solid green line as shown in Fig. 3 is easily distinguishable and color-blind friendly. We checked Fig. 10 with a color-blind person who find the colors to be distinguishable.  Nevertheless, we will aim to further improve the color schemes for even better visibility in Fig.10 to the extent possible.
Terminology The terminology (in this case, which simulation has what description or short tag) is inconsistent. The terminology should be stated once in the methods and then clear and consistent in all places thereafter.
The description and the terminology referring to individual simulations appears clear as there has not been any misunderstanding during the past reviews.

[revised manuscript text omitted]

---

## Author Response (AR3)

**Reply to comments by editor and reviewers**

We thank the editor and the reviewers for their time and effort to assess our manuscript and for editorial guidance. The comments by the editor and reviewers are given in black, our reply in blue, and text changes implemented in the manuscript in red. A manuscript version with the changes highlighted is attached to this reply.

**Reply to comments by Editor**

Dear Authors,

I now have two reviews of your manuscript. Despite reservations about the process and previous revisions of the manuscript, R1 deems your manuscript suitable for publication. The second reviewer is more positive about the contribution of your work and also agrees that this should be published. I have read their assessments and upon reading your manuscript myself, I am recommending minor revisions. I think this paper makes an interesting contribution to the literature despite some reservations in the model-data disagreement (and so interpretation). As well as revising your manuscript to address the reviewer comments, I make three further points.

1. Whilst I appreciated the effort to contextualise the importance of your study and relationship to future predictions, I struggled with the considerable focus afforded mycorrhizal association/systems in the introduction. It seems very (completely?) unrelated to the focus on your manuscript. You might say the same about the Reich citation, it is a stretch - at best. I'd really prefer to see a few lines afforded your period of interest and a better link to the study in part 1.

We shortened the text on mycorrhizal system by deleting the text on line 5 to 9 on page 3. The reference to Reich is deleted. We expanded the text on part I in the introduction to read:

In part I of this study (Fischer et al., 2019), this earlier work was extended to reconstruct the evolution in terrestrial versus oceanic emissions of $N_2O$ from the Last Glacial Maximum (LGM; 21 ka BP) to the late preindustrial Holocene using novel, high-precision $N_2O$ stable isotope and concentration data. The age scales of the records were aligned with the absolutely counted GICC05 age scale (Rasmussen et al., 2006) using high-resolution methane measurements leading to particularly small age-scale uncertainties around past rapid warming events, namely, at the onset of the Bølling/Allerød Northern Hemisphere warm period and at the onset of the Holocene interglacial. The ice-core data of $N_2O$ concentration and $\delta^{15}N(N_2O)$ were used in a model deconvolution to determine changes in global marine and terrestrial $N_2O$ emissions. The results indicate that $N_2O$ emissions from land and ocean increased over the deglaciation largely in parallel by 1.8±0.3 and 0.7±0.3 TgN $yr^{-1}$, respectively, relative to the Last Glacial Maximum level. However, during the abrupt Northern Hemisphere warming events, terrestrial emissions changed more rapidly than marine emissions. The ice core data reveal large step-like changes in terrestrial emissions by up to 1 TgN $yr^{-1}$ at the onset of the warming events, realized over decades and, given the proxy data resolution, last 200 years at maximum. These multi-decadal timescales are also relevant for 21st century climate projections and much longer than accessible in typical laboratory or field experiments. Further, the abrupt $N_2O$ emission changes indicate a very high sensitivity of the terrestrial C-N cycle to environmental change. But a detailed process-based investigation of terrestrial

$N_2O$ emission changes over the deglaciation, rapid past warming events, and the Holocene warm period, and their links to the flow of N and C in land ecosystems, has not been attempted before.

2. In the discussion of modelling BNF - one (potentially) overlooked point is that while models that simulate BNF as a function of ET may not simulate marked change, they will simulate different magnitudes if they simulate different ET (which they do). Overall I do take your point in the discussion about it being hard (but by no means very difficult) to carry out a more quantitative assessment about how BNF approaches may impact your results. Nevertheless, as I look at Figure 3 I can't help wondering how robust the interpretation can be to the assumed BNF method. If a model makes an alternative BNF assumption, they likely arrive at a different answer and so interpretation. I don't have a specific recommendation here beyond highlighting how I suspect others may interpret the results.

We agree that it is not very difficult to replace the BNF module per se by other simplified parameterizations, though testing the implementations and running multiple 20,000-year long simulations would still require some work. As this MS is already long, we do not want to add additional complexity and leave this to future efforts. The conclusion that an increase in N source provides a plausible explanation of the deglacial increase in $N_2O$ emissions does, in our view, not critically depend on the details of how the N source is implemented as long as the N source adjusts to deglacial change. The last sentence on p22, l13 is modified to read:

However, a corresponding quantitative analysis is beyond the scope of this study and left to future efforts.

3. The key point for me is how well the model captures the data and as such what the interpretation of the drivers can be. The text points out that the model simulates the increase in N2O to be about 47% lower. That is a sizeable amount. I do not currently feel that the abstract appropriately captures the remaining gap. Arguably, you will suggest that the model-based analysis has closed some of this gap (potentially), which is fair. But I think a reader may well expect to see a more open concluding sentence about the need to search for alternative or complementary explanations. I'd expect to see this sentiment in the discussion too.

We added the following text at the end of the abstract: The deglacial $N_2O$ record provides a constraint for future studies.  to point towards the need of further research. We also changed "as well as" to "and" in the first sentence to remain within the 300-word limit for the abstract

We added the following text in the last paragraph of the discussion on p24: Future research efforts are needed to test the robustness of our results and to further evaluate the potential role of alternative or complementary explanations. In particular, the significantly smaller amplitude of our modeled $N_2O$ changes compared to the data reconstruction justify further research.

Best wishes,

Martin De Kauwe

**Reply to comments by Reviewer #3**

Review of "N2O changes from the Last Glacial Maximum to the preindustrial - part II: terrestrial N2O emissions and carbon-nitrogen cycle interactions" by Fortunat Joos, Renato Spahni, Benjamin D. Stocker, Sebastian Lienert, Jurek Müller, Hubertus Fischer, Jochen Schmitt, I. Colin Prentice, Bette Otto-Bliesner, and Zhengyu Liu.

In the submitted manuscript – by now apparently into the third revision – Joos and co-authors investigate the changes in terrestrial N2O emissions from the last glacial maximum to preindustrial times. They describe their model setup and the model experiments performed, and describe model results, as well as results from two sensitivity experiments. The authors can explain about half of the change in terrestrial N2O emissions between LGM and preindustrial and discuss some factors that might lead to the underestimates by their model.

Overall, the manuscript is very well written, and about ready for publication. I am torn between recommending publication as is, and publication with small changes. However, the manuscript should definitely be published, as it is a pioneering effort in modelling the changes in N2O emissions from the LGM to PI. While the authors cannot yet explain the full change, this publication is required by the community as a base to build upon in improving our understanding of well-documented biospheric changes from the past to the present.

Thank you. We appreciate this supportive comments.

I have no major issues with the manuscript. While the previous discussion between authors and reviewers indicates that there may have been some rather strong claims in previous iterations of the paper, I can find no fault in this respect with the present version of the manuscript. Claims by the authors appear well-supported by the author's results, and the model appears to be documented adequately. It may well be that some details of the implementation of the Nitrogen cycle in LPX-Bern leave something to be desired in the light of the most recent findings, but personally I am rather glad that there still is room for improvement, as it gives us something to build upon in the future.

Our scientific conclusions have not changed since the first submission of this work. We included in the first submitted version a discussion on shortcomings and potential caveats, e.g. on model simplifications and unknown changes in the $N_2O$ yield per reactive nitrogen transformed. However, presenting complex issues is always a challenge and sometimes formulations are perceived differently than intended.

However, there are two minor issues that may warrant revisiting the manuscript. Looking at the ice core reconstruction in Figure 3, it is striking that more than half of the emission change was realised in two very rapid steps. The model reproduces these step-like changes, although it underestimates the magnitude and the rapidity of these step-changes, as discussed in the manuscript. Later on in the Holocene, however, a further quarter of the reconstructed emission change was realised as a slow upward trend in N2O emission, but the model completely fails to reproduce this upwards trend, it rather shows nearly constant emissions. I may have overlooked it, but so far I am missing a discussion of this feature.

The following paragraph is added in the discussion section 6.2 on page 23, line 14:

 LPX-Bern simulates a slight decrease in $N_2O$ emissions over the Holocene period, whereas the reconstruction shows an increase by about 0.4 TgN yr$^{-1}$ over this period. The exact reason for this discrepancy between modelled and reconstructed change is unclear. The model results suggest that the global $N_2O$ emission change results from the difference of in some regions large positive minus in other regions large negative emission changes (Fig. 4c) and that these regional emission changes depend sensitively on regional changes in precipitation and temperature (Fig.8). Thus, it may remain somewhat challenging to simulate the balance between positive and negative emission changes correctly, given uncertainties in land models and the climatic drivers as well as in the deconvolution of the ice core concentration and isotope data.

A second issue is that I find some of the Figures slightly confusing, but I appreciate that this may be a matter of personal taste. In Figures 4-6, 8, and 9 I was irritated by the fact that I needed to read the Figure caption very carefully in order to understand what was shown. I was expecting to see a Figure title, indication Figure content, and overlooked the units and quantity shown next to the colour bar – obviously my mistake, but maybe the authors can find a way to make this slightly clearer.

Thank you for this suggestion. We added titles to Fig. 4-6 and 8-9.

**Reply to comments by Reviewer #1**

On this third set of major revisions the authors have finally succeeded: they have fixed enough of the major problems to be scientifically viable. In response to comments from the previous Associate Editor, the authors have reduced their claims adequately and amended the most misleading parts of the paper. I could continue to argue that some of the referencing is not done to a high standard, the terminology is inconsistent, etc. etc. but I am worn down. I no longer care.

The reason I recommended this paper be rejected was I felt there was a lack of integrity in the way that the results were presented, both initially and on the second round of review. Given my own, and colleagues', experiences of peer review, I was surprised the paper was given a third chance, after my opinion and that of the previous Associate Editor agreed that it did not meet the standard required for publication after major revisions. What, I wondered, justified a paper which continued to be misleading, and had data representation that was highly dubious, being reviewed again? Would this paper have been given the same chances if the first author was not an Associate Editor for the journal? Would a junior researcher be given this benefit of the doubt? There is, at least the appearance of, cronyism.

The wider issue here is one of integrity and respect for colleagues. Of 7 responses made by the authors to my statements in this latest round of review, 1 they deem "incorrect", 3 they "reject", 2 they make no change, and just 1 is acknowledged to have been changed. That 1 is a color scheme on 1 figure (of 2 suggested color scheme changes for better inclusivity). The previous AE fared slightly better than me, having only 1 of their comments rebuffed as "criticism is not justified", 1 telling the previous AE that they are "incorrect", and the remaining 3 where the authors took some responsibility for the paper they wrote. The authors, in the response to reviewers, accuse me of being "disrespectful". I suggest if we want to talk about "disrespectful", let's start with the authors' approach to peer review. Scientific integrity means that you present your results in the clearest and most honest way. It means when two fellow scientists point out that your plots are misleading, you change them rather than argue about it.

These repeated iterations of major revision review use reviewers as additional, unpaid and unacknowledged labor, and are disrespectful of reviewer time. The role of a reviewer is not to help the authors get the paper published, it is to ensure that papers meet a good standard of scientific quality. When papers are resubmitted repeatedly, despite clear reasons for concern, it looks as if the reviewers are considered in the service of the author. Authors and journals have a lot to gain from a paper being published (academic reputation and money, respectively). The reviewers get nothing. How journals and authors treat unpaid reviewers says a lot. This is abusing the work of reviewers, who are usually more junior scientists, for the benefit of senior scientists.

I imagine the authors will be content to hear that I will not be reviewing for anyone on the author list again, as either an author or an editor. They have demonstrated it is not worth my time, energy, or scientific integrity. I'm also disappointed in Biogeoscienes approach to its AEs, and treatment of reviewers, in this case.

We always aim to present findings in a balanced, scientific and technically correct, and transparent way and are always open to adjust wording and figures or to add results from additional simulations to improve clarity. We feel we demonstrated this in our responses to the first set of review comments.

As noted by the reviewer, we were not able to accommodate most of the comments by the reviewer #1 and the editor after the second-round of peer review, partly because the criticisms were, in our view, not justified or demonstrably wrong. Reviewer #2 had no comments and recommended publication. The revision of the manuscript after this second-round was therefore in general rather minor and, in the words of the former associate editor, of "*cosmetic nature*".
We welcome that reviewer #1 has revised his assessment from "reject" to "accept as is", despite the modest revision.

The insinuations and accusations towards the editorial board of Biogeosciences by the reviewer #1 are not justified and misplaced. These public allegations by the anonymous reviewer #1 are misplaced as the Editors-in-chief have not the possibility to respond directly. Further, the EGU Publications Committee acts as the Ombudsperson (publications@egu.eu)  for such matters.
They are unfounded. There are clear procedures by the journal in case the authors disagree with an editorial decision (https://www.biogeosciences.net/for_authors/appeals_and_complaints.html). This option is available to junior and senior researchers alike.
In this specific case, it was the associate editor himself who asked the editorial board for advice and forwarded our reply to the review and editor comments to the editorial board (e-mail dated 21.12.2019). We were advised by the associate editor (e-mail 31.12.2019) to forward our appeal to the editorial board.
The manuscript is rated excellent in all three available categories ("scientific significance", "scientific quality", and "presentation quality") by reviewer #2 and #3 and the new associate editor recommends publication after minor revision.
The  associate editors and the editorial board followed established procedures and the outcome of the reassessment supports the steps taken by the editorial board.

[revised manuscript text omitted]